# Hypergraph Neural Networks through the Lens of Message Passing: A Common Perspective to Homophily and Architecture Design

**Lev Telyatnikov**                                     *lev.telyatnikov@uniroma1.it*
*Sapienza University of Rome*

**Maria Sofia Bucarelli**                               *mariasofia.bucarelli@uniroma1.it*
*Sapienza University of Rome*

**Guillermo Bernardez**                                 *guillermo_bernardez@ucsb.edu*
*University of California, Santa Barbara*

**Olga Zaghen**                                         *o.zaghen@uva.nl*
*University of Amsterdam*

**Simone Scardapane**                                   *simone.scardapane@uniroma1.it*
*Sapienza University of Rome*

**Pietro Liò**                                          *pl219@cam.ac.uk*
*University of Cambridge*

**Reviewed on OpenReview:** *https://openreview.net/forum?id=8rxtL0kZnX*

## Abstract

Most of the current learning methodologies and benchmarking datasets in the hypergraph realm are obtained by *lifting* procedures from their graph analogs, leading to overshadowing specific characteristics of hypergraphs. This paper attempts to confront some pending questions in that regard: **Q1** Can the concept of homophily play a crucial role in Hypergraph Neural Networks (HNNs)? **Q2** How do models that employ unique characteristics of higher-order networks perform compared to lifted models? **Q3** Do well-established hypergraph datasets provide a meaningful benchmark for HNNs? To address them, we first introduce a novel conceptualization of homophily in higher-order networks based on a Message Passing (MP) scheme, unifying both the analytical examination and the modeling of higher-order networks. Further, we investigate some natural strategies for processing higher-order structures within HNNs (such as keeping hyperedge-dependent node representations or performing node/hyperedge stochastic samplings), leading us to the most general MP formulation up to date –MultiSet. Finally, we conduct an extensive set of experiments that contextualize our proposals.

## 1 Introduction

Hypergraph learning techniques have multiplied in recent years, demonstrating their effectiveness in processing higher-order interactions in numerous fields, spanning from recommender systems (Yu et al., 2021; La Gatta et al., 2022), to bioinformatics (Zhang et al., 2018; Yadati et al., 2020) and computer vision (Li et al., 2022; Xu et al., 2022). However, so far, the development of Hypergraph Neural Networks (HNNs) has

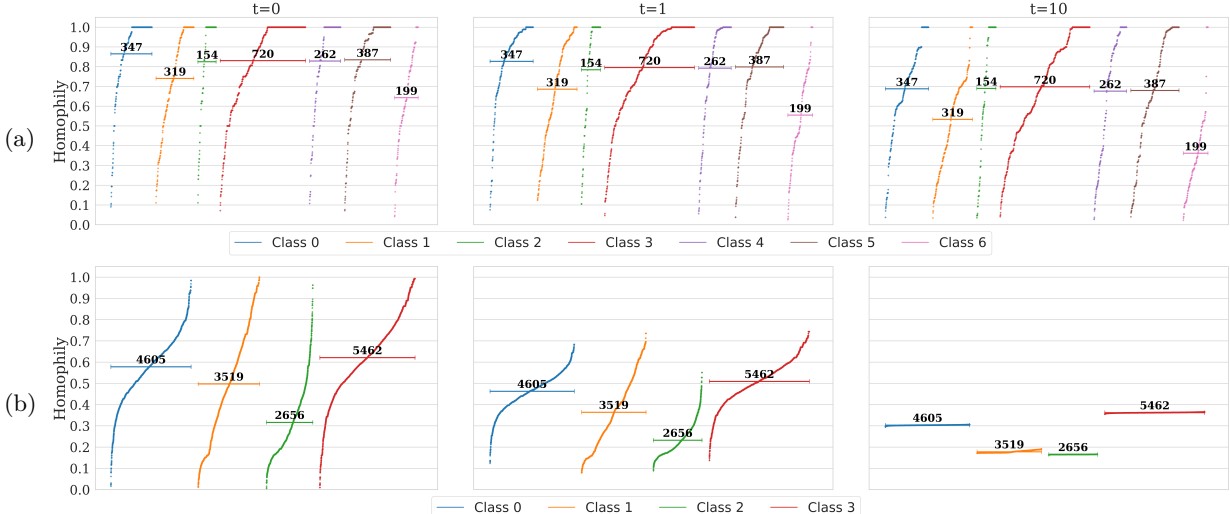

Figure 1: Node Homophily Distribution for CORA-CA (a) and 20Newsgroups (b). The plots depict node homophily scores computed using Equation 2 at $t = 0$, $t = 1$, and $t = 10$. For each dataset class, points are sorted in ascending order of homophily and visualized sequentially along the x-axis. Horizontal lines represent the mean homophily score for each class, with the numbers above indicating the total number of points in each class.

been largely influenced by the well-established Graph Neural Network (GNN) field. In fact, most of the current methodologies and benchmarking datasets in the hypergraph realm are obtained by *lifting* procedures from their graph counterparts.

The advancement of hypergraph research has been significantly propelled by drawing inspiration from graph-based models (Chien et al., 2022; Feng et al., 2019; Yadati et al., 2019), but it has simultaneously led to overshadowing hypergraph network foundations. We argue that it is now the time to address fundamental questions in order to pave the way for further innovative ideas in the field. In that regard, this study explores some of these open questions to understand better current HNN architectures and benchmarking datasets along three axes. **Q1** Can the concept of homophily play a crucial role in HNNs, similar to its significance in graph-based research? **Q2** Given that current HNNs are predominantly extensions of GNN architectures adapted to the hypergraph domain, are these extended methodologies suitable, or should we explore new strategies tailored specifically for handling hypergraph-based data? **Q3** Are existing hypergraph benchmarking datasets truly *meaningful* and representative to draw robust conclusions?

To begin with, we explore how the concept of homophily can be characterized in complex, higher-order networks. Notably, there are many ways of characterizing homophily in hypergraphs –such as the distribution of node features, the analogous distribution of the labels, or the group connectivity similarity (as already discussed in Veldt et al. (2023)). In particular, this work places the *node class distribution* at the core of the analysis, and introduces a novel definition of homophily that relies on a Message Passing (MP) scheme. This enables us to analyze both hypergraph datasets and architecture designs from the same perspective, as well as to successfully describe model performances (see Section 3 and 5.2).

Next, we shift our focus towards the design of hypergraph-specific methodologies that HNNs could benefit from, no longer relying on lifting strategies. To this end, after examining state-of-the-art HNN architectures, we first describe the most versatile MP framework up to date, called MultiSet (see Section 4.2). Our formulation, which enables hyperedge-dependent node representations and residual connections, inherently generalizes most existing HNN frameworks and models, including AllSet (Chien et al., 2022), UniGCNII (Huang & Yang, 2021), EDHNN (Wang et al., 2023) and a recently proposed model WHATsNet that allows for hyperedge-dependent node representations (Choe et al., 2023).

Subsequently, to facilitate a comparison between standard lifted hypergraph models and a new class of hypergraph-specific architectures, in Section 4.3, we present the realization of a hypergraph model—MultiSetMixer—that accommodates multiple hyperedge-based node hidden representations.[1] By allowing multiple node representations based on hyperedge memberships—meaning the same node is represented differently depending on the hyperedges it belongs to—this new class of architectures results in an exponential increase in potential node representations as the number of connections within the hypergraph network grows. To address this complexity, we propose and evaluate novel connectivity-based mini-batching strategies tailored to the specific characteristics of hypergraph networks, as detailed in Section 4.4. These sampling procedures not only facilitate processing large hyperedges but also give rise to an interesting behavior –which we term connectivity-based distribution shift– thoroughly discussed in the paper (see Section 5.3).

Last but not least, we provide an extensive set of experiments that, driven by the general questions stated above, aim to gain a better understanding on fundamental aspects of hypergraph representation learning. In fact, the obtained results not only help us contextualize the proposals introduced in this work but open new challenges in hypergraph modeling, such as signal oversquashing caused by large hyperedges, which impacts the performance of all HNNs (see Section 5.1) or the tendency of HNNs to ignore connectivity for common benchmark datasets (see Section 5.4.1).

**Summary of contributions**:

- We introduce a novel definition of homophily for hypergraphs, capable of effectively describing HNN model performances (**Q1** and **Q3**, Sections 3 and 5.2).

- We present the novel MultiSet framework, which incorporates hyperedge-dependent node representations and generalizes most of the existing hypergraph models in the literature (**Q2**, Section 4.2).

- We explore novel mini-batch sampling strategies for architectures with and without hyperedge-dependent node classification (**Q2**, Section 4.3).

- We perform a large set of experiments assessing benchmarking both datasets and HNN architectures, as well as connecting the proposed MP homophily with models' performance (**Q1**, **Q2**, **Q3**, Sections 3 and 5).

## 2  Related Works

**Homophily in hypergraphs.**  Homophily measures are typically defined only for pairwise relationships. In the context of Graph Neural Networks (GNNs), many of the current models implicitly use the homophily assumption, which is shown to be crucial for achieving a robust performance with relational data (Zhou et al., 2020; Chien et al., 2020; Halcrow et al., 2020). Nevertheless, despite the pivotal role that homophily plays in graph representation learning, its hypergraph counterpart mainly remains unexplored. In fact, to the best of our knowledge, Veldt et al. (2023) is the only work that faces the challenge of defining homophily for higher-order networks. This work introduces a framework in which homophily is quantified through group interactions, measuring the distribution of classes among hyperedges (see Appendix G for a detailed description). However, their definition of homophily is restricted to uniform hypergraphs –where all hyperedges have exactly the same size–, which hinders its practical application to a great extent. Clique-expanded (CE) homophily, as used in Wang et al. (2023), is calculated by determining node homophily Pei et al. (2020) on the graph derived from the clique-expanded hypergraph. Thus, CE homophily is not directly defined on the hypergraph but requires a clique expansion. It is important to note that this expansion is a non-invertible transformation, which leads to a loss of information. Refer to Appendix G for a detailed explanation of these homophily measures and a comparison with the proposed method.

---

[1]Note that MultiSetMixer is a simple yet powerful model architecture that serves primarily as a baseline, providing an example of a model that incorporates and employs the unique characteristics of hypergraph networks.

**Hypergraph Neural Networks.** Numerous machine-learning techniques have been developed for hypergraph data processing. A common early approach is transforming hypergraphs into graphs via clique expansion (CE), where each hyperedge is replaced with edges between all pairs of vertices in the hyperedge, enabling graph-based algorithm analysis (Agarwal et al., 2006; Zhou et al., 2006; Zhang et al., 2018; Li & Milenkovic, 2017).

Hypergraph Neural Networks (HNNs) have also been applied to semi-supervised learning. One early method extends graph convolution by incorporating the normalized hypergraph Laplacian (Feng et al., 2019), with weighted CE as spectral convolution (Dong et al., 2020). HyperGCN (Yadati et al., 2019) uses mediators for a reduced CE, lowering the number of edges required to represent a hyperedge, and applying spectral convolution for information diffusion. Hypergraph Convolution and Hypergraph Attention (HCHA) (Bai et al., 2021) introduces modified normalizations and attention weights dependent on node and hyperedge features.

However, CE can result in the loss of structural information, leading to suboptimal performance (Hein et al., 2013; Chien et al., 2022). These models often perform best with shallow architectures, while deeper layers can cause oversmoothing (Huang & Yang, 2021). Recent work has tried to mitigate oversmoothing with residual connections but still relies on hypergraph Laplacians and clique expansion (Chen & Zhang, 2022). Another method, line expansion (LE), treats vertices and hyperedges equally and models vertex-hyperedge pairs to induce a homogeneous structure, though both CE and LE demand significant computational resources (Yang et al., 2020).

Another research line focuses on two-stage message passing: nodes communicate with hyperedges, which then relay information back to the nodes (Wei et al., 2021; Yi & Park, 2020; Dong et al., 2020; Arya et al., 2020; Huang & Yang, 2021; Yadati et al., 2020). HyperSAGE (Arya et al., 2020) exemplifies this approach, offering improvements over spectral methods but limited by a single learnable transformation and inefficient computation due to nested loops (Chien et al., 2022).

The work of Chien et al. (2022) introduced AllSet, a general framework that describes HNNs through the composition of two learnable permutation invariant functions, defining a two-step message passing based mechanism –from nodes to hyperedges, then back from hyperedges to nodes. In particular, AllSet is shown to generalize most commonly used HNNs, including all clique expansion based (CE) methods, HNN (Feng et al., 2019), HNHN (Dong et al., 2020), HCHA (Bai et al., 2021), HyperSAGE (Arya et al., 2020) and HyperGCN (Yadati et al., 2019). Chien et al. (2022) also proposes two novel AllSet-like learnable layers: the first one –AllDeepSet– exploits Deep Set (Zaheer et al., 2017), and the second one –AllSetTransformer– Set Transformer (Lee et al., 2019), both of them achieving state-of-the-art results in the most common hypergraph benchmarking datasets. Concurrent to AllSet, Huang & Yang (2021) also aimed at designing a common framework for graph and hypergraph NNs, and its more advanced UniGCNII method leverages initial residual connections and identity mappings in the hyperedge-to-node propagation to address over-smoothing issues; notably, UniGCNII does not fall under the AllSet framework due to these residual connections. Likewise, the more recent EDHNN model (Wang et al., 2023) also goes beyond this framework by incorporating hyperedge-dependent messages from hyperedges to nodes, a step closer to the hyperedge-dependent node representations that we propose in this work.

Recently, Choe et al. (2023) introduced WHATsNet, the first model capable of producing hyperedge-dependent node representations. In particular, WHATsNet model is a particular instance of the MultiSet framework introduced in our work (see Section 4.2), but its architecture is specifically designed to perform hyperedge-dependent node classification tasks –where each node has as many labels as the number of hyperedges it belongs to. This prevents benchmarking this model in our evaluation, where we focus on standard node-level classification tasks.

Please refer to Appendix A for an extended literature review.

**Notation.** A hypergraph is an ordered pair of sets $\mathcal{G} = (\mathcal{V}, \mathcal{E})$, where $\mathcal{V}$ is the set of nodes and $\mathcal{E}$ is the set of hyperedges. Each hyperedge $e \in \mathcal{E}$ is a subset of $\mathcal{V}$, i.e., $e \subseteq \mathcal{V}$. A hypergraph is a generalization of the concept of a graph where (hyper)edges can connect more than two nodes. A vertex $v$ and a hyperedge $e$ are said to be incident if $v \in e$. For each node $v$, we denote its class by $y_v$, and we denote by $\mathcal{E}_v = \{e \in \mathcal{E} : v \in e\}$ the subset of hyperedges in which it is contained. We represent by $d_v = |\mathcal{E}_v|$ the node degree. The set of classes is represented by $\mathcal{C} = \{c_i\}_{i=1}^{|\mathcal{C}|}$.

## 3 Homophily Metrics in Hypergraphs

In this section, we present a new definition of homophily that employs a two-step message passing scheme applicable to general, non-uniform hypergraphs, in contrast to the definition by Veldt et al. (2023). In essence, our definition focuses on capturing hyperedge interconnections by the exchange of information following the message passing scheme. Following that, we illustrate its applicability in examining higher-order networks through qualitative analysis. Finally, we demonstrate the applicability of the proposed concept by deriving a $\Delta$ homophily measure. In Section 5.2, we show its capability to describe HNNs' performance. These play a pivotal role in our attempt to answer the fundamental question **Q1** raised in the Introduction.

**Message passing homophily.** Given a hyperedge $e \in \mathcal{E}$, we define the 0-level hyperedge homophily $h_e^0(c)$ as the fraction of nodes within $e$ that belong to class $c$, i.e.

$$h_e^0(c) = \frac{1}{|e|} \sum_{v \in e} \mathbb{1}_{y_v = c}. \tag{1}$$

This score describes how homophilic the initial connectivity is with respect to class $c$. Computing the score for each class $c_i \in \mathcal{C}$ generates a categorical distribution for each $e \in \mathcal{E}$, i.e. $h_e^0 = [h_e^0(c_0), \ldots, h_e^0(c_{|C|})]$. Using this information as a starting point, we calculate higher-level homophily measurements for both nodes and hyperedges through the two-step message passing approach. Formally, we define the $t$-level homophily score as

$$h_v^{t-1} = \text{AGG}_{\mathcal{E}} \left( \{h_e^{t-1}(y_v)\}_{e \in \mathcal{E}_v} \right), \tag{2}$$

$$h_e^t(c) = \text{AGG}_{\mathcal{V}} \left( \{h_v^{t-1}\}_{v \in e, y_v = c} \right), \tag{3}$$

where $\text{AGG}_{\mathcal{E}}$ and $\text{AGG}_{\mathcal{V}}$ are functions aggregating edge and node homophily scores, respectively (we consider the mean operator in our implementation). We note that our homophily measure enables the definition of a score for each node and hyperedge for any neighborhood resolution.

**Qualitative analysis.** One straightforward way to make use of the message passing homophily measure is to visualize how the node homophily score dynamically changes, as described in Eq. 2. Figure 1 depicts this process for non-isolated nodes on CORA-CA and 20NewsGroup datasets (Appendix G shows the plots of the rest of considered datasets, which in turn are described in Section 5). Looking at CORA-CA (Figure 1 (a)), we note that there are a significant number of nodes with high 0-level homophily at each class (except number 6), and this homophily distribution is kept mostly unchanged as we move to the 1-hop neighborhood ($t = 1$). Interestingly, the same trend holds even when shifting to the 10-level node homophily –only classes 1 and 6 show a relevant drop in highly homophilic nodes. This suggests the presence of isolated homophilic subnetworks within the hypergraph. In contrast, 20Newsgroups dataset (Figure 1 (b)) displays relatively low node homophily scores from the 0-level (specifically for class 2, with a mean value around 0.3). Moving to $t = 1$, there is a significant decrease in the homophily scores for every class. Finally, at time step $t = 10$, we can observe that all the classes converge to approximately the same homophily values within each class. This convergence and low homophily scores suggest that the network is highly interconnected.

**$\Delta$ homophily.** Rather than measuring homophily in individual discrete timestamps, we next derive $\Delta$ *homophily* measure, which offers a dynamic perspective to explore hypernetworks. This measure is based on the assumption that if the 1-hop neighborhood of a node $u \in \mathcal{V}$ is predominantly homophilic, then the change in homophily score between two consecutive timestamps will be small. Conversely, a substantial change in

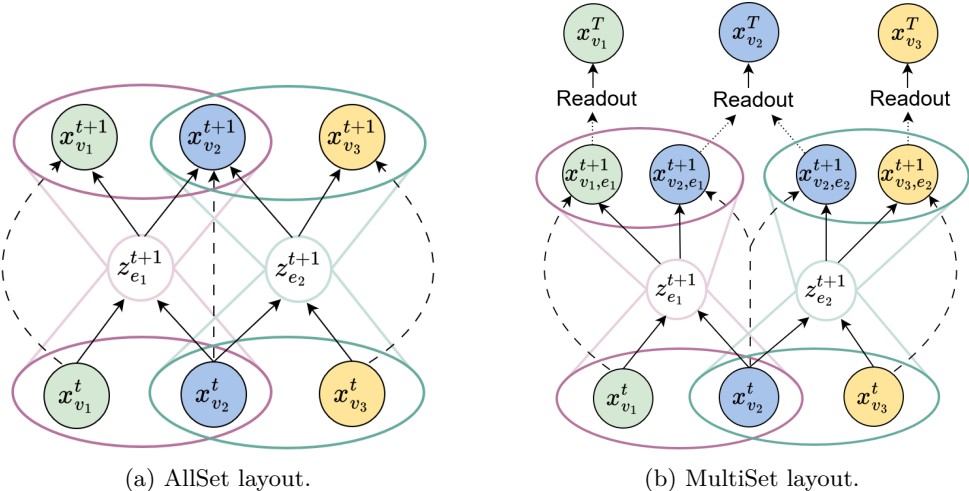

(a) AllSet layout.        (b) MultiSet layout.

Figure 2: Visual representation of the AllSet and MultiSet frameworks.

$u$'s homophily implies that the node resides in a neighborhood characterized as heterophilic. Specifically, for each node, we quantify the homophily change after $t$-th step of message passing by computing the difference between the node homophily at that step from its value at previous one, $t-1$. Subsequently, we look at the proportion of nodes whose homophily difference is below a certain threshold $\mu \in \mathbb{R}^+$, i.e.

$$\Delta_\mu^t = \frac{1}{|\mathcal{V}|} \sum_{v \in \mathcal{V}} \mathbb{1}_{\left| h_v^t - h_v^{t-1} \right| < \mu}. \tag{4}$$

As we show in Section 5.2, this dynamic measure becomes a helpful tool to analyze the performance of HNNs.

## 4 Methods

Current HNNs aim to generalize GNN concepts to the hypergraph domain, and are specifically focused on redefining graph-based propagation rules to accommodate higher-order structures. In this regard, the work of Chien et al. (2022) introduced a general notation framework, called AllSet, that encompasses most of currently available HNN layers, including CEGCN/CEGAT, HNN (Feng et al., 2019), HNHN (Dong et al., 2020), HCHA (Bai et al., 2021), HyperGCN (Yadati et al., 2019), as well as AllDeepSet and AllSet-Transformer presented in the same work (Chien et al., 2022). This section first revisits the original AllSet formulation, and then introduces a new framework –MultiSet– which extends AllSet by allowing multiple hyperedge-dependent representations of nodes. Finally, we present a particular realization of an architecture that takes into account hyperedge dependent-node representations. Then, we open a discussion on the scalability issues of hypergraph architectures and propose a novel mini-batching scheme. In contrast to previous formulations, our proposed framework and implementation are inspired by hypergraph needs and features, and motivated by the fundamental question **Q2**.

### 4.1 AllSet Propagation Setting

For a given node $v \in \mathcal{V}$ and hyperedge $e \in \mathcal{E}$ in a hypergraph $\mathcal{G} = (\mathcal{V}, \mathcal{E})$, let $\boldsymbol{x}_v^{(t)} \in \mathbb{R}^f$ and $\boldsymbol{z}_e^{(t)} \in \mathbb{R}^d$ denote their vector representations at propagation step $t$. We say that a function $f$ is a multiset function if it is permutation invariant w.r.t. each of its arguments in turn. Typically, $\boldsymbol{x}_v^{(0)}$ and $\boldsymbol{z}_e^{(0)}$ are initialized based on the corresponding node and hyperedge original features, if available. The vectors $\boldsymbol{x}_v^{(0)}$ and $\boldsymbol{z}_e^{(0)}$ represent the initial node and hyperedge features, respectively. In this context, the AllSet framework (Chien et al., 2022) consists in the following two-step update rule:

$$z_e^{(t+1)} = f_{\mathcal{V}\to\mathcal{E}}(\{\boldsymbol{x}_u^{(t)}\}_{u:u\in e}; \boldsymbol{z}_e^{(t)}), \tag{5}$$

$$\boldsymbol{x}_v^{(t+1)} = f_{\mathcal{E}\to\mathcal{V}}(\{\boldsymbol{z}_e^{(t+1)}\}_{e\in\mathcal{E}_v}; \boldsymbol{x}_v^{(t)}), \tag{6}$$

where $f_{\mathcal{V}\to\mathcal{E}}$ and $f_{\mathcal{E}\to\mathcal{V}}$ are two permutation invariant functions with respect to their first input. Equations 5 and 6 describe the propagation from nodes to hyperedges and vice versa, respectively. We extend the original AllSet formulation to accommodate UniGCNII (Huang & Yang, 2021) by modifying the node update rule (Eq. 6) as follows

$$\boldsymbol{x}_v^{(t+1)} = f_{\mathcal{E}\to\mathcal{V}}(\{\boldsymbol{z}_e^{(t+1)}\}_{e\in\mathcal{E}_v}; \{\boldsymbol{x}_v^{(k)}\}_{k=0}^t), \tag{7}$$

i.e. allowing residual connections. Again, the only requirement is to be invariant w.r.t. the first input.

**Proposition 4.1.** *UniGCNII (Huang & Yang, 2021) is a special case of AllSet considering 5 and 7.*

In the practical implementation of a model, $f_{\mathcal{V}\to\mathcal{E}}$ and $f_{\mathcal{E}\to\mathcal{V}}$ are parametrized and learnt for each dataset and task; particular choices of these functions give rise to most of the HNN architectures considered in this paper (see Appendix B).

## 4.2 MultiSet Framework

In this Section, we introduce our proposed MultiSet framework, which can be seen as an extension of AllSet where nodes can have multiple co-existing hyperedge–based representations. For a given hyperedge $e \in \mathcal{E}$ in a hypergraph $\mathcal{G} = (\mathcal{V}, \mathcal{E})$, we denote by $\boldsymbol{z}_e^{(t)} \in \mathbb{R}^d$ its vector representation at step $t$. For a node $v \in \mathcal{V}$, MultiSet allows for as many representations of the node as the number of hyperedges it belongs to. We denote by $\boldsymbol{x}_{v,e}^{(t)} \in \mathbb{R}^f$ the vector representation of node $v$ in a hyperedge $e \in \mathcal{E}_v$ at propagation time $t$, and by $\mathbb{X}_v^{(t)} = \{\boldsymbol{x}_{v,e}^{(t)}\}_{e\in\mathcal{E}_v}$ the set of all $d_v$ hidden states of that node in the specified time-step. Accordingly, the hyperedge and node update rules of MultiSet are formulated to accommodate hyperedge–dependent node representations:

$$\boldsymbol{z}_e^{(t+1)} = f_{\mathcal{V}\to\mathcal{E}}(\{\mathbb{X}_u^{(t)}\}_{u:u\in e}; \boldsymbol{z}_e^{(t)}), \tag{8}$$

$$\boldsymbol{x}_{v,e}^{(t+1)} = f_{\mathcal{E}\to\mathcal{V}}(\{\boldsymbol{z}_e^{(t+1)}\}_{e\in\mathcal{E}_v}; \{\mathbb{X}_v^{(k)}\}_{k=0}^t), \tag{9}$$

where $f_{\mathcal{V}\to\mathcal{E}}$ and $f_{\mathcal{E}\to\mathcal{V}}$ are two multiset functions with respect to their first input. After $T$ iterations of message passing, MultiSet also considers a last readout-based step to obtain a unique final representation $\boldsymbol{x}_v^T \in \mathbb{R}^{f'}$ for each node from the set of its hyperedge–based representations:

$$\boldsymbol{x}_v^{(T)} = f_{\mathcal{V}\to\mathcal{V}}(\{\mathbb{X}_v^{(k)}\}_{k=0}^T) \tag{10}$$

where $f_{\mathcal{V}\to\mathcal{V}}$ is also a multiset function.

As we show in the following propositions (proofs can be found in Appendices C.2 C.3 and C.4), both AllSet framework and the more recent EDHNN and WHATsNet architectures (Wang et al., 2023; Choe et al., 2023) can be expressed in terms of the general MultiSet notation.

**Proposition 4.2.** *AllSet 5-6, as well as its extension 5-7, are special cases of MultiSet 8-9-10.*

**Proposition 4.3.** *EDHNN (Wang et al., 2023) is a special case of MultiSet 8-9-10.*

**Proposition 4.4.** *WHATsNet (Choe et al., 2023) is a special case of MultiSet 8-9-10.*

Figure 2 represents the AllSet and MultiSet layouts. Please notice that to avoid clutter notation at step $t$ for the MultiSet layout, we omit populating nodes at step $t$. As shown at the top of the figure, we obtain $x_{v_2, e_1^{t+1}}$ and $x_{v_2, e_2^{t+1}}$, which can subsequently be used to update the corresponding hyperedges and nodes.

## 4.3 Training MultiSet Networks

This section describes a possible realization of a MultiSet layer implementation –which we refer to as MultiSetMixer model.

**Learning MultiSet layers.** Following the mixer-style block designs (Tolstikhin et al., 2021) and standard practice, we use the following MultiSet layer implementation:

$$z_e^{(t+1)} = f_{\mathcal{V} \to \mathcal{E}}(\{x_{u,e}^{(t)}\}_{u:u \in e}; z_e^{(t)}) \tag{11}$$

$$:= \frac{1}{|e|} \sum_{v \in e} x_{u,e}^{(t)} + \text{MLP}\left(\text{LN}\left(\frac{1}{|e|} \sum_{v \in e} x_{u,e}^{(t)}\right)\right),$$

$$x_{v,e}^{(t+1)} = f_{\mathcal{E} \to \mathcal{V}}(z_e^{(t+1)}; x_{v,e}^{(t)}) \tag{12}$$

$$:= x_{v,e}^{(t)} + \text{MLP}\left(\text{LN}(x_{v,e}^{(t)})\right) + z_e^{(t+1)},$$

$$x_v^{(T)} = f_{\mathcal{V} \to \mathcal{V}}(\mathbb{X}_v^{(T)}) := \frac{1}{d_v} \sum_{e \in \mathcal{E}_v} x_{v,e}^{(t)} \tag{13}$$

where MLPs are composed of two fully-connected layers, and LN stands for layer normalization. This architecture, which we call MultiSetMixer, is based on a mixer-based pooling operation for *(i)* updating hyperedges from its node's representations, and *(ii)* generate and update hyperedge-dependent representations of the nodes.

**Proposition 4.5.** *The functions $f_{\mathcal{V} \to \mathcal{E}}$, $f_{\mathcal{E} \to \mathcal{V}}$ and $f_{\mathcal{V} \to \mathcal{V}}$ defined in MultiSetMixer 11-12-13 are permutation invariant. Furthermore, these functions are universal approximators of multiset functions when the size of the input multiset is finite.*

## 4.4 Mini-batching

The motivation for introducing a new strategy to iterate over hypergraph datasets is twofold. First, current HNN pipelines face scalability issues when processing datasets with a large number of nodes and hyperedges. This problem is particularly pronounced in architectures that allow for hyperedge-dependent node representations, as each node must be represented multiple times. This scaling challenge becomes significant as the number of nodes, hyperedges, and hyperedge sizes grow (see Table 8 in the Appendix for relevant statistics). Second, pooling operations over large sets can lead to signal oversquashing, negatively impacting the performance of HNNs that do not account for this issue (see Table 1), as demonstrated by the 20Newsgroups dataset, where hyperedges have a median of 537 nodes per hyperedge (see Table 7 in the Appendix for detailed statistics).

To address these issues, we propose sampling $X$ mini-batches of a certain size $B$ at each iteration in two steps. At *step 1*, we sample $B$ hyperedges from $\mathcal{E}$. The hyperedge sampling over $\mathcal{E}$ can be either uniform or weighted (e.g. by taking into account hyperedge cardinalities). Then in *step 2* $L$ nodes are in turn sampled from each sampled hyperedge $e$, padding the hyperedge with $L - |e|$ special padding tokens if $|e| < L$ –consisting of **0** vectors that can be easily discarded in some computations. Overall, the shape of the obtained mini-batch $X$ is $B \times L$.

*Step 1* is particularly beneficial for hypergraph datasets with a large number of hyperedges, but it can be skipped when the entire network fits into memory. In Sections 5.1 and 5.3, we demonstrate that *step 2* (node mini-batching within a hyperedge) is useful for two key reasons: (i) pooling operations over large sets may lead to signal oversquashing, while pooling over smaller sets helps mitigate this issue, and (ii) node batching introduces an intriguing effect—what we refer to as connectivity-based distribution shift—which offers a novel way to leverage connectivity to rebalance the training distribution. Therefore, it can still be beneficial to use node mini-batching even when the entire hyperedge fits into memory.

When both *step 1* and *step 2* are employed, the memory required during the forward pass is $B \times L \times d$, where $d$ is the hidden dimension size. If only *step 2* is used, the batch size becomes $|\mathcal{E}| \times L \times d$, where $|\mathcal{E}|$ is the total number of hyperedges in the hypernetwork. Finally, if no mini-batching is applied, the batch size is $|\mathcal{E}| \times \max_{e \in \mathcal{E}} |e| \times d$, where $\max_{e \in \mathcal{E}} |e|$ denotes the size of the largest hyperedge. For a detailed empirical analysis of memory utilization across different datasets and sampling schemes, see Section 5.3. For the theoretical analysis of the proposed sampling scheme, refer to Appendix F.

## 5 Experimental Results

The questions that we raised in Section 1 have shaped our research, leading to a new definition of higher-order homophily and unexplored architectural designs that can potentially fit better the properties of hypergraph networks. In subsequent subsections, we set four questions that follow up from these fundamental inquiries and can help contextualize the technical contributions of this paper.

**Datasets and models.** We use the same datasets used in Chien et al. (2022), which includes Cora, Citeseer, Pubmed, ModelNet40, NTU2012, 20Newsgroups, Mushroom, ZOO, CORA-CA, and DBLP-CA. More information about datasets and corresponding statistics are in Appendix D.2. We also utilize the benchmark implementation provided by Chien et al. (2022) to conduct the experiments with several models, including AllDeepSets, AllSetTransformer, EDHNN, UniGCNII, CEGAT, CEGCN, HCHA, HNN, HNHN, HyperGCN, HAN, and HAN mini-batching. Since the design of these architectures is primarily inspired by the GNN literature, we refer to them as lifted architectures. Additionally, we consider vanilla MLP applied to node features and a transformer architecture, as well as MultiSetMixer representing a baseline that employs hyperedge-dependent node representations and a new MLP baseline leveraging Connectivity Batching (MLP CB). We refer to Section 4.3 for more details about all these architectures. All models are optimized using 15 splits with 2 model initializations, resulting in a total of 30 runs; see Appendix D.1 for further details.

### 5.1 How do the lifted models perform in comparison to hypergraph-specific models?

Our first experiment directly targets our fundamental **Q2** by assessing the performance of lifted architectures in comparison with those that leverage unique characteristics of hypergraph networks, such as hyperedge-dependent node representations and hyperedge-based mini-batching.

Figure 3 shows the average rankings –across all models and datasets– of the top-5 best-performing models for different training splits, exhibiting that those splits can impact the relative performance among models. In the main body of this work, we focus our analysis on the 50% split results presented in Table 1,[2] while the corresponding tables for other scenarios are provided in Appendix E.4. Table 1 emphasizes MultiSet-Mixer solid performance, obtaining the highest test accuracy on NTU2012, ModelNet40, and 20Newsgroups datasets. Notably, MultiSetMixer and MLP CB share similar patterns, and both significantly outperform all the other architectures on 20Newsgroups, which we further discuss in Section 5.3.

In fact, the comparable performance among the rest of HNN models on this dataset suggests that existing architectures can not account for the dataset connectivity. According to what we observed in the qualitative homophily analysis performed in Section 3, 20Newsgroup is densely interconnected, making it highly heterophilic as the MP evolves; we argue this presents a challenge for most of current HNNs architectures. In contrast, CORA-CA exhibits a high degree of homophily within its hyperedges and shows the most significant performance gap between HNNs and the baselines. A similar trend is observed for DBLP-CA (see

---

[2]Unless otherwise specified, all tables in the main body of the paper use a 50%/25%/25% split between training and testing. The results are shown as Mean Accuracy Standard Deviation, with the best result highlighted in bold and shaded in grey, and results within one standard deviation are displayed in blue-shaded boxes.

Table 1: Hypergraph model performance benchmarks. Test accuracy in % averaged over 15 splits.

| Model | Cora | Citeseer | Pubmed | CORA-CA | DBLP-CA | Mushroom | NTU2012 | ModelNet40 | 20Newsgroups | ZOO | avg. ranking |
|---|---|---|---|---|---|---|---|---|---|---|---|
| AllDeepSets | 77.11 ± 1.00 | 70.67 ± 1.42 | 89.04 ± 0.45 | 82.23 ± 1.46 | 91.34 ± 0.27 | 99.96 ± 0.05 | 86.49 ± 1.86 | 96.70 ± 0.25 | 81.19 ± 0.49 | 89.10 ± 7.00 | 6.80 |
| AllSetTransformer | 79.54 ± 1.02 | 72.52 ± 0.88 | 88.74 ± 0.51 | 84.43 ± 1.14 | 91.61 ± 0.19 | 99.95 ± 0.05 | 88.22 ± 1.42 | 98.00 ± 0.12 | 81.59 ± 0.59 | 91.03 ± 7.31 | 3.25 |
| UniGCNII | 78.46 ± 1.14 | 73.05 ± 1.48 | 88.07 ± 0.47 | 83.92 ± 1.02 | 91.56 ± 0.18 | 99.89 ± 0.07 | 88.24 ± 1.56 | 97.84 ± 0.16 | 81.16 ± 0.49 | 89.61 ± 8.09 | 4.75 |
| EDHNN | 80.74 ± 1.00 | 73.22 ± 1.14 | 89.12 ± 0.47 | 85.17 ± 1.02 | 91.94 ± 0.23 | 99.94 ± 0.11 | 88.04 ± 1.65 | 97.70 ± 0.19 | 81.64 ± 0.49 | 89.49 ± 6.99 | 2.90 |
| CEGAT | 76.53 ± 1.58 | 71.58 ± 1.11 | 87.11 ± 0.49 | 77.50 ± 1.51 | 88.74 ± 0.31 | 96.81 ± 1.41 | 82.27 ± 1.60 | 92.79 ± 0.44 | NA | 44.62 ± 9.18 | 12.11 |
| CEGCN | 77.03 ± 1.31 | 70.87 ± 1.19 | 87.01 ± 0.62 | 77.55 ± 1.65 | 88.12 ± 0.25 | 94.91 ± 0.44 | 80.90 ± 1.74 | 90.04 ± 0.47 | NA | 49.23 ± 6.81 | 12.56 |
| HCHA | 79.53 ± 1.33 | 72.57 ± 1.06 | 86.97 ± 0.55 | 83.53 ± 1.12 | 91.21 ± 0.28 | 98.94 ± 0.54 | 86.60 ± 1.96 | 94.50 ± 0.33 | 80.75 ± 0.53 | 89.23 ± 6.81 | 7.85 |
| HGNN | 79.53 ± 1.33 | 72.24 ± 1.08 | 86.97 ± 0.55 | 83.45 ± 1.22 | 91.26 ± 0.26 | 98.94 ± 0.54 | 86.71 ± 1.48 | 94.50 ± 0.33 | 80.75 ± 0.52 | 89.23 ± 6.81 | 7.95 |
| HNHN | 77.68 ± 1.08 | 73.47 ± 1.36 | 87.88 ± 0.47 | 78.53 ± 1.15 | 86.73 ± 0.40 | 99.97 ± 0.04 | 88.28 ± 1.50 | 97.84 ± 0.15 | 81.53 ± 0.55 | 89.23 ± 7.85 | 5.55 |
| HyperGCN | 74.78 ± 1.11 | 66.06 ± 1.58 | 82.32 ± 0.62 | 77.48 ± 1.14 | 86.07 ± 3.32 | 69.51 ± 4.98 | 47.65 ± 5.01 | 46.10 ± 7.95 | 80.84 ± 0.49 | 51.54 ± 9.88 | 14.30 |
| HAN | 80.73 ± 1.37 | 73.69 ± 0.95 | 86.34 ± 0.61 | 84.19 ± 0.81 | 91.10 ± 0.20 | 91.33 ± 0.91 | 83.78 ± 1.75 | 93.85 ± 0.33 | 79.67 ± 0.55 | 80.26 ± 6.42 | 8.90 |
| HAN minibatch | 80.24 ± 2.17 | 73.55 ± 1.13 | 85.41 ± 2.32 | 82.04 ± 2.56 | 90.52 ± 0.50 | 93.87 ± 1.04 | 80.62 ± 2.00 | 92.06 ± 0.63 | 79.76 ± 0.56 | 70.39 ± 11.29 | 10.60 |
| MultiSetMixer | 78.06 ± 1.24 | 71.85 ± 1.50 | 87.19 ± 0.53 | 82.74 ± 1.23 | 90.68 ± 0.19 | 99.58 ± 0.16 | 88.90 ± 1.30 | 98.38 ± 0.21 | 88.57 ± 1.96 | 88.08 ± 8.04 | 6.20 |
| MLP CB | 74.06 ± 1.26 | 71.93 ± 1.53 | 85.83 ± 0.51 | 74.39 ± 1.40 | 84.91 ± 0.44 | 99.93 ± 0.08 | 85.43 ± 1.51 | 96.41 ± 0.32 | 86.13 ± 2.82 | 81.61 ± 10.98 | 10.30 |
| MLP | 73.27 ± 1.09 | 72.07 ± 1.65 | 87.13 ± 0.49 | 73.27 ± 1.09 | 84.77 ± 0.41 | 99.91 ± 0.08 | 79.70 ± 1.56 | 95.31 ± 0.28 | 80.93 ± 0.59 | 85.13 ± 6.90 | 11.50 |
| Transformer | 74.15 ± 1.17 | 71.82 ± 1.51 | 87.37 ± 0.49 | 73.61 ± 1.55 | 85.26 ± 0.38 | 99.95 ± 0.08 | 82.88 ± 1.93 | 96.29 ± 0.29 | 81.17 ± 0.54 | 88.72 ± 10.25 | 9.85 |

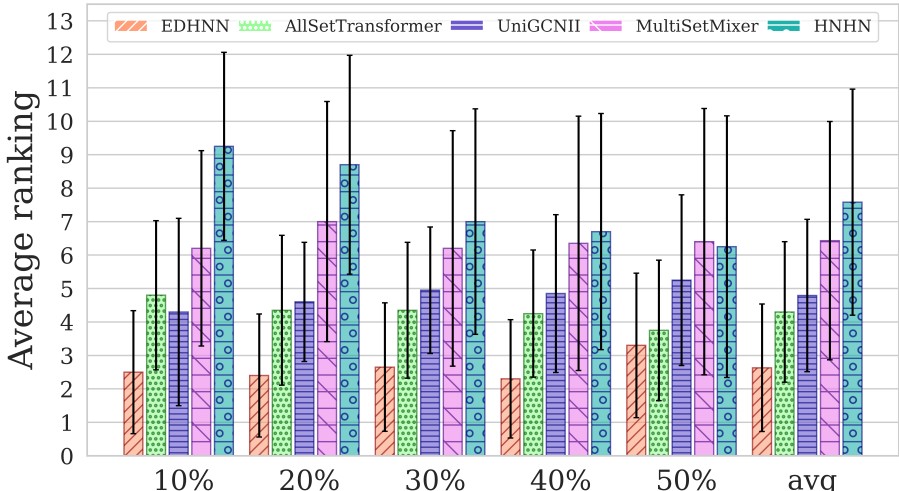

Figure 3: Average rankings at different training percentages, with the overall average performance shown on the right. The error bars indicate the standard deviation of the average rankings across multiple datasets.

node homophily plot in Appendix G). Please refer to Section 5.4 for more experiments on the impact of connectivity.

Finally, we highlight the strong overall performance of the non-inductive baselines (MLP, MLP CB) across most datasets. Notably, HNN architectures significantly outperform them in only 3 out of 10 cases (Cora, CORA-CA, DBLP-CA). This fact showcases that, so far, features are being more representative than connectivity in most considered hypergraph datasets –a relevant insight for **Q3**.

## 5.2    When are HNNs exploiting the connectivity?

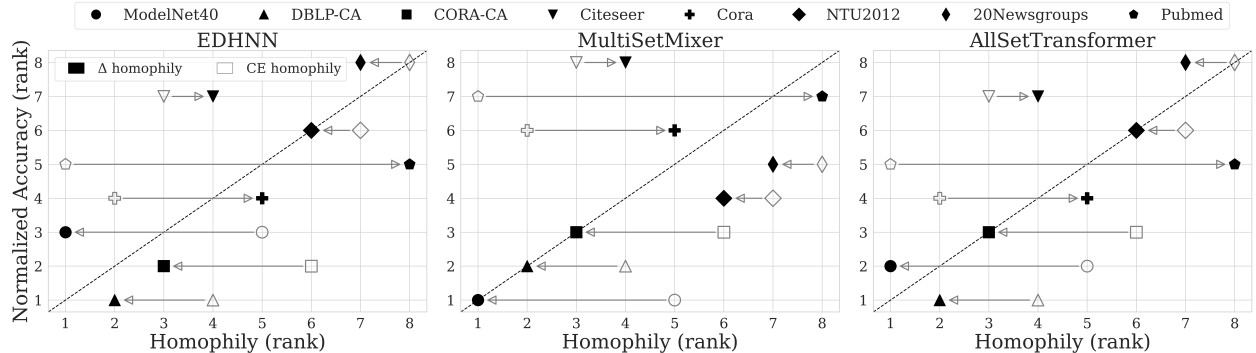

Figure 4: Joint visualization of rank dependencies, showing Norm. Acc. versus $\Delta$ Homophily (Eq. 4 at step $t = 1$ and $\mu = 0.1$) and CE Homophily (Wang et al., 2023). Norm. Acc. (Eq. 14) is assessed for various instances of model $A$ (specified in column titles), with model $B$ being MLP CB. Both axes represent rank values, with lower values indicating better metrics. Arrows denote the rank shift in homophily between CE homophily and $\Delta$ homophily for each dataset.

Motivated by the previous observation of the general results, in this section, we investigate when HNN models actually take advantage of the inductive bias provided by the message-passing scheme. To do so, we first propose a way to capture the influence of the bias in the downstream task performance –in an attempt to decouple it from the impact of just the dataset features–, and then investigate if the datasets' homophily scores are able to account for the resulting observations. We argue this study provides a valuable contribution to questions **Q2** and **Q3**.

In order to quantitatively assess the impact of inductive bias, we compare the results of HNNs –i.e., model $A$–, with those of another architecture that does not leverage the connectivity –i.e. a non-inductive baseline, model $B$. Specifically, we measure the difference between the accuracy of model $A$ ($acc_A$) and $B$ ($acc_B$) (Table 2 shows the computed differences considering MLP and MLP CB as baselines). The real-world datasets employed in this study span diverse domains and, as depicted in Table 1, this implies considerable variations in performance values across datasets. In order to mitigate such variability, we introduce the following normalized accuracy relative to $acc_B$:

$$\text{Norm. Acc.} = (acc_A - acc_B)/(100 - acc_B). \tag{14}$$

Table 2: Difference in Accuracy between Model A and Model B; they represent, respectively, models with and without inductive bias.

| Model A | Model B | Cora | Citeseer | Pubmed | CORA-CA | DBLP-CA | Mushroom | NTU2012 | ModelNet40 | 20Newsgroups | ZOO |
|---|---|---|---|---|---|---|---|---|---|---|---|
| AllDeepSets | MLP | 3.84 | -1.40 | 1.91 | 8.96 | 6.57 | 0.05 | 6.79 | 1.39 | 0.26 | 3.97 |
| | MLP CB | 3.05 | -1.26 | 3.21 | 7.84 | 6.43 | 3.13 | 1.06 | 0.29 | -4.94 | 7.49 |
| AllSetTransformer | MLP | 6.27 | 0.45 | 1.61 | 11.16 | 6.84 | 0.04 | 8.52 | 2.69 | 0.66 | 5.90 |
| | MLP CB | 5.48 | 0.59 | 2.91 | 10.04 | 6.70 | 3.12 | 2.79 | 1.59 | -4.54 | 9.42 |
| EDHNN | MLP | 7.47 | 1.15 | 1.99 | 11.90 | 7.17 | 0.03 | 8.34 | 2.39 | 0.71 | 4.36 |
| | MLP CB | 6.68 | 1.29 | 3.29 | 10.78 | 7.03 | 0.01 | 2.61 | 1.29 | -4.49 | 7.88 |
| MultiSetMixer | MLP | 4.79 | -0.22 | 0.06 | 9.47 | 5.91 | -0.33 | 9.20 | 3.07 | 7.64 | 2.95 |
| | MLP CB | 4.00 | -0.08 | 1.36 | 8.35 | 5.77 | -0.35 | 3.47 | 1.97 | 2.44 | 6.47 |

Next, we are interested in assessing whether the homophily of the dataset can shed some light on the resulting normalized accuracy measurements. To that end, we consider two different homophily measures: on the one hand, our proposed $\Delta$ homophily between the two first steps of the MP (Eq 4 with $t = 1$ and $\mu = 0.1$). On the other hand, Clique Expansion (CE) homophily, calculated over the clique expansion of the hypergraph following the approach of Wang et al. (2023). Figure 4 illustrates the rank dependency of normalized accuracy against these two homophily measures, with MLP CB serving as the strongest non-inductive baseline (as indicated by Table 1). Additionally, we show the ideal correlation –performance directly proportional to homophily– with the dashed diagonal, as well as the rank shift in homophily between the two homophily measures through the arrows. The difference between EDHNN, MultiSetMixer, and AllSetTransformer lies in the way message passing propagates information (see Section 4.2). Mushroom and Zoo datasets were excluded due to Mushroom's discriminatory node features and Zoo's small hypernetwork size.

The comparison between CE homophily and $\Delta$ homophily reveals a notable trend, with $\Delta$ homophily consistently aligning closer, on average, to the middle dashed line –indicating a higher positive correlation between performance and $\Delta$ homophily level in comparison to CE homophily. Remarkably, across all architectures, the highest normalized accuracy is consistently distributed across ModelNet40, DBLP-CA, and CORA-CA, with $\Delta$ homophily ranking them as the top three accordingly. A striking shift in rankings is observed for the Pubmed dataset, transitioning from the most homophilic under the CE homophily measure to the least homophilic under $\Delta$ homophily. We associate this to the high percentage of isolated nodes (80.52%, see Table 8): while CE homophily scores are largely influenced by them, our proposed measure ignores self-connections. Additionally, the 20Newsgroup dataset occupies the last positions in both homophily ranks, aligning with our quantitative analysis findings.

In summary, we show the applicability of $\Delta$ homophily by showing that it exhibits a positive correlation with respect to the ability of exploiting the connectivity by HNN architectures, significantly stronger than the CE homophily that is commonly used nowadays. Our findings underscore the crucial role of accurately expressing homophily in hypergraph networks, entangling with the complexity in capturing higher-order dependencies. Additionally, please see Appendix E.3 for the analysis on how parameter $\mu$ impacts $\Delta$ homophily.

## 5.3 What is the impact of the mini-batch sampling?

Next, we examine the role of our proposed mini-batching sampling in explaining the general results shown in Table 1, and investigate how it influences other models' performance. These experiments provide valu-

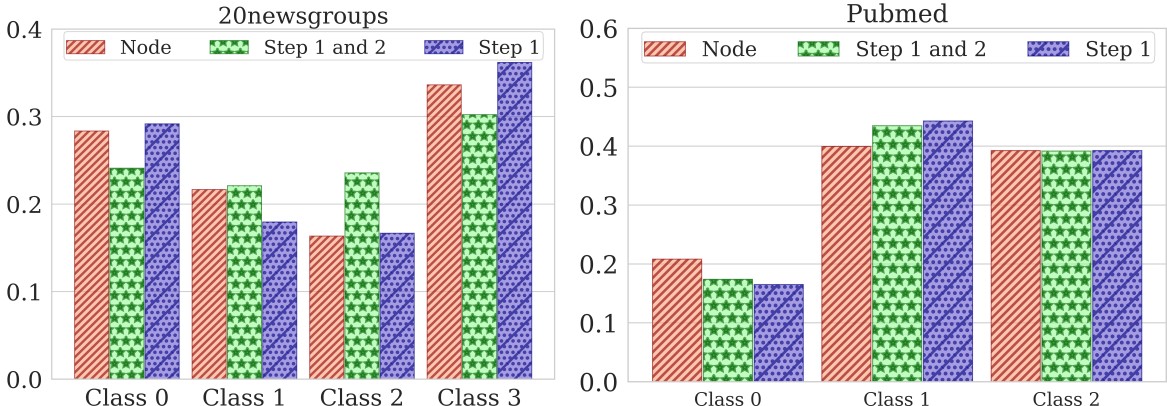

Figure 5: Class distribution shift induced by mini-batching: '*Node*' represents the original node class distribution, '*Step 1 and 2*' the resulting one after sampling both hyperedges and nodes, and '*Step 1*' when only sampling hyperedges.

able insights on **Q2**. Please refer to Section 5.5 for a deeper analysis of how hyperedge-dependent node representations impact HNNs.

Table 3: Mini-batching experiment. Test accuracy in % averaged over 15 splits.

| Model | Cora | Citeseer | Pubmed | CORA-CA | DBLP-CA | Mushroom | NTU2012 | ModelNet40 | 20Newsgroups | ZOO | avg. ranking |
|---|---|---|---|---|---|---|---|---|---|---|---|
| AllSetTransformer (batched) | 74.34 ± 1.08 | 69.67 ± 1.46 | **87.75 ± 0.30** | 75.75 ± 1.46 | 86.06 ± 0.22 | 99.91 ± 0.05 | 87.55 ± 0.86 | 96.42 ± 0.17 | 81.37 ± 0.28 | **93.20 ± 5.38** | 2.70 |
| EDHNN | 77.88 ± 0.69 | 69.51 ± 0.87 | 86.82 ± 0.33 | **83.12 ± 0.89** | 90.45 ± 0.28 | **99.95 ± 0.04** | 87.64 ± 0.99 | 97.55 ± 0.17 | 81.23 ± 0.31 | 90.00 ± 4.43 | 2.40 |
| MultiSetMixer | 78.06 ± 1.24 | 71.85 ± 1.50 | 87.19 ± 0.53 | 82.74 ± 1.23 | **90.68 ± 0.19** | 99.58 ± 0.16 | **88.90 ± 1.30** | **98.38 ± 0.21** | **88.57 ± 1.96** | 88.08 ± 8.04 | **1.80** |
| HAN minibatch | **80.24 ± 2.17** | **73.55 ± 1.13** | 85.41 ± 2.32 | 82.04 ± 2.56 | 90.52 ± 0.50 | 93.87 ± 1.04 | 80.62 ± 2.00 | 92.06 ± 0.63 | 79.76 ± 0.56 | 70.39 ± 11.29 | 3.10 |

**Class distribution analysis.** To evaluate and motivate the potential of the proposed mini-batching sampling, we investigate the reason behind both *(i)* the superior performance of MultiSetMixer and MLP CB on 20NewsGroup, NTU2012, ModelNet40, and *(ii)* their poor performance on Pubmed. Framing mini-batching from the connectivity perspective presents a challenge that conceals significant potential for improvement (Teney et al., 2023). It is important to note that connectivity, by definition, describes relationships among the nodes, implying that some parts of the dataset might interconnect more densely, creating some sort of hubs within the network. Thus, mini-batching might introduce unexpected skew in training distribution. In particular, Figure 5, depicts the original class distribution of the dataset, and compares it to the skewed distributions resulting from employing the corresponding steps of mini-batching (see Section 4.3). Note that, when sampling both hyperedges and nodes ('*Step 1 and 2*'), dominant classes 0 and 3 undergo undersampled, contributing to a more balanced distribution in the case of 20Newsgroup. Conversely, for Pubmed, class 2 is undersampled, while the predominant 1st class experiences oversampling, further skewing the distribution in this dataset. This observation leads to the hypothesis that, in some cases, the sampling procedure produces a natural shift that rebalances the class distributions, which in turn helps to improve the performance.

**Application to other models.** Furthermore, we explore the proposed mini-batch sampling procedure with the AllSetTransformer and EDHNN models by just sampling hyperedges without additional hyperparameter optimization. From Table 3, we can observe a drop in performance for most of the datasets both for AllSetTransformer and for EDHNN, but overall they in turn outperform the HAN (mini-batching) model. This suggests the substantial potential of the proposed sampling procedure.

**Scalability.** To evaluate the scalability of the proposed mini-batching approach, we compare memory utilization across the mid-sized datasets used in this study, as well as the large-scale Amazon dataset, which consists of 100k hyperedges and 547k nodes. This comparison highlights the efficiency of our method when applied to larger datasets. On Amazon, MultiSetMixer achieves a substantial improvement in memory efficiency, reducing utilization by an order of magnitude compared to AllSetTransformer (from 19020MiB to 1232MiB) while achieving comparable performance (92.22 ± 0.84 vs 93.22 ± 0.089). In contrast, the batched

version of AllSetTransformer performs worse on the same dataset, achieving $82.634 \pm 0.029$, while consuming around eight times more memory compared to MultiSetMixer.

Table 4: Memory usage comparison across datasets.

| Dataset | Pubmed | DBLP CA | NTU2012 | ModelNet40 | 20newsW100 | Amazon |
|---|---|---|---|---|---|---|
| AllSetTransformer | 2536MiB | 2306MiB | 1206MiB | 2426MiB | 2352MiB | 19020MiB |
| AllSetTransformer (batched) | 2652MiB | 1940MiB | 1138MiB | 2038MiB | 4224MiB | 8022MiB |
| MultiSetMixer (no batching) | 12480MiB | 17850MiB | 1576MiB | 2728MiB | 3039MiB | OOM |
| MultiSetMixer (batched) | 1290MiB | 2364MiB | 1174MiB | 1280MiB | 1112MiB | 1232MiB |

## 5.4 How do connectivity changes affect performance?

In this section, we extensively explore the structure of datasets and assess model performance by manipulating the original connectivity of the datasets. The extent to which the performance of the models is affected by changes in connectivity provides valuable information both on the properties of the datasets and on the considered architectures.

We design two different experimental approaches, aiming to systematically modify the original connectivity of datasets. The first experiment tests the performance when some hyperedges are removed following different *drop connectivity* strategies. The second one examines the models performance by introducing two preprocessing strategies on the hypergraph connectivity. Our findings below shed some light on the fundamental questions **Q1**, **Q2** and **Q3**.

### 5.4.1 Reducing connectivity

This experiment aims to investigate the significance of connectivity information in datasets and the extent to which it influences the performance of the models. We divide this experiment into two parts: (i) drop connectivity and (ii) connectivity rewiring. In the first part of the experiment, we employ three strategies to introduce variations in the initial dataset's connectivity. The first two strategies involve ordering hyperedges based on their lengths in **ascending order**. In the first approach, referred to as *trimming*, we remove the initial $x\%$ of ordered hyperedges, for a certain fixed fraction $x$. The second approach, referred to as *retention*, involves keeping the first $x\%$ of hyperedges and discarding the remaining $100 - x\%$. The last strategy instead involves randomly dropping $x\%$ of hyperedges from the dataset, and it is referred to as *random drop*.

**Results.** The results are shown in Table 5, and they indicate that connectivity minimally impacts CEGCN and AllSetTransformer for the Citeseer and Pubmed datasets. On the other hand, MultiSetMixer performs better at the *trimming 25%* setting, although the achieved performance is on par with MLP, as reported in Table 1. This indicates that the proposed model is negatively affected by the distribution shift. Conversely, for the Pubmed dataset, we observe that MultiSetMixer's performance improves, likely due to the reduced impact of the distribution shift in this case. Notably, CEGCN shows improvements on 7 out of 9 datasets, with the most significant gains seen in the ZOO dataset, where performance nearly doubles. Another interesting pattern is observed in the Cora, CORA-CA, and DBLP-CA datasets, where retaining only 25% of the highest relationships (*retention 25%*) consistently results in better performance compared to retaining 50% or 75%. This outcome is surprising, as at the 25% level, only a small fraction of the higher-order relationships is preserved. In contrast, the *trimming* strategy shows the opposite trend, with performance declining as more higher-order relationships are removed. This phenomenon remains consistent across all models. Finally, when hyperedges are removed randomly, the performance consistently degrades as the percentage of removed hyperedges increases.

Table 5: Drop connectivity. Test accuracy in % averaged over 15 splits.

| Model | Type | Cora | Citeseer | Pubmed | CORA-CA | DBLP-CA | Mushroom | NTU2012 | ModelNet40 | 20Newsgroups | ZOO | avg. ranking |
|---|---|---|---|---|---|---|---|---|---|---|---|---|
| AllDeepSets | Original | 77.11 ± 1.00 | 70.67 ± 1.42 | 89.04 ± 0.45 | 82.23 ± 1.46 | 91.34 ± 0.27 | 99.96 ± 0.05 | 86.49 ± 1.86 | 96.70 ± 0.25 | 81.19 ± 0.49 | 89.10 ± 7.00 | 2.85 |
| | Random 25% | 76.65 ± 1.03 | 70.83 ± 1.70 | 88.87 ± 0.47 | 80.39 ± 1.51 | 90.36 ± 0.28 | 99.91 ± 0.09 | 85.49 ± 1.74 | 96.85 ± 0.26 | 81.15 ± 0.52 | 88.72 ± 5.97 | 4.05 |
| | Random 50% | 75.66 ± 1.18 | 70.70 ± 1.77 | 88.86 ± 0.41 | 77.97 ± 1.18 | 89.36 ± 0.25 | 99.93 ± 0.05 | 84.42 ± 1.75 | 96.88 ± 0.27 | 81.21 ± 0.39 | 84.49 ± 6.66 | 4.90 |
| | Random 75% | 74.96 ± 1.08 | 70.46 ± 1.66 | 88.76 ± 0.45 | 76.09 ± 1.27 | 87.68 ± 0.30 | 99.53 ± 0.13 | 81.70 ± 1.57 | 96.84 ± 0.27 | 81.31 ± 0.50 | 81.15 ± 6.88 | 7.50 |
| | Retention 25% | 76.87 ± 0.98 | 70.96 ± 1.82 | 88.94 ± 0.48 | 81.63 ± 1.26 | 90.93 ± 0.18 | 99.83 ± 0.09 | 85.13 ± 2.05 | 96.85 ± 0.26 | 81.09 ± 0.46 | 86.67 ± 7.26 | 3.65 |
| | Retention 50% | 75.80 ± 1.06 | 70.44 ± 1.63 | 88.84 ± 0.40 | 80.50 ± 1.38 | 90.55 ± 0.21 | 99.91 ± 0.09 | 82.25 ± 2.21 | 96.42 ± 0.23 | 81.04 ± 0.45 | 69.61 ± 9.28 | 6.75 |
| | Retention 75% | 75.52 ± 1.49 | 70.36 ± 1.71 | 88.78 ± 0.47 | 79.50 ± 1.09 | 89.48 ± 0.21 | 99.97 ± 0.04 | 78.85 ± 1.93 | 96.44 ± 0.27 | 81.19 ± 0.45 | 74.49 ± 9.97 | 6.85 |
| | Trimming 25% | 74.12 ± 1.30 | 70.95 ± 1.92 | 88.77 ± 0.45 | 74.87 ± 1.32 | 86.39 ± 0.31 | 99.85 ± 0.15 | 77.47 ± 1.67 | 96.18 ± 0.28 | 81.61 ± 0.47 | 89.23 ± 8.11 | 7.10 |
| | Trimming 50% | 75.24 ± 0.99 | 70.42 ± 1.62 | 88.87 ± 0.46 | 75.89 ± 1.53 | 87.14 ± 0.31 | 99.93 ± 0.06 | 82.76 ± 1.61 | 96.75 ± 0.23 | 81.47 ± 0.48 | 86.28 ± 8.73 | 6.15 |
| | Trimming 75% | 76.03 ± 1.39 | 70.86 ± 1.48 | 88.83 ± 0.48 | 77.50 ± 1.52 | 88.64 ± 0.27 | 99.93 ± 0.09 | 84.74 ± 1.81 | 96.82 ± 0.20 | 81.20 ± 0.48 | 86.03 ± 8.48 | 5.20 |
| AllSetTransformer | Original | 79.54 ± 1.02 | 72.52 ± 0.88 | 88.74 ± 0.51 | 84.43 ± 1.14 | 91.61 ± 0.19 | 99.95 ± 0.05 | 88.22 ± 1.42 | 97.20 ± 0.20 | 81.59 ± 0.59 | 91.03 ± 7.31 | 1.95 |
| | Random 25% | 79.11 ± 0.99 | 72.75 ± 1.14 | 88.67 ± 0.47 | 82.36 ± 1.38 | 90.61 ± 0.29 | 99.94 ± 0.09 | 87.50 ± 1.36 | 97.98 ± 0.17 | 81.70 ± 0.52 | 89.87 ± 7.66 | 3.10 |
| | Random 50% | 77.77 ± 1.34 | 72.21 ± 1.25 | 88.50 ± 0.45 | 79.73 ± 1.58 | 89.46 ± 0.27 | 99.96 ± 0.04 | 87.34 ± 1.55 | 97.83 ± 0.17 | 81.55 ± 0.66 | 89.49 ± 6.30 | 5.85 |
| | Random 75% | 76.92 ± 1.20 | 72.40 ± 1.22 | 88.54 ± 0.47 | 77.88 ± 1.74 | 87.73 ± 0.32 | 99.76 ± 0.15 | 86.31 ± 1.34 | 97.52 ± 0.20 | 81.46 ± 0.62 | 87.69 ± 6.09 | 8.10 |
| | Retention 25% | 79.19 ± 1.11 | 72.49 ± 0.86 | 88.73 ± 0.40 | 83.58 ± 1.30 | 91.18 ± 0.17 | 99.93 ± 0.09 | 87.21 ± 1.58 | 97.82 ± 0.17 | 81.63 ± 0.48 | 86.92 ± 7.18 | 4.10 |
| | Retention 50% | 78.16 ± 0.98 | 72.55 ± 1.13 | 88.70 ± 0.37 | 82.90 ± 1.15 | 90.80 ± 0.22 | 99.89 ± 0.18 | 86.67 ± 1.64 | 97.36 ± 0.21 | 81.61 ± 0.49 | 88.08 ± 7.51 | 5.20 |
| | Retention 75% | 77.38 ± 1.35 | 72.43 ± 0.98 | 88.71 ± 0.39 | 81.07 ± 1.20 | 89.83 ± 0.25 | 99.97 ± 0.04 | 85.58 ± 1.70 | 97.27 ± 0.22 | 81.58 ± 0.48 | 88.97 ± 6.91 | 5.80 |
| | Trimming 25% | 75.83 ± 1.31 | 72.39 ± 1.50 | 88.40 ± 0.45 | 76.51 ± 1.35 | 86.38 ± 0.32 | 99.84 ± 0.13 | 86.88 ± 1.66 | 97.10 ± 0.24 | 81.55 ± 0.55 | 93.08 ± 7.79 | 8.05 |
| | Trimming 50% | 77.37 ± 1.17 | 72.32 ± 1.30 | 88.49 ± 0.40 | 77.41 ± 1.73 | 87.03 ± 0.27 | 99.91 ± 0.12 | 86.86 ± 1.53 | 97.86 ± 0.21 | 81.45 ± 0.50 | 89.74 ± 8.53 | 7.50 |
| | Trimming 75% | 78.15 ± 1.11 | 72.67 ± 1.00 | 88.48 ± 0.39 | 78.91 ± 1.54 | 88.55 ± 0.26 | 99.92 ± 0.09 | 87.68 ± 1.56 | 97.90 ± 0.23 | 81.41 ± 0.61 | 91.03 ± 7.17 | 5.35 |
| UniGCNII | Original | 78.46 ± 1.14 | 73.05 ± 1.48 | 88.07 ± 0.47 | 83.92 ± 1.02 | 91.56 ± 0.18 | 99.89 ± 0.07 | 88.24 ± 1.56 | 97.84 ± 0.16 | 81.16 ± 0.49 | 89.61 ± 8.09 | 2.25 |
| | Random 25% | 78.31 ± 1.29 | 72.91 ± 1.24 | 88.09 ± 0.47 | 82.42 ± 1.05 | 90.86 ± 0.22 | 99.85 ± 0.10 | 87.82 ± 1.46 | 97.87 ± 0.19 | 81.08 ± 0.52 | 89.10 ± 7.76 | 3.55 |
| | Random 50% | 77.36 ± 1.34 | 72.54 ± 1.40 | 87.94 ± 0.52 | 80.17 ± 1.16 | 89.96 ± 0.24 | 99.85 ± 0.12 | 87.42 ± 1.44 | 97.77 ± 0.15 | 81.06 ± 0.55 | 87.31 ± 8.21 | 6.10 |
| | Random 75% | 76.70 ± 1.35 | 72.23 ± 1.81 | 87.91 ± 0.50 | 77.75 ± 1.31 | 88.54 ± 0.25 | 99.87 ± 0.09 | 87.56 ± 1.56 | 97.49 ± 0.18 | 81.06 ± 0.54 | 87.05 ± 6.50 | 7.30 |
| | Retention 25% | 78.80 ± 0.92 | 72.67 ± 1.24 | 88.10 ± 0.51 | 83.68 ± 0.96 | 91.37 ± 0.20 | 99.87 ± 0.06 | 87.56 ± 1.43 | 97.76 ± 0.16 | 81.01 ± 0.49 | 87.18 ± 7.58 | 4.05 |
| | Retention 50% | 77.18 ± 1.32 | 72.51 ± 1.54 | 88.02 ± 0.47 | 82.81 ± 1.32 | 90.99 ± 0.17 | 99.83 ± 0.08 | 87.16 ± 1.33 | 97.29 ± 0.17 | 80.82 ± 0.49 | 86.15 ± 8.49 | 7.15 |
| | Retention 75% | 76.63 ± 1.23 | 72.64 ± 1.15 | 88.07 ± 0.52 | 81.33 ± 1.27 | 90.17 ± 0.20 | 99.83 ± 0.14 | 86.71 ± 1.33 | 97.04 ± 0.16 | 80.87 ± 0.45 | 87.44 ± 7.49 | 7.20 |
| | Trimming 25% | 75.34 ± 1.26 | 72.68 ± 1.57 | 87.81 ± 0.47 | 76.18 ± 1.19 | 87.42 ± 0.30 | 99.87 ± 0.10 | 87.43 ± 1.53 | 97.00 ± 0.17 | 81.50 ± 0.47 | 92.95 ± 8.15 | 6.70 |
| | Trimming 50% | 76.75 ± 1.10 | 72.30 ± 1.64 | 87.87 ± 0.48 | 77.19 ± 1.42 | 88.00 ± 0.27 | 99.90 ± 0.10 | 87.47 ± 1.56 | 97.71 ± 0.18 | 81.22 ± 0.54 | 90.26 ± 7.40 | 6.15 |
| | Trimming 75% | 77.27 ± 1.08 | 72.69 ± 1.31 | 87.93 ± 0.52 | 78.68 ± 0.96 | 89.26 ± 0.28 | 99.84 ± 0.10 | 87.93 ± 1.69 | 97.86 ± 0.16 | 81.22 ± 0.45 | 90.64 ± 6.90 | 4.55 |
| EDHNN | Original | 80.74 ± 1.00 | 73.22 ± 1.14 | 89.12 ± 0.47 | 85.17 ± 1.02 | 91.94 ± 0.23 | 99.94 ± 0.11 | 88.04 ± 1.65 | 97.70 ± 0.19 | 81.64 ± 0.49 | 89.49 ± 6.99 | 1.30 |
| | Random 25% | 79.88 ± 1.17 | 72.03 ± 1.59 | 89.02 ± 0.33 | 80.54 ± 1.37 | 89.97 ± 0.28 | 99.64 ± 0.16 | 85.37 ± 1.51 | 97.19 ± 0.27 | 78.70 ± 0.49 | 89.23 ± 7.03 | 3.75 |
| | Random 50% | 78.45 ± 1.39 | 72.21 ± 1.60 | 88.81 ± 0.37 | 78.35 ± 1.52 | 88.77 ± 0.25 | 99.82 ± 0.16 | 84.19 ± 1.67 | 97.02 ± 0.24 | 74.37 ± 0.59 | 87.69 ± 6.18 | 6.00 |
| | Random 75% | 77.55 ± 1.60 | 72.33 ± 1.61 | 88.94 ± 0.35 | 77.29 ± 1.29 | 87.41 ± 0.24 | 99.45 ± 0.20 | 82.72 ± 1.43 | 96.44 ± 0.22 | 69.37 ± 0.61 | 80.13 ± 6.78 | 7.85 |
| | Retention 25% | 79.79 ± 1.31 | 71.97 ± 1.39 | 89.09 ± 0.44 | 81.57 ± 1.34 | 90.37 ± 0.22 | 99.93 ± 0.09 | 84.49 ± 1.51 | 97.03 ± 0.22 | 78.43 ± 0.58 | 86.28 ± 5.95 | 4.00 |
| | Retention 50% | 78.71 ± 1.44 | 72.26 ± 1.44 | 88.97 ± 0.36 | 80.26 ± 1.28 | 89.91 ± 0.27 | 99.97 ± 0.06 | 82.12 ± 1.91 | 96.40 ± 0.28 | 73.97 ± 0.53 | 69.10 ± 7.86 | 5.90 |
| | Retention 75% | 78.37 ± 1.45 | 72.60 ± 1.38 | 88.94 ± 0.41 | 78.86 ± 1.14 | 88.82 ± 0.18 | 99.99 ± 0.03 | 81.01 ± 1.69 | 96.18 ± 0.24 | 67.38 ± 0.76 | 73.97 ± 6.44 | 6.55 |
| | Trimming 25% | 77.03 ± 1.37 | 72.71 ± 1.45 | 88.63 ± 0.52 | 77.34 ± 1.46 | 87.00 ± 0.25 | 98.75 ± 0.46 | 82.00 ± 1.66 | 95.87 ± 0.30 | 78.58 ± 0.54 | 86.15 ± 9.07 | 8.00 |
| | Trimming 50% | 78.42 ± 1.33 | 72.49 ± 1.30 | 88.80 ± 0.38 | 78.10 ± 1.48 | 87.48 ± 0.26 | 99.74 ± 0.14 | 82.92 ± 1.35 | 96.83 ± 0.25 | 77.78 ± 0.62 | 83.59 ± 10.00 | 6.80 |
| | Trimming 75% | 79.10 ± 1.30 | 72.18 ± 1.34 | 88.82 ± 0.48 | 78.68 ± 1.32 | 88.44 ± 0.28 | 99.86 ± 0.11 | 85.04 ± 1.50 | 96.96 ± 0.28 | 78.70 ± 0.64 | 89.74 ± 7.78 | 4.85 |
| CEGAT | Original | 76.53 ± 1.58 | 71.58 ± 1.11 | 87.11 ± 0.49 | 77.50 ± 1.51 | 88.74 ± 0.31 | 96.81 ± 1.41 | 82.27 ± 1.60 | 92.79 ± 0.44 | NA | 44.62 ± 9.18 | 5.17 |
| | Random 25% | 75.88 ± 1.53 | 71.81 ± 1.05 | 87.03 ± 0.47 | 78.00 ± 1.68 | 87.58 ± 0.35 | 94.56 ± 2.09 | 82.03 ± 1.47 | 93.14 ± 0.34 | NA | 46.03 ± 9.01 | 5.94 |
| | Random 50% | 75.34 ± 1.52 | 71.86 ± 1.22 | 86.91 ± 0.48 | 76.92 ± 1.00 | 86.81 ± 0.32 | 95.14 ± 2.00 | 81.73 ± 1.44 | 93.62 ± 0.35 | NA | 47.69 ± 8.78 | 6.89 |
| | Random 75% | 75.26 ± 1.45 | 72.17 ± 1.59 | 87.02 ± 0.47 | 76.37 ± 1.26 | 85.79 ± 0.35 | 96.90 ± 1.40 | 82.66 ± 1.39 | 94.54 ± 0.43 | NA | 59.87 ± 8.55 | 5.22 |
| | Retention 25% | 75.96 ± 1.16 | 71.39 ± 1.33 | 87.13 ± 0.50 | 77.35 ± 1.52 | 88.48 ± 0.30 | 96.65 ± 1.49 | 80.39 ± 1.53 | 93.20 ± 0.45 | NA | 45.26 ± 9.40 | 6.39 |
| | Retention 50% | 75.36 ± 1.30 | 71.56 ± 1.27 | 87.16 ± 0.53 | 77.35 ± 1.52 | 88.14 ± 0.31 | 96.73 ± 1.59 | 80.56 ± 2.16 | 93.86 ± 0.41 | NA | 45.38 ± 9.97 | 6.00 |
| | Retention 75% | 75.02 ± 1.64 | 72.06 ± 1.41 | 87.22 ± 0.48 | 77.20 ± 1.47 | 87.54 ± 0.28 | 97.49 ± 0.89 | 81.82 ± 1.23 | 94.94 ± 0.29 | NA | 45.38 ± 9.22 | 5.06 |
| | Trimming 25% | 75.40 ± 1.45 | 72.67 ± 1.76 | 87.68 ± 0.52 | 76.14 ± 1.10 | 85.32 ± 0.42 | 99.72 ± 0.10 | 84.94 ± 1.57 | 94.42 ± 0.33 | NA | 89.23 ± 7.38 | 3.67 |
| | Trimming 50% | 75.90 ± 1.48 | 72.15 ± 1.62 | 87.41 ± 0.51 | 76.09 ± 1.65 | 85.69 ± 0.40 | 99.80 ± 0.12 | 82.04 ± 1.32 | 93.22 ± 0.38 | NA | 67.05 ± 7.87 | 4.44 |
| | Trimming 75% | 76.19 ± 1.68 | 71.82 ± 1.32 | 87.03 ± 0.56 | 76.04 ± 0.95 | 86.18 ± 0.38 | 99.31 ± 0.24 | 81.74 ± 1.48 | 92.79 ± 0.33 | NA | 53.33 ± 6.60 | 6.22 |
| CEGCN | Original | 77.03 ± 1.31 | 70.87 ± 1.19 | 87.01 ± 0.62 | 77.55 ± 1.65 | 88.12 ± 0.25 | 94.91 ± 0.44 | 80.90 ± 1.74 | 90.04 ± 0.47 | NA | 49.23 ± 6.81 | 4.61 |
| | Random 25% | 76.08 ± 1.55 | 71.35 ± 1.44 | 86.89 ± 0.59 | 76.51 ± 1.53 | 87.01 ± 0.39 | 93.11 ± 0.46 | 80.68 ± 1.86 | 90.36 ± 0.46 | NA | 49.74 ± 6.22 | 6.22 |
| | Random 50% | 75.55 ± 1.63 | 71.42 ± 1.60 | 86.70 ± 0.48 | 75.27 ± 1.22 | 86.24 ± 0.35 | 93.28 ± 0.61 | 80.63 ± 1.78 | 90.69 ± 0.54 | NA | 56.92 ± 7.24 | 6.33 |
| | Random 75% | 75.34 ± 1.62 | 71.73 ± 1.90 | 86.97 ± 0.51 | 74.53 ± 1.56 | 85.36 ± 0.26 | 93.01 ± 0.45 | 80.56 ± 1.76 | 91.91 ± 0.53 | NA | 63.20 ± 5.59 | 6.33 |
| | Retention 25% | 76.12 ± 1.58 | 70.87 ± 1.42 | 86.94 ± 0.56 | 76.98 ± 1.53 | 87.90 ± 0.29 | 94.94 ± 0.48 | 79.20 ± 1.42 | 90.59 ± 0.59 | NA | 49.87 ± 7.59 | 5.50 |
| | Retention 50% | 75.43 ± 1.28 | 70.83 ± 1.52 | 86.95 ± 0.54 | 76.87 ± 1.49 | 87.58 ± 0.28 | 94.97 ± 0.40 | 78.53 ± 1.90 | 90.09 ± 0.56 | NA | 45.77 ± 6.88 | 6.89 |
| | Retention 75% | 75.53 ± 1.25 | 71.72 ± 1.42 | 87.11 ± 0.53 | 76.36 ± 1.42 | 87.03 ± 0.28 | 94.74 ± 0.39 | 79.82 ± 1.41 | 92.29 ± 0.46 | NA | 40.38 ± 5.42 | 5.44 |
| | Trimming 25% | 75.58 ± 1.56 | 72.26 ± 1.52 | 86.36 ± 0.51 | 74.84 ± 1.31 | 84.97 ± 0.31 | 99.60 ± 0.11 | 83.10 ± 1.69 | 91.85 ± 0.42 | NA | 87.69 ± 7.31 | 3.44 |
| | Trimming 50% | 76.57 ± 1.47 | 71.81 ± 1.44 | 87.07 ± 0.55 | 74.66 ± 1.68 | 85.24 ± 0.33 | 99.54 ± 0.18 | 80.72 ± 1.64 | 90.64 ± 0.54 | NA | 71.28 ± 6.60 | 4.00 |
| | Trimming 75% | 76.53 ± 1.50 | 71.45 ± 1.45 | 86.75 ± 0.54 | 74.56 ± 1.32 | 85.56 ± 0.33 | 99.14 ± 0.23 | 80.38 ± 1.91 | 90.06 ± 0.37 | NA | 58.46 ± 7.17 | 6.22 |
| HCHA | Original | 79.53 ± 1.33 | 72.57 ± 1.06 | 86.97 ± 0.55 | 83.53 ± 1.12 | 91.21 ± 0.28 | 98.94 ± 0.54 | 86.60 ± 1.96 | 94.50 ± 0.33 | 80.75 ± 0.53 | 89.23 ± 6.81 | 2.40 |
| | Random 25% | 78.74 ± 1.30 | 72.33 ± 1.28 | 86.84 ± 0.56 | 81.98 ± 1.34 | 90.09 ± 0.35 | 98.55 ± 0.55 | 85.94 ± 1.76 | 94.78 ± 0.28 | 80.16 ± 0.46 | 89.10 ± 6.71 | 4.30 |
| | Random 50% | 77.65 ± 1.46 | 72.11 ± 1.42 | 86.67 ± 0.48 | 79.23 ± 1.41 | 88.84 ± 0.42 | 98.61 ± 0.48 | 85.32 ± 1.75 | 95.17 ± 0.28 | 79.68 ± 0.50 | 87.56 ± 6.97 | 6.60 |
| | Random 75% | 76.56 ± 1.60 | 72.23 ± 1.33 | 86.72 ± 0.56 | 77.11 ± 1.28 | 87.07 ± 0.34 | 98.59 ± 0.76 | 84.88 ± 1.44 | 95.57 ± 0.34 | 79.49 ± 0.43 | 82.18 ± 6.58 | 7.30 |
| | Retention 25% | 79.09 ± 1.25 | 72.29 ± 1.17 | 86.95 ± 0.52 | 83.06 ± 1.09 | 90.63 ± 0.25 | 98.82 ± 0.50 | 85.56 ± 1.66 | 94.73 ± 0.34 | 80.27 ± 0.44 | 86.03 ± 5.20 | 3.85 |
| | Retention 50% | 77.77 ± 1.38 | 72.20 ± 1.20 | 86.82 ± 0.48 | 82.16 ± 1.27 | 90.17 ± 0.29 | 98.22 ± 0.28 | 84.80 ± 1.79 | 94.54 ± 0.22 | 79.96 ± 0.44 | 75.77 ± 6.86 | 6.70 |
| | Retention 75% | 77.05 ± 1.53 | 72.37 ± 1.20 | 86.79 ± 0.47 | 80.79 ± 0.95 | 88.97 ± 0.24 | 97.62 ± 0.30 | 84.35 ± 1.65 | 95.21 ± 0.30 | 79.40 ± 0.52 | 84.36 ± 6.77 | 6.70 |
| | Trimming 25% | 75.68 ± 1.21 | 72.15 ± 1.72 | 86.83 ± 0.47 | 75.94 ± 1.35 | 85.85 ± 0.40 | 99.87 ± 0.11 | 85.60 ± 1.92 | 95.02 ± 0.26 | 80.66 ± 0.56 | 91.92 ± 7.10 | 5.50 |
| | Trimming 50% | 77.76 ± 1.28 | 72.10 ± 1.49 | 86.96 ± 0.48 | 77.26 ± 1.17 | 86.57 ± 0.36 | 99.84 ± 0.11 | 84.58 ± 1.37 | 94.73 ± 0.28 | 80.18 ± 0.60 | 83.08 ± 8.72 | 6.45 |
| | Trimming 75% | 78.38 ± 1.30 | 72.22 ± 1.12 | 86.81 ± 0.53 | 78.78 ± 1.04 | 88.09 ± 0.27 | 99.73 ± 0.18 | 86.05 ± 1.60 | 94.61 ± 0.27 | 80.02 ± 0.57 | 89.36 ± 8.49 | 5.20 |
| HGNN | Original | 79.53 ± 1.33 | 72.24 ± 1.08 | 86.97 ± 0.55 | 83.45 ± 1.22 | 91.26 ± 0.26 | 98.94 ± 0.54 | 86.71 ± 1.48 | 94.50 ± 0.33 | 80.75 ± 0.52 | 89.23 ± 6.81 | 2.50 |
| | Random 25% | 78.74 ± 1.30 | 72.15 ± 1.36 | 86.84 ± 0.56 | 81.94 ± 1.31 | 90.11 ± 0.34 | 98.55 ± 0.55 | 85.82 ± 1.65 | 94.78 ± 0.28 | 80.16 ± 0.43 | 89.10 ± 6.71 | 4.70 |
| | Random 50% | 77.65 ± 1.46 | 72.20 ± 1.62 | 86.67 ± 0.48 | 79.20 ± 1.48 | 88.84 ± 0.42 | 98.61 ± 0.48 | 85.59 ± 1.49 | 95.17 ± 0.28 | 79.68 ± 0.50 | 87.56 ± 6.97 | 5.95 |
| | Random 75% | 76.56 ± 1.60 | 72.16 ± 1.56 | 86.72 ± 0.56 | 77.03 ± 1.37 | 86.95 ± 0.34 | 98.59 ± 0.76 | 85.12 ± 1.27 | 95.57 ± 0.34 | 79.50 ± 0.42 | 82.18 ± 6.58 | 7.30 |
| | Retention 25% | 79.09 ± 1.25 | 72.13 ± 1.17 | 86.95 ± 0.52 | 83.11 ± 1.09 | 90.61 ± 0.23 | 98.82 ± 0.50 | 85.33 ± 1.52 | 94.73 ± 0.34 | 80.23 ± 0.44 | 86.03 ± 5.20 | 4.25 |
| | Retention 50% | 77.77 ± 1.38 | 72.20 ± 1.32 | 86.82 ± 0.48 | 82.20 ± 1.29 | 90.21 ± 0.27 | 98.22 ± 0.28 | 84.51 ± 1.77 | 94.54 ± 0.22 | 79.93 ± 0.44 | 75.77 ± 6.86 | 6.45 |
| | Retention 75% | 77.02 ± 1.53 | 72.35 ± 1.40 | 86.79 ± 0.47 | 80.88 ± 0.93 | 89.02 ± 0.27 | 97.62 ± 0.30 | 84.19 ± 1.49 | 95.21 ± 0.30 | 79.36 ± 0.54 | 84.36 ± 6.77 | 6.60 |
| | Trimming 25% | 75.68 ± 1.21 | 71.91 ± 1.61 | 86.83 ± 0.47 | 75.73 ± 1.44 | 85.78 ± 0.41 | 99.87 ± 0.11 | 85.89 ± 1.67 | 95.02 ± 0.26 | 80.66 ± 0.56 | 91.92 ± 7.10 | 5.55 |
| | Trimming 50% | 77.76 ± 1.28 | 71.91 ± 1.50 | 86.96 ± 0.48 | 77.29 ± 1.18 | 86.48 ± 0.34 | 99.84 ± 0.11 | 84.67 ± 1.43 | 94.73 ± 0.28 | 80.18 ± 0.60 | 83.08 ± 8.72 | 6.30 |
| | Trimming 75% | 78.38 ± 1.30 | 72.07 ± 1.25 | 86.81 ± 0.53 | 78.78 ± 1.04 | 87.99 ± 0.34 | 99.73 ± 0.18 | 86.00 ± 1.55 | 94.61 ± 0.27 | 80.02 ± 0.57 | 89.36 ± 8.49 | 5.40 |
| HyperGCN | Original | 74.78 ± 1.11 | 66.06 ± 1.58 | 82.32 ± 0.62 | 77.48 ± 1.14 | 86.07 ± 3.32 | 69.51 ± 4.98 | 47.65 ± 5.01 | 46.10 ± 7.95 | 80.84 ± 0.49 | 51.54 ± 9.88 | 3.40 |
| | Random 25% | 35.60 ± 1.76 | 34.71 ± 1.62 | 68.80 ± 0.62 | 55.31 ± 1.83 | 81.18 ± 0.39 | 69.61 ± 4.77 | 57.42 ± 3.22 | 47.78 ± 7.33 | 77.50 ± 0.54 | 51.41 ± 9.82 | 6.05 |
| | Random 50% | 33.75 ± 2.58 | 39.94 ± 1.72 | 69.37 ± 0.59 | 40.11 ± 1.97 | 67.36 ± 2.94 | 67.59 ± 6.63 | 49.36 ± 3.42 | 48.12 ± 5.98 | 71.74 ± 0.58 | 51.67 ± 9.40 | 6.80 |
| | Random 75% | 42.42 ± 2.51 | 49.31 ± 1.85 | 70.99 ± 0.65 | 35.27 ± 1.94 | 50.33 ± 0.74 | 66.01 ± 8.15 | 45.31 ± 3.01 | 49.08 ± 2.52 | 62.76 ± 0.73 | 51.92 ± 9.02 | 6.60 |
| | Retention 25% | 37.56 ± 1.65 | 35.87 ± 1.80 | 68.73 ± 0.53 | 63.64 ± 1.22 | 84.26 ± 0.32 | 69.61 ± 4.81 | 61.33 ± 2.63 | 72.36 ± 3.39 | 79.24 ± 0.48 | 51.54 ± 8.84 | 4.55 |
| | Retention 50% | 34.87 ± 2.14 | 37.98 ± 1.70 | 69.04 ± 0.53 | 56.45 ± 1.70 | 77.98 ± 0.36 | 69.58 ± 4.75 | 76.59 ± 2.60 | 81.69 ± 1.75 | 75.60 ± 0.57 | 51.54 ± 9.45 | 4.70 |
| | Retention 75% | 36.71 ± 1.95 | 44.39 ± 1.69 | 69.98 ± 0.52 | 45.09 ± 2.09 | 63.78 ± 3.04 | 69.20 ± 5.16 | 77.44 ± 3.62 | 84.44 ± 2.23 | 67.99 ± 0.51 | 52.18 ± 8.61 | 4.60 |
| | Trimming 25% | 50.59 ± 1.72 | 55.15 ± 1.57 | 74.16 ± 0.66 | 52.78 ± 1.99 | 68.13 ± 0.79 | 52.37 ± 1.41 | 79.05 ± 2.74 | 81.31 ± 4.34 | 73.13 ± 0.92 | 46.67 ± 21.96 | 4.60 |
| | Trimming 50% | 36.20 ± 2.74 | 44.47 ± 1.38 | 71.35 ± 0.58 | 39.84 ± 2.35 | 55.53 ± 0.45 | 54.57 ± 7.40 | 59.52 ± 1.81 | 65.29 ± 3.52 | 68.07 ± 1.26 | 51.03 ± 10.20 | 6.70 |
| | Trimming 75% | 34.73 ± 1.52 | 36.60 ± 1.89 | 69.59 ± 0.54 | 37.81 ± 1.86 | 61.89 ± 1.82 | 61.73 ± 3.19 | 65.10 ± 2.77 | 73.05 ± 1.78 | 73.42 ± 0.63 | 50.90 ± 11.14 | 7.00 |
| MultiSetMixer | Original | 78.06 ± 1.24 | 71.85 ± 1.56 | 87.19 ± 0.53 | 82.74 ± 1.23 | 90.68 ± 0.19 | 99.58 ± 0.16 | 88.90 ± 1.30 | 98.38 ± 0.21 | 88.57 ± 1.96 | 88.08 ± 8.04 | 2.65 |
| | Random 25% | 77.73 ± 1.24 | 71.73 ± 1.73 | 86.98 ± 1.02 | 81.14 ± 1.19 | 89.74 ± 0.21 | 99.42 ± 0.30 | 88.00 ± 1.25 | 97.83 ± 0.20 | 80.84 ± 1.56 | 88.46 ± 7.00 | 4.75 |
| | Random 50% | 76.76 ± 1.28 | 71.69 ± 1.75 | 87.29 ± 0.63 | 78.32 ± 1.20 | 88.55 ± 0.31 | 99.43 ± 0.35 | 86.68 ± 1.55 | 97.46 ± 0.22 | 77.28 ± 1.16 | 86.33 ± 7.05 | 6.40 |
| | Random 75% | 76.06 ± 1.28 | 72.11 ± 1.81 | 87.31 ± 0.51 | 76.54 ± 1.41 | 86.32 ± 0.33 | 99.42 ± 0.48 | 85.50 ± 1.91 | 96.89 ± 0.26 | 77.35 ± 0.65 | 86.28 ± 8.14 | 7.25 |
| | Retention 25% | 77.88 ± 1.13 | 71.33 ± 1.43 | 86.86 ± 0.96 | 82.27 ± 1.40 | 90.30 ± 0.26 | 99.67 ± 0.18 | 87.24 ± 1.76 | 97.82 ± 0.27 | 87.12 ± 1.36 | 86.77 ± 8.32 | 4.40 |
| | Retention 50% | 76.86 ± 1.20 | 71.49 ± 1.76 | 87.02 ± 0.99 | 81.15 ± 1.06 | 89.95 ± 0.24 | 99.56 ± 0.30 | 86.11 ± 1.93 | 96.88 ± 0.27 | 85.15 ± 1.42 | 89.12 ± 7.59 | 5.10 |
| | Retention 75% | 76.29 ± 1.62 | 71.23 ± 2.00 | 87.16 ± 0.77 | 79.59 ± 1.05 | 88.75 ± 0.25 | 99.84 ± 1.00 | 85.16 ± 1.35 | 96.76 ± 0.32 | 83.87 ± 1.27 | 88.80 ± 5.97 | 7.00 |
| | Trimming 25% | 75.07 ± 1.44 | 72.41 ± 1.61 | 87.19 ± 0.56 | 75.84 ± 1.35 | 85.86 ± 0.31 | 99.95 ± 0.07 | 84.61 ± 1.47 | 96.68 ± 0.25 | 79.29 ± 0.55 | 88.85 ± 8.23 | 6.55 |
| | Trimming 50% | 76.34 ± 1.33 | 72.17 ± 1.46 | 87.42 ± 0.50 | 76.97 ± 1.50 | 86.46 ± 0.37 | 99.75 ± 0.13 | 85.29 ± 2.04 | 97.47 ± 0.26 | 74.42 ± 0.68 | 86.92 ± 8.24 | 5.70 |
| | Trimming 75% | 77.11 ± 1.37 | 71.76 ± 1.56 | 87.35 ± 0.56 | 77.80 ± 0.94 | 88.16 ± 0.25 | 99.59 ± 0.20 | 87.19 ± 1.58 | 97.73 ± 0.21 | 67.00 ± 1.22 | 88.43 ± 8.07 | 5.20 |

Table 6: Rewiring connectivity. Test accuracy in % averaged over 15 splits.

| Model | Type | Cora | Citeseer | Pubmed | CORA-CA | DBLP-CA | Mushroom | NTU2012 | ModelNet40 | 20Newsgroups | ZOO | avg. ranking |
|---|---|---|---|---|---|---|---|---|---|---|---|---|
| AllDeepSets | Label Based | 82.24 ± 1.12 | 75.65 ± 1.57 | 90.49 ± 0.40 | 91.12 ± 0.92 | 96.59 ± 0.17 | 99.96 ± 0.04 | 93.13 ± 1.29 | 99.52 ± 0.11 | 99.79 ± 0.13 | 91.54 ± 7.24 | 1.05 |
| | k-means | 75.20 ± 1.11 | 70.87 ± 1.54 | 88.96 ± 0.48 | 79.59 ± 1.42 | 89.75 ± 0.25 | 99.94 ± 0.09 | 84.23 ± 1.50 | 97.17 ± 0.13 | 81.18 ± 0.54 | 86.92 ± 7.73 | 2.80 |
| | Original | 77.11 ± 1.00 | 70.67 ± 1.42 | 89.04 ± 0.45 | 82.23 ± 1.46 | 91.34 ± 0.27 | 99.96 ± 0.05 | 86.49 ± 1.86 | 96.70 ± 0.25 | 81.19 ± 0.49 | 89.10 ± 7.00 | 2.15 |
| AllSetTransformer | Label Based | 83.43 ± 1.36 | 76.45 ± 1.43 | 90.19 ± 0.42 | 91.71 ± 0.89 | 96.75 ± 0.16 | 99.96 ± 0.05 | 94.81 ± 1.04 | 99.68 ± 0.09 | 99.93 ± 0.03 | 94.10 ± 6.91 | 1.05 |
| | k-means | 77.14 ± 1.46 | 72.83 ± 1.07 | 88.60 ± 0.43 | 81.92 ± 1.35 | 89.79 ± 0.30 | 99.96 ± 0.06 | 87.95 ± 1.28 | 97.29 ± 0.20 | 81.58 ± 0.55 | 88.72 ± 7.69 | 2.75 |
| | Original | 79.54 ± 1.02 | 72.52 ± 0.88 | 88.74 ± 0.51 | 84.43 ± 1.14 | 91.61 ± 0.19 | 99.95 ± 0.05 | 88.22 ± 1.42 | 98.00 ± 0.12 | 81.59 ± 0.59 | 91.03 ± 7.31 | 2.20 |
| UniGCNII | Label Based | 82.12 ± 1.11 | 75.23 ± 1.64 | 89.18 ± 0.50 | 89.80 ± 0.95 | 94.78 ± 0.13 | 99.93 ± 0.07 | 92.87 ± 1.32 | 99.31 ± 0.10 | 99.70 ± 0.10 | 94.74 ± 6.35 | 1.00 |
| | k-means | 76.49 ± 1.01 | 72.73 ± 1.50 | 88.02 ± 0.48 | 81.13 ± 1.41 | 90.13 ± 0.26 | 99.88 ± 0.07 | 88.05 ± 1.78 | 97.10 ± 0.21 | 81.06 ± 0.48 | 89.23 ± 7.52 | 3.00 |
| | Original | 78.46 ± 1.14 | 73.05 ± 1.48 | 88.07 ± 0.47 | 83.92 ± 1.02 | 91.56 ± 0.18 | 99.89 ± 0.07 | 88.24 ± 1.56 | 97.84 ± 0.16 | 81.16 ± 0.49 | 89.61 ± 8.09 | 2.00 |
| EDHNN | Label Based | 84.51 ± 1.23 | 76.76 ± 1.51 | 90.63 ± 0.53 | 92.28 ± 0.86 | 97.35 ± 0.15 | 99.96 ± 0.09 | 93.64 ± 1.12 | 99.59 ± 0.09 | 99.88 ± 0.08 | 92.44 ± 8.72 | 1.00 |
| | k-means | 78.43 ± 1.08 | 73.21 ± 1.25 | 88.98 ± 0.43 | 82.99 ± 1.33 | 90.45 ± 0.25 | 99.94 ± 0.08 | 86.91 ± 1.51 | 96.83 ± 0.16 | 81.34 ± 0.55 | 86.54 ± 7.68 | 2.95 |
| | Original | 80.74 ± 1.00 | 73.22 ± 1.14 | 89.12 ± 0.47 | 85.17 ± 1.02 | 91.94 ± 0.23 | 99.94 ± 0.11 | 88.04 ± 1.65 | 97.70 ± 0.19 | 81.64 ± 0.49 | 89.49 ± 6.99 | 2.05 |
| CEGAT | Label Based | 83.05 ± 1.08 | 77.82 ± 1.59 | 90.25 ± 0.39 | 91.42 ± 0.88 | 96.25 ± 0.13 | 99.91 ± 0.07 | 94.23 ± 0.77 | 99.26 ± 0.14 | OOM | 93.85 ± 7.39 | 1.00 |
| | k-means | 75.45 ± 1.54 | 72.57 ± 1.12 | 87.32 ± 0.47 | 77.11 ± 1.51 | 87.27 ± 0.29 | 97.66 ± 0.72 | 85.48 ± 1.66 | 96.39 ± 0.26 | OOM | 68.08 ± 8.28 | 2.33 |
| | Original | 76.53 ± 1.58 | 71.58 ± 1.11 | 87.11 ± 0.49 | 77.50 ± 1.51 | 88.74 ± 0.31 | 96.81 ± 1.41 | 82.27 ± 1.60 | 92.79 ± 0.44 | OOM | 44.62 ± 9.18 | 2.67 |
| CEGCN | Label Based | 83.70 ± 1.02 | 77.50 ± 1.53 | 90.08 ± 0.42 | 91.28 ± 0.97 | 96.68 ± 0.14 | 99.95 ± 0.05 | 94.03 ± 1.24 | 99.30 ± 0.14 | OOM | 95.00 ± 7.08 | 1.00 |
| | k-means | 75.89 ± 1.53 | 72.07 ± 1.18 | 87.13 ± 0.51 | 76.43 ± 1.41 | 86.76 ± 0.24 | 94.84 ± 0.47 | 85.34 ± 1.71 | 95.77 ± 0.31 | OOM | 73.72 ± 7.89 | 2.44 |
| | Original | 77.03 ± 1.31 | 70.87 ± 1.19 | 87.01 ± 0.62 | 77.55 ± 1.65 | 88.12 ± 0.25 | 94.91 ± 0.44 | 80.90 ± 1.74 | 90.04 ± 0.47 | OOM | 49.23 ± 6.81 | 2.56 |
| HCHA | Label Based | 84.06 ± 1.08 | 77.12 ± 1.37 | 88.81 ± 0.43 | 92.77 ± 0.73 | 96.70 ± 0.12 | 99.96 ± 0.06 | 95.21 ± 1.27 | 99.67 ± 0.09 | 99.93 ± 0.04 | 94.61 ± 6.97 | 1.00 |
| | k-means | 77.51 ± 1.41 | 72.62 ± 1.33 | 86.89 ± 0.48 | 81.19 ± 1.31 | 89.42 ± 0.29 | 99.56 ± 0.27 | 87.62 ± 1.33 | 96.98 ± 0.15 | 80.58 ± 0.57 | 84.10 ± 9.83 | 2.60 |
| | Original | 79.53 ± 1.33 | 72.57 ± 1.06 | 86.97 ± 0.55 | 83.53 ± 1.12 | 91.21 ± 0.28 | 98.94 ± 0.54 | 86.60 ± 1.96 | 94.50 ± 0.33 | 80.75 ± 0.53 | 89.23 ± 6.81 | 2.40 |
| HGNN | Label Based | 84.06 ± 1.08 | 77.11 ± 1.47 | 88.81 ± 0.43 | 92.86 ± 0.65 | 96.70 ± 0.11 | 99.96 ± 0.06 | 95.34 ± 1.07 | 99.67 ± 0.09 | 99.93 ± 0.04 | 94.61 ± 6.97 | 1.00 |
| | k-means | 77.51 ± 1.41 | 72.41 ± 1.55 | 86.89 ± 0.48 | 81.19 ± 1.38 | 89.42 ± 0.27 | 99.56 ± 0.27 | 87.52 ± 1.51 | 96.98 ± 0.15 | 80.58 ± 0.57 | 84.10 ± 9.94 | 2.60 |
| | Original | 79.53 ± 1.33 | 72.24 ± 1.08 | 86.97 ± 0.55 | 83.45 ± 1.22 | 91.26 ± 0.26 | 98.94 ± 0.54 | 86.71 ± 1.48 | 94.50 ± 0.34 | 80.75 ± 0.52 | 89.23 ± 6.81 | 2.40 |
| HyperGCN | Label Based | 72.88 ± 1.23 | 66.10 ± 1.79 | 82.18 ± 0.62 | 76.20 ± 1.50 | 84.86 ± 0.39 | 69.68 ± 4.90 | 43.37 ± 4.65 | 47.19 ± 6.42 | 82.14 ± 0.43 | 53.97 ± 8.24 | 1.50 |
| | k-means | 45.76 ± 1.97 | 49.96 ± 1.68 | 77.97 ± 0.75 | 47.63 ± 1.36 | 40.88 ± 4.04 | 69.53 ± 4.91 | 32.21 ± 2.57 | 41.96 ± 2.40 | 80.85 ± 0.46 | 53.46 ± 8.65 | 2.70 |
| | Original | 74.78 ± 1.11 | 66.06 ± 1.58 | 82.32 ± 0.62 | 77.48 ± 1.14 | 86.07 ± 3.32 | 69.51 ± 4.98 | 47.65 ± 5.01 | 46.10 ± 7.95 | 80.84 ± 0.49 | 51.54 ± 9.88 | 1.80 |
| MultiSetMixer | Label Based | 83.31 ± 1.09 | 74.98 ± 1.45 | 89.42 ± 0.56 | 92.34 ± 1.09 | 97.37 ± 0.20 | 99.98 ± 0.03 | 93.89 ± 1.34 | 99.49 ± 0.10 | 99.87 ± 0.06 | 91.28 ± 7.35 | 1.00 |
| | kmeans based | 76.02 ± 1.33 | 72.47 ± 1.50 | 87.43 ± 0.50 | 79.80 ± 1.50 | 88.38 ± 0.33 | 99.92 ± 0.08 | 87.55 ± 1.21 | 97.03 ± 0.19 | 80.54 ± 0.57 | 88.85 ± 8.65 | 2.60 |
| | Original | 78.06 ± 1.24 | 71.85 ± 1.50 | 87.19 ± 0.53 | 82.74 ± 1.23 | 90.68 ± 0.19 | 99.58 ± 0.16 | 88.90 ± 1.30 | 98.38 ± 0.21 | 88.57 ± 1.96 | 88.08 ± 8.04 | 2.40 |
| MLP CB | Label Based | 74.54 ± 1.51 | 72.41 ± 1.47 | 86.02 ± 0.50 | 74.71 ± 1.16 | 84.88 ± 0.38 | 99.97 ± 0.06 | 85.94 ± 1.59 | 96.38 ± 0.32 | 81.09 ± 0.52 | 87.56 ± 7.33 | 1.45 |
| | k-means | 74.53 ± 1.34 | 72.23 ± 1.55 | 85.99 ± 0.39 | 74.46 ± 1.32 | 84.78 ± 0.35 | 99.92 ± 0.07 | 86.16 ± 1.54 | 96.31 ± 0.27 | 81.09 ± 0.55 | 87.44 ± 7.75 | 2.25 |
| | Original | 74.06 ± 1.26 | 71.93 ± 1.53 | 85.83 ± 0.51 | 74.39 ± 1.40 | 84.91 ± 0.44 | 99.93 ± 0.08 | 85.43 ± 1.51 | 96.41 ± 0.32 | 86.13 ± 2.82 | 81.61 ± 10.98 | 2.30 |

### 5.4.2 Rewiring connectivity

In this experiment, we preserve the original connectivity while investigating the influence of homophilic hyperedges on performance. To do so, we adjust the given connectivity in two different ways.

- **Label-based Rewiring.** Our first strategy aims to unveil the full potential of the homophily measure for each dataset. To that end, we split hyperedges into fully homophilic ones based on their true *node labels*. While this approach is not applicable to practical evaluations, it represents a meaningful baseline to assess the meaningfulness of pure homophilic connections.

- **Feature-based Rewiring.** In contrast, the second strategy explores the possibility of partitioning hyperedges based on their *initial node features*. This approach, supported by the hypothesis that nodes of the same class might have similar features, represents an attempt of easily dividing hyperedges into more homophilic connections –its practicality assured given that only features are taken into account. In particular, the hyperedge splitting results from applying multiple times the $k$-means algorithm for each hyperedge $e \in \mathcal{E}$. At each iteration, the number of centroids $m$ is varied from 2 to $\min(C, |e|)$, where we recall $C$ is the number of classes; the elbow method is then used to determine the $m$ value for the optimal hyperedge partitioning.

**Results.** As shown in Table 6, the Label-based strategy enhances performance for all datasets and models; as expected, pure homophilic connections enhance performance. Notably, the graph-based method CEGCN achieves similar results to HNNs with this strategy. Additionally, on average, only CEGCN performs better with the $k$-means strategy, and this method also mitigates distribution shifts for MultiSetMixer. These findings collectively suggest the crucial role of connectivity preprocessing, especially for graph-based models.

### 5.5 Benefits and challenges of hyperedge-dependent node representations in hypergraphs

In this section, we provide a deeper comparison between lifted models, such as AllSetTransformer, AllDeepSets, and EDHNN, and the MultiSetMixer, which allows for hyperedge-dependent node representation and thus leverages specific characteristics of hypergraph networks.

Table 1 highlights MultiSetMixer's superior performance on three datasets: NTU2012, ModelNet40, and 20Newsgroups. The improved results on 20Newsgroups can be attributed to two aspects (i) the pooling

operations over the hyperedge-dependent node representations, and (ii) distribution shift (see Section 5.3). Regarding (i), Table 1 highlights that the distribution shift affects the MLP CB model, resulting in a performance of $86.13 \pm 2.82$, which already outperforms all HNN baselines. However, the MultiSetMixer achieves an even higher performance of $88.57 \pm 1.96$. Notably, the best-performing HNN baseline achieves $81.64 \pm 0.49$, while the MLP baseline achieves $80.93 \pm 0.59$, showing an absolute difference of approximately 0.5%. In contrast, the MultiSetMixer leverages the introduced MultiSet message-passing mechanism to achieve $88.57 \pm 1.96$, resulting in an improvement of 2%. Our hypothesis is that, in this case, MultiSetMixer works by giving each document a separate representation for every group (hyperedge) to which it belongs. This means the model can adjust how it sees a document depending on the context of each group. By doing this, it captures how a document is similar to or different from others in each group, which is a strong clue for classification. This flexibility makes it much better at handling situations where important relationships aren't limited to one group but spread across many. Traditional methods, with unique representations, struggle in these cases because they miss those cross-group connections. Regarding (ii), the uncontrolled distribution shift might negatively impact MultiSetMixer's performance on the Zoo, Mushroom, and Pubmed datasets. As shown in Sections 5.4 and 5.4.2, this distribution shift can be mitigated by modifying the initial connectivity. Specifically, Table 6 demonstrates that applying $k$-means clustering to adjust the initial connectivity allows MultiSetMixer to achieve better performance on Zoo, Mushroom, and Pubmed. In the case of Mushroom, this approach enables performance comparable to models like AllSetTransformer, AllDeepSets, and EDHNN. At the same time, modifying the connectivity on 20Newsgroups mitigates the distribution shift but results in a performance decline, bringing MultiSetMixer's results closer to those of AllSetTransformer, AllDeepSets, and EDHNN.

Additionally, we observed that MultiSetMixer excels in processing Computer Vision/Graphics datasets such as NTU2012 and ModelNet40. This success is due to the initial graph construction, which involves lifting the k-uniform graph by constructing hyperedges based on the one-hop neighborhood of each node. On the Cora and Citeseer datasets, MultiSetMixer outperforms AllDeepSets and performs comparably to AllSetTransformer, all without requiring an attention mechanism. However, on CORA-CA, AllSetTransformer outperforms MultiSetMixer, which instead produces results similar to AllDeepSets. The strongest connectivity effects appear in naturally-occurring hypergraph datasets like CORA-CA and DBLP-CA, where hyperedges are derived from metadata rather than constructed from existing graphs. In these cases, MultiSetMixer's underperformance suggests the need to explore alternative MultiSet framework architectures, including attention-based approaches and permutation-equivariant continuous hyperedge diffusion operators similar to EDHNN. Moreover, MultiSetMixer's lower performance on Pubmed and DBLP-CA can be attributed to the difficulty of processing the entire hypergraph in a single forward pass due to memory constraints when storing all hyperedge-dependent node representations. However, employing the proposed mini-batching scheme still allows MultiSetMixer to achieve strong performance on these datasets.

At the conclusion of this section, we summarize the main benefits and challenges of hyperedge-dependent node representations in hypergraphs. The key advantage lies in their ability to model and capture complex interactions within hypergraph structures, as demonstrated by their superior performance on certain datasets, especially those involving computer vision and large-scale hyperedges. However, challenges arise in managing the scalability of these models, particularly in scenarios with large hyperedges or datasets, where memory constraints and distribution shifts can negatively impact performance. Despite these issues, careful modifications to the connectivity and the use of mini-batching strategies can mitigate some of the limitations, offering a promising path forward for further optimization.

## 6 Conclusion and Discussion

This section summarizes the key findings of our extensive evaluation and the proposed frameworks, relating them to the fundamental questions that motivated our work.

**Q1**: **Can the concept of homophily play a crucial role in HNNs, similar to its significance in graph-based research?** We show that the concept of homophily in higher-order networks is considerably more complicated compared to networks that exhibit only pairwise connections. To address the issue, we introduce a novel message passing homophily framework that is capable of characterizing homophily

in hypergraphs through the distribution of node features as well as node class distribution. In Section 3, we present $\Delta$ homophily, based on the dynamic nature of the proposed message passing homophily, showing that it correlates better with HNN models' performance than classical homophily measures over the clique-expanded hypergraph. Our findings underscore the crucial role of accurately expressing homophily in HNNs, emphasizing its complexity and the potential in capturing higher-order dynamics. Moreover, in our experiments (see Section 5.4.2), we demonstrate that rewiring hyperedges for perfect homophily leads to similar results for graph-based methods (CEGCN, CEGAT) and HNN models. In conclusion, the proposed framework offers a robust foundation for exploring homophily in higher-order networks and can be seamlessly adapted to incorporate other characteristics (such as the distribution of node features, which parallel the node label homophily examined in this study). While we presented this framework in the context of hypergraphs, its adaptability allows for straightforward extensions to other topological domains.

**Q2**: **Given that current HNNs are predominantly extensions of GNN architectures adapted to the hypergraph domain, are these extended methodologies suitable, or should we explore new strategies tailored specifically for handling hypergraph-based data?** The three main contributions presented in this paper –Message Passing Homophily, the MultiSet framework (integrating existing HNNs within a unified framework), and the formulation of a novel mini-batch sampling scheme to address scalability issues– are directly inspired by the inherent properties of hypernetworks and their higher-order dynamics. Based on our experimental results and analysis, the proposed methodologies open an interesting discussion about the impact of ways of processing hypergraph data and defining HNNs. For instance, our mini-batching sampling strategy –which helps address scalability issues of current solutions– allowed us to realize the implicit introduction of node class distribution shifts in the process. This study could potentially lead to the definition of meaningful connectivity rewiring techniques, as we already explore in Section 5.4. Furthermore, we show that the introduced message passing and $\Delta$ homophily measures allow for a deeper understanding of the hypernetwork topology and its correlation to the HNN models' performances. Overall, this study focuses on comparing graph-based extensions for modeling hypergraph networks and methodologies that incorporate hypergraph-specific characteristics. This approach allows for identifying common failure modes in current hypergraph modeling techniques (Section 5). We argue that these contributions offer a new *perspective* on processing higher-order networks that extend beyond the graph domain.

**Q3**: **Are the existing hypergraph benchmarking datasets meaningful and representative enough to draw robust and valid conclusions?** In Sections 5.4 and 5.4.2, we demonstrate that the significant performance gap of HNNs models on Cora, CORA-CA, and DBLP-CA is primarily attributed to the hyperedges with the largest cardinalities. Further analysis using $\Delta$ homophily reveals that their notable improvement is strongly tied to the homophilic nature of the one-hop neighborhood. Additionally, the experimental results in Section 5.1 and 5.4 highlight challenges for current HNNs with certain benchmark hypergraph datasets. Specifically, we find that lifted HNN models ignore connectivity for Citeseer, Pubmed, and 20Newsgroups, as well as for the Mushroom dataset, due to highly discriminative features. Furthermore, we observe that models that do not rely on inductive bias (i.e. do not use connectivity in the architecture), consistently exhibit good performance across the majority of datasets. This suggests that the expressive power of node features alone is sufficient for efficient task execution. Addressing this gap presents an open challenge for future research endeavors, and we posit the necessity for additional benchmark datasets where connectivity plays a pivotal role. In addition to this, we believe it would be also interesting to analyze datasets involving higher-order relationships where node classes explicitly depend on hyperedges, as introduced in Choe et al. (2023). This represents an insightful line of research to further exploit hyperedge-based node representations.

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

# Supplementary Materials

## A  Extended Related Works on Hypergraph Neural Networks

Numerous machine-learning techniques have been developed for processing hypergraph data. One commonly used approach in early literature is to transform the hypergraph into a graph through clique expansion (CE). This technique involves substituting each hyperedge with an edge for every pair of vertices within the hyperedge, creating a graph that can be analyzed using graph-based algorithms (Agarwal et al., 2006; Zhou et al., 2006; Zhang et al., 2018; Li & Milenkovic, 2017).

Several techniques have been proposed that use Hypergraph Neural Networks (HNNs) for semi-supervised learning. One of the earliest methods extends the graph convolution operator by incorporating the normalized hypergraph Laplacian (Feng et al., 2019). As pointed out in Dong et al. (2020), spectral convolution with the normalized Laplacian corresponds to performing a weighted CE of the hypergraph. HyperGCN (Yadati et al., 2019) employs mediators for incomplete CE on the hypergraph, which reduces the number of edges required to represent a hyperedge from a quadratic to a linear number of edges. The information diffusion is then carried out using spectral convolution for hypergraph-based semi-supervised learning. Hypergraph Convolution and Hypergraph Attention (HCHA) (Bai et al., 2021) employs modified degree normalizations and attention weights, with the attention weights depending on node and hyperedge features.

CE may cause the loss of important structural information and result in suboptimal learning performance (Hein et al., 2013; Chien et al., 2022). Furthermore, these models typically obtain the best performance with shallow 2-layer architectures. Adding more layers can lead to reduced performance due to oversmoothing (Huang & Yang, 2021). In the recent study Chen & Zhang (2022), an attempt was made to address oversmoothing in this type of network by incorporating residual connections; however, the method still relies on using hypergraph Laplacians to build a weighted graph through clique expansion. Another method presented in Yang et al. (2020) introduces a new hypergraph expansion called line expansion (LE) that treats vertices and hyperedges equally. The LE bijectively induces a homogeneous structure from the hypergraph by modeling vertex-hyperedge pairs. In addition, the LE and CE techniques require significant computational resources to transform the original hypergraph into a graph and perform subsequent computations, hence making the methods unpractical for large hypergraphs.

Another line of research explores hypergraph modeling involving a two-stage procedure: information is transmitted from nodes to hyperedges and then back from hyperedges to nodes (Wei et al., 2021; Yi & Park, 2020; Dong et al., 2020; Arya et al., 2020; Huang & Yang, 2021; Yadati et al., 2020). This procedure can be viewed as a two-step message passing mechanism. HyperSAGE (Arya et al., 2020) is a prominent early example of this line of research allowing transductive and inductive learning over hypergraphs. Although HyperSAGE has shown improvement in capturing information from hypergraph structures compared to spectral-based methods, it involves only one learnable linear transformation and cannot model arbitrary multiset function (Chien et al., 2022). Moreover, the algorithm utilizes nested loops resulting in inefficient computation and poor parallelism.

UniGNN (Huang & Yang, 2021) addresses some of these limitations by using a permutation-invariant function to aggregate vertex features within each hyperedge in the first stage and using learnable weights only during the second stage to update each vertex with its incident hyperedges. One of the variations of UniGNN, called UniGCNII addresses the oversmoothing problem, which is common for most of the methods described above. It accomplishes this by adapting GCNII (Chen et al., 2020) to hypergraphs. The AllSet method, proposed in Chien et al. (2022), employs a composition of two learnable multiset functions to model hypergraphs. It presents two model variations: the first one exploits Deep Set (Zaheer et al., 2017) and the second one Set Transformer (Lee et al., 2019). The AllSet method can be seen as a generalization of the most commonly used hypergraph HNNs (Yadati et al., 2019; Feng et al., 2019; Bai et al., 2021; Dong et al., 2020; Arya et al., 2020). More implementation details and detailed drawbacks discussion can be found in Section 4.1. Although AllSet achieves state-of-the-art results, it suffers from the drawbacks of the message passing mechanism, including the local receptive field, resulting in a limited ability to model long-range interactions (Gu et al.,

2020; Balcilar et al., 2021). Two additional issues are poor scalability to large hypergraph structures and oversmoothing that occurs when multiple layers are stacked.

Finally, we would like to mention two related papers that put the focus on hyperedge-dependent computations. On the one hand, EDHNN (Wang et al., 2023) incorporates the option of hyperedge-dependent messages from hyperedges to nodes; however, at each iteration of the message passing it aggregates all these messages to generate a unique node hidden representation, and thus it doesn't enable to keep different hyperedge-based node representations across the whole procedure –as our MultiSetMixer does. On the other hand, the work Aponte et al. (2022) does allow multiple hyperedge-based representations across the message passing, but the theoretical formulation of this unpublished paper is not clear and rigorous, and the evaluation is neither reproducible nor comparable to other hypergraph models. Hence, we argue that our MultiSet framework represents a step forward by rigorously formulating a simple but general MP framework for hypergraph modeling that is flexible enough to deal with hyperedge-based node representations and residual connections, demonstrating as well that it generalizes previous hypergraph and graph models.

## B Details of the Implemented Methods

We provide a detailed overview of the models analyzed and tested in this work. In order to make their similarities and differences more evident, we express their update steps through a standard and unified notation.

**Notation.** A hypergraph with $n$ nodes and $m$ hyperedges can be represented by an incidence matrix $\boldsymbol{H} \in \mathbb{R}^{n \times m}$. If the hyperedge $e_j$ is incident to a node $v_i$ (meaning $v_i \in e_j$), the entry in the incidence matrix $H_{i,j}$ is set to 1. Instead, if $v_i \notin e_j$, then $H_{i,j} = 0$.

We denote with $\boldsymbol{W}$ and $\boldsymbol{b}$ a learnable weight matrix and bias of a neural network, respectively. Generally, $\boldsymbol{x}_v$ and $\boldsymbol{z}_e$ are used to denote features for a node $v$ and a hyperedge $e$ respectively. Stacking all node features together we obtain the node feature matrix $\boldsymbol{X}$, while $\boldsymbol{Z}$ is instead the hyperedge feature matrix. $\sigma(\cdot)$ indicates a nonlinear activation function (such as ReLU, ELU or LeakyReLU) that depends on the model used. Finally, we use $\|$ to denote concatenation.

### B.1 AllSet-like models

This Section addresses the models that are covered in the AllSet unified framework introduced in 4.1, and that can potentially be expressed as particular instances of equations 5 and 7. For a detailed proof of the claim for most of the following models, refer to Theorem 3.4 in Chien et al. (2022).

**CEGCN / CEGAT.** As introduced in the previous Sections, the CE of a hypergraph $\mathcal{G} = (\mathcal{V}, \mathcal{E})$ is a weighted graph obtained from $\mathcal{G}$ with the same set of nodes. In terms of incidence matrix, it can be described as $\boldsymbol{H}^{(CE)} = \boldsymbol{H}\boldsymbol{H}^T$ (Chien et al., 2022). A one-step update of the node feature matrix $\boldsymbol{X} \in \mathbb{R}^{n \times f}$ can be expressed both in a compact way as $\boldsymbol{H}^{(CE)}\boldsymbol{X}$ or directly as a node-level update rule, as

$$\boldsymbol{x}_v^{(t+1)} = \sum_{e \in \mathcal{E}_v} \sum_{u:u \in e} \boldsymbol{x}_v^{(t)}. \tag{15}$$

Some types of hypergraph convolutional layers in the literature adopt a CE-based propagation, for example generalizing popular graph-targeting models such as Graph Convolutional Networks (Kipf & Welling, 2017) and Graph Attention Networks (Veličković et al., 2017).

**HNN.** Before describing how HNN (Feng et al., 2019) works, it is necessary to define some notation. Let $\boldsymbol{H}$ be the hypergraph's incidence matrix. Suppose that each hyperedge $e \in \mathcal{E}$ is assigned a fixed positive weight $z_e$, and let $\boldsymbol{Z} \in \mathbb{R}^{m \times m}$ now denote the matrix stacking all these weights in the diagonal entries. Additionally, the vertex degree is defined as

$$d_v = \sum_{e \in \mathcal{E}_v} z_e, \tag{16}$$

while the hyperedge degree, instead, is

$$b_e = \sum_{v:v \in e} 1. \tag{17}$$

The degree values can be used to define two diagonal matrices, $\boldsymbol{D} \in \mathbb{R}^{n \times n}$ and $\boldsymbol{B} \in \mathbb{R}^{m \times m}$.

The core of the hypergraph convolution introduced in Feng et al. (2019) can be expressed as

$$\boldsymbol{x}_v^{(t+1)} = \sigma \left( \sum_{e \in \mathcal{E}_v} \sum_{u:u \in e} z_e \boldsymbol{x}_v^{(t)} \boldsymbol{W}^{(t)} \right), \tag{18}$$

where $\sigma$ is a non-linear activation function like LeakyReLU and ELU, and $\boldsymbol{W}^{(t)} \in \mathbb{R}^{f^{(t)} \times f^{(t+1)}}$ is a weight matrix between the $(t)$-th and $(t+1)$-th layer, to be learnt during training. Note that in this case the dimensionality of the node feature vectors $f^{(t)}$ can be layer-dependent.

The update step can be rewritten also in matrix form as

$$\boldsymbol{X}^{(t+1)} = \sigma(\boldsymbol{H}\boldsymbol{Z}\boldsymbol{H}^T\boldsymbol{X}^{(t)}\boldsymbol{W}^{(t)}), \tag{19}$$

where $\boldsymbol{X}^{(t+1)} \in \mathbb{R}^{n \times f^{(t+1)}}$ and $\boldsymbol{X}^{(t)} \in \mathbb{R}^{n \times f^{(t)}}$.

In practice, a normalized version of this update procedure is proposed. The matrix-based formulation allows to clearly express the symmetric normalization that is actually put in place through the vertex and hyperedge degree matrices $\boldsymbol{D}$ and $\boldsymbol{B}$ defined above:

$$\boldsymbol{X}^{(t+1)} = \sigma(\boldsymbol{D}^{-1/2}\boldsymbol{H}\boldsymbol{Z}\boldsymbol{B}^{-1}\boldsymbol{H}^T\boldsymbol{D}^{-1/2}\boldsymbol{X}^{(t)}\boldsymbol{W}^{(t)}). \tag{20}$$

**HCHA.** With respect to the previously described models, HCHA (Bai et al., 2021) uses a different kind of weights that depend on the node and hyperedge features. Specifically, starting from the same convolutional model proposed by Feng et al. (2019) and described in Equation 20, they explore the idea of introducing an attention learning model on $\boldsymbol{H}$.

Their starting point is the intuition that hypergraph convolution as implemented in Equation 20 implicitly puts in place some attention mechanism, which derives from the fact that the afferent and efferent information flow to vertexes may be assigned different importance levels, which are statically encoded in the incidence matrix $\boldsymbol{H}$, hence depend only on the graph structure. In order to allow for such information on magnitude of importance to be determined dynamically and possibly vary from layer to layer, they introduce an attention learning module on the incidence matrix $\boldsymbol{H}$: instead of maintaining $\boldsymbol{H}$ as a binary matrix with predefined and fixed entries depending on the hypergraph connectivity, they suggest that its entries could be learnt during the training process. The entries of the matrix should express a probability distribution describing the degree of node-hyperedge connectivity, through non-binary and real values.

Nevertheless, the proposed hypergraph attention is only feasible when the hyperedge and vertex sets share the same homogeneous domain, otherwise, their similarities would not be compatible. In case the comparison is feasible, the computation of attention scores is inspired by (Veličković et al., 2017): for a given vertex $v$ and a hyperedge $e$, the score is computed as

$$H_{v,e} = \frac{\exp(\sigma(\mathrm{sim}(\boldsymbol{x}_v\boldsymbol{W}, \boldsymbol{z}_e\boldsymbol{W})))}{\sum_{g \in \mathcal{E}_v} \exp(\sigma(\mathrm{sim}(\boldsymbol{x}_v\boldsymbol{W}, \boldsymbol{z}_g\boldsymbol{W})))}, \tag{21}$$

where $\sigma$ is a non-linear activation function, and sim is a similarity function defined as

$$\mathrm{sim}(\boldsymbol{x}_v, \boldsymbol{z}_e) = \boldsymbol{a}^T[\boldsymbol{x}_v \parallel \boldsymbol{z}_e], \tag{22}$$

in which $\boldsymbol{a}$ is a weight vector, and the resulting similarity value is a scalar.

**HyperGCN.** The method proposed by Yadati et al. (2019) can be described as performing two steps sequentially: first, a graph structure is defined starting from the input hypergraph, through a particular procedure, and then the well known CGN model (Kipf & Welling, 2017) for standard graph structures is executed on it. Depending on the approach followed in the first step, three slight variations of the same model can be identified: 1-HyperGCN, HyperGCN (enhancing 1-HyperGCN with so-called *mediators*) and FastHyperGCN.

Before analyzing the differences among the three techniques, we introduce some notation and express how the GCN-update step is performed. Suppose that the input hypergraph $\mathcal{G} = (\mathcal{V}, \mathcal{E})$ is equipped with initial edge weights $\{z_e\}_{e \in \mathcal{E}}$ and node features $\{\boldsymbol{x}_v\}_{v \in \mathcal{V}}$ (if missing, suppose to initialize them randomly or with constant values). Let $\bar{\boldsymbol{A}}^{(t)}$ denote the normalized adjacency matrix associated to the graph structure at time-step $(t)$. The node-level one-step update for a specific node $v$ can be formalized as:

$$\boldsymbol{x}_v^{(t+1)} = \sigma\left(\left(\boldsymbol{W}^{(t)}\right)^T \sum_{u \in \mathcal{E}_v} \bar{A}_{u,v}^{(t)} \cdot \boldsymbol{x}_u^{(t)}\right), \tag{23}$$

in which $\boldsymbol{x}_v^{(t+1)}$ is the $(t+1)$-th step hidden representation of node $v$ and $\mathcal{E}_v$ is the set of neighbors of $v$. For what concerns $\bar{A}_{u,v}^{(t)}$, it refers to the element at index $u,v$ of $\bar{\boldsymbol{A}}^{(t)}$, which can be defined in the following ways according to the method:

1. 1-HyperGCN: starting from the hypergraph $\mathcal{G} = (\mathcal{V}, \mathcal{E})$, a simple graph is defined by considering exactly one representative simple edge for each hyperedge $e \in \mathcal{E}$, and it is defined as $(v_e, u_e)$ such that $(v_e, u_e) = \mathrm{argmax}_{v,u \in e} \|\left(\boldsymbol{W}^{(t)}\right)^T (\boldsymbol{x}_v^{(t)} - \boldsymbol{x}_u^{(t)})\|_2$. This implies that each hyperedge $e$ is represented by just one pairwise edge $(v_e, u_e)$, and this may also change from one step to the other, which leads to the graph adjacency matrix $\bar{\boldsymbol{A}}^{(t)}$ being layer-dependent, too.

2. HyperGCN: the model extends the graph construction procedure of 1-HyperGCN by also considering *mediator* nodes, that for each hyperedge $e$ consist in $K_e := \{k \in e : k \neq v_e, k \neq u_e\}$. Once the representative edge $(v_e, u_e)$ is determined and added to the newly defined graph, two edges for each mediator are also introduced, connecting the mediator to both $v_e$ and $u_e$. Because there are $2|e| - 3$ edges for each hyperedge $e$, each weight is chosen to be $\frac{1}{2|e|-3}$ in order for the weights in each hyperedge to sum to 1. The generalized Laplacian obtained this way satisfies all the properties of the HyperGCN's Laplacian (Yadati et al., 2019).

3. FastHyperGCN: in order to save training time, in this case the adjacency matrix $\bar{\boldsymbol{A}}^{(t)}$ is computed only once before training, by using only the initial node features of the input hypergraph.

**UniGCNII.** This model aims to extend to hypergraph structures the GCNII model proposed by Chen et al. (2020) for simple graph structures, that is a deep graph convolutional network that puts in place an initial residual connection and identity mapping as a way to reduce the oversmoothing problem (Huang & Yang, 2021).

Let $d_v$ denote the degree of vertex $v$, while $d_e = \frac{1}{|e|}\sum_{i \in e} d_i$ for each hyperedge $e \in \mathcal{E}$. A single node-level update step performed by UniGCNII can be expressed as:

$$\hat{\boldsymbol{x}}_v^{(t)} = \frac{1}{\sqrt{d_v}} \sum_{e \in \mathcal{E}_v} \frac{\boldsymbol{z}_e^{(t)}}{\sqrt{d_e}}, \tag{24}$$

$$\boldsymbol{x}_v^{(t+1)} = ((1 - \beta)\boldsymbol{I} + \beta\boldsymbol{W}^{(t)})((1 - \alpha)\hat{\boldsymbol{x}}_v^{(t)} + \alpha\boldsymbol{x}_v^{(0)}). \tag{25}$$

in which $\alpha$ and $\beta$ are hyperparameters, $\boldsymbol{I}$ is identity matrix and $\boldsymbol{x}_v^{(0)}$ is the initial feature of vertex $i$.

**HNHN.** For the HNHN model by Dong et al. (2020), hypernode and hyperedge features are supposed to share the same dimensionality $d$, hence in this case $\boldsymbol{X} \in \mathbb{R}^{n \times d}$ and $\boldsymbol{Z} \in \mathbb{R}^{m \times d}$. The update rule in this case can be easily expressed using the incidence matrix as

$$\boldsymbol{Z}^{(t+1)} = \sigma(\boldsymbol{H}^T\boldsymbol{X}^{(t)}\boldsymbol{W}_Z + \boldsymbol{b_Z}), \tag{26}$$

$$\boldsymbol{X}^{(t+1)} = \sigma(\boldsymbol{H}\boldsymbol{Z}^{(t+1)}\boldsymbol{W}_X + \boldsymbol{b_X}). \tag{27}$$

in which $\sigma$ is a nonlinear activation function, $\boldsymbol{W}_X, \boldsymbol{W}_Z \in \mathbb{R}^{d \times d}$ are weight matrices, and $\boldsymbol{b_X}, \boldsymbol{b_Z} \in \mathbb{R}^d$ are bias terms.

**AllSet.** The general formulation for the propagation setting of AllSet (Chien et al., 2022) is introduced in Subsection 4.1 and, starting from that, we now analyze the different instances of the model obtained by imposing specific design choices in the general framework.

In the practical implementation of the model, the update functions $f_{\mathcal{V} \to \mathcal{E}}$ and $f_{\mathcal{E} \to \mathcal{V}}$, that are required to be permutation invariant with respect to their first input, are parametrized and *learnt* for each dataset and task. Furthermore, the information of their second argument is not utilized in practice, hence their input can be more generally denoted as a set $\mathbb{S}$.

The two architectures AllDeepSets and AllSetTransformer are obtained in the following way, depending on whether the update functions are defined either as MLPs or Transformers:

1. AllDeepSets (Chien et al., 2022): $f_{\mathcal{V} \to \mathcal{E}}(\mathbb{S}) = f_{\mathcal{E} \to \mathcal{V}}(\mathbb{S}) = \mathrm{MLP}(\sum_{s \in \mathbb{S}} \mathrm{MLP}(s))$;

2. AllSetTransformer (Chien et al., 2022), in which the update functions are defined iteratively through multiple steps as they were first designed by Vaswani et al. (2017).

   The first set of operations corresponds to the self-attention module. Suppose that $h$ attention heads are considered: first of all, $h$ pairs of matrices $\boldsymbol{K}_i$ (keys) and $\boldsymbol{V}_i$ (values) with $i \in \{1, ..., h\}$ are computed from the input set through different MLPs. Additionally, $h$ weights $\theta_i, i \in \{1, ..., h\}$ are also learned and together with the keys and values they allow for the computation of each head-specific attention value $\boldsymbol{O}_i$ using an activation function $\omega$ (Vaswani et al., 2017). The $h$ attention heads are processed in parallel and they are then concatenated, leading to a unique vector being the result of the multi-head attention module $\mathrm{MH}_{h,\omega}$. After that, a sum operation and a Layer Normalization (LN) (Ba et al., 2016) are applied:

$$\boldsymbol{K}_i = \mathrm{MLP}_i^K(\mathbb{S}), \boldsymbol{V}_i = \mathrm{MLP}_i^V(\mathbb{S}), \quad \text{where} \quad i \in \{1, ..., h\}, \tag{28}$$

$$\theta \triangleq \|_{i=1}^h \theta_i, \tag{29}$$

$$\boldsymbol{O}_i = \omega(\theta_i(\boldsymbol{K}_i)^T)\boldsymbol{V}_i, \quad \text{where} \quad i \in \{1, ..., h\}, \tag{30}$$

$$\mathrm{MH}_{h,\omega}(\theta, \mathbb{S}, \mathbb{S}) = \|_{i=1}^h \boldsymbol{O}^{(i)}, \tag{31}$$

$$\boldsymbol{Y} = \mathrm{LN}(\theta + \mathrm{MH}_{h,\omega}(\theta, \mathbb{S}, \mathbb{S})). \tag{32}$$

   A feed-forward module follows the self-attention computations, in which a MLP is applied to the feature matrix and then sum and LN are performed again, corresponding to the last operations to be performed:

$$f_{\mathcal{V} \to \mathcal{E}}(\mathbb{S}) = f_{\mathcal{E} \to \mathcal{V}}(\mathbb{S}) = \mathrm{LN}(\boldsymbol{Y} + \mathrm{MLP}(\boldsymbol{Y})). \tag{33}$$

## B.2 Other models

This Section describes the models that are considered for the experiments but that don't fall directly under the AllSet unified framework defined in Section 4.1.

**EDHNN.** The Equivariant Diffusion-based HNN model, shortened as EDHNN (Wang et al., 2023) represents the first attempt to draw a connection between the class of hypergraph diffusion algorithms and the design of Hypergraph Neural Networks. The underlying motivation is that, by enabling the model to approximate any continuous equivariant hypergraph diffusion operator, a broad spectrum of higher-order relations can be encoded.

In EDHNN, the hypergraph diffusion operators are learned directly from data, harnessing the expressive power of Neural Networks. This leads to the development of a novel Hypergraph Neural Network inspired by hypergraph diffusion solvers, with the subsequent operations executed at each layer described in the following.

The hyperedge-level feature update is performed as

$$\boldsymbol{z}_e^{(t+1)} = \sum_{u \in e} \hat{\phi}(\boldsymbol{x}_u^{(t)}), \tag{34}$$

and starting from that, the node-level update is defined as

$$\boldsymbol{x}_v^{(t+1)} = \hat{\psi} \left( \boldsymbol{x}_v^{(t)}, \sum_{e \in \mathcal{E}_v} \hat{\rho}(\boldsymbol{x}_v^{(t)}, \boldsymbol{z}_e^{(t+1)}), \boldsymbol{x}_v^0, d_v \right). \tag{35}$$

In the equations above, $\hat{\psi}$, $\hat{\phi}$ and $\hat{\rho}$ are three MLPs shared across layers.

**HAN.** The Heterogeneous Graph Attention Network model (Wang et al., 2019) is specifically designed for processing and performing inference on heterogeneous graphs. Heterogeneous graphs have various types of nodes and/or edges, and standard GNN models that treat all of them equally are not able to properly handle such complex information.

In order to apply this model on hypergraphs, (Chien et al., 2022) define a preprocessing step to derive a heterogeneous graph from a hypergraph. Specifically, a bipartite graph is defined such that there is a bijection between its nodes and the set of nodes and hyperedges in the original hypergraph. The nodes obtained in this way belong to one of two distinct types, that are the sets $\mathbb{V}$ and $\mathbb{E}$ (if they correspond to either a node or a hyperedge in the original hypergraph, respectively). Edges only connect nodes of two different types, and one edge exists between a node $u_v \in \mathbb{V}$ and a node $u_e \in \mathbb{E}$ if and only if $v \in e$ in the input hypergraph. We consider two types of so-called meta-paths (in this case, paths of length 2) in the heterogeneous graph, that are $\mathbb{V} \to \mathbb{E} \to \mathbb{V}$ and $\mathbb{E} \to \mathbb{V} \to \mathbb{E}$. We denote the sets of such meta-paths as $\Phi_{\mathcal{V}}$ and $\Phi_{\mathcal{E}}$ respectively. Furthermore, let $\mathcal{N}_{u_v}^{\Phi_{\mathcal{V}}}$ denote the neighbors of node $u_v \in \mathbb{V}$ through paths $\gamma_v \in \Phi_{\mathcal{V}}$, and vice-versa let $\mathcal{N}_{u_e}^{\Phi_{\mathcal{E}}}$ denote the neighbors of node $u_e \in \mathbb{E}$ through paths $\gamma_e \in \Phi_{\mathcal{V}}$.

At each step, the model updates separately and sequentially the node features of nodes in $\mathbb{V}$ and $\mathbb{E}$. Consider for example the case of nodes in $\mathbb{V}$ (for nodes in $\mathbb{E}$ the process is the same, except that $\Phi_{\mathcal{E}}$ is considered instead of $\Phi_{\mathcal{V}}$). The node-level update is performed as follows, for a certain $u \in \mathbb{V}$:

$$\hat{\boldsymbol{x}}_u^{(t)} = \boldsymbol{W}_{\Phi_{\mathcal{V}}}^{(t)} \boldsymbol{x}_u^{(t)}, \tag{36}$$

$$\boldsymbol{x}_u^{(t+1)} = \sigma \left( \sum_{w \in \mathcal{N}_u^{\Phi_{\mathcal{V}}}} \alpha_{u,w}^{\Phi_{\mathcal{V}}} \hat{\boldsymbol{x}}_u^{(t)} \right). \tag{37}$$

In the equations above, $\boldsymbol{W}_{\Phi_{\mathcal{V}}}^{(t)}$ is a meta-path dependent weight matrix while $\alpha_{u,w}^{\Phi_{\mathcal{V}}}$ is an attention score computed between neighboring nodes in the same way as proposed in Veličković et al. (2017), through similar equations as 21 and 22. More generally, $h$ attention heads may be considered, that give rise to different attention scores for each head and consequently multiple results for the node feature update, that are then concatenated to obtain a unique feature vector $\boldsymbol{x}_u^{(t+1)}$.

**MLP.**  We also add the MLP model as a baseline; this model doesn't use connectivity at all and only relies on the initial node features to predict their class. The node feature matrix $\boldsymbol{X}$ is obtained as

$$\boldsymbol{X}^{(t)} = \text{MLP}(\boldsymbol{X}^{(t-1)}). \tag{38}$$

**MLP CB.**  This model employs a sampling procedure as outlined in Section 4.2, in which we straight-forwardly apply a Multilayer Perceptron to the initial features of nodes. During the training phase, we incorporate dropout by applying an MLP with distinct weights dropped out for each hyperedge, resulting in slightly different representations for nodes for each hyperedge they belong to. Furthermore, we execute the mini-batching procedure in accordance with the guidelines presented in Section 4.2. Importantly, that these two choices affect the training approach significantly so that the results of this model are very different from MLP's performances: see, for example, Table 1.

During the validation phase dropout is not utilized, ensuring that the representations used for each hyperedge remain exactly the same. Consequently, there is no need for the readout operation in this context. The node-level update is described by:

$$\boldsymbol{x}_{v,e}^{(t+1)} = \boldsymbol{x}_{v,e}^{(t)} + \text{MLP}(\text{LN}(\boldsymbol{x}_{v,e}^{(t)})), \tag{39}$$

$$\boldsymbol{x}_v^{(T)} = \frac{1}{d_v} \sum_{e \in \mathcal{E}_v} \boldsymbol{x}_{v,e}^{(T)}. \tag{40}$$

**Transformer.**  Along with MLP, we consider another simple baseline that is the basic Transformer model (Vaswani et al., 2017).

Also in this case, let $\mathbb{S}$ denote the set of input vectors, and define as $\boldsymbol{S} = \text{MLP}(\mathbb{S})$ the matrix of input embeddings, obtained from the input set through a MLP. The operations performed on $\boldsymbol{S}$ generalize the ones described for the Transformer module adopted in AllSetTransformer, and they can be split in two main modules, that are the self-attention module and the feed-forward module:

1. Suppose that $h$ attention heads are considered in the self attention module. First of all, $h$ triples of matrices $\boldsymbol{K}_i$ (keys), $\boldsymbol{V}_i$ (values) and $\boldsymbol{Q}_i$ (queries) with $i \in \{1, ..., h\}$ are obtained from $\boldsymbol{S}$ through linear matrix multiplications with weight matrices $\boldsymbol{W}_i^K, \boldsymbol{W}_i^V$ and $\boldsymbol{W}_i^Q$ that are learned during training. The result for each attention module is computed through the key, query and key matrices using an activation function $\omega$ (Vaswani et al., 2017) and a normalization factor $d_k$, that corresponds to the dimension of the key and query vectors associated to each input element. The $h$ outputs of the different attention heads are then concatenated, leading to a unique result matrix. After that, a sum operation and a Layer Normalization (LN) (Ba et al., 2016) are applied:

$$\boldsymbol{K}_i = \boldsymbol{S}\boldsymbol{W}_i^K, \boldsymbol{V}_i = \boldsymbol{S}\boldsymbol{W}_i^V, \boldsymbol{Q}_i = \boldsymbol{S}\boldsymbol{W}_i^Q, \quad \text{where} \quad i \in \{1, ..., h\}, \tag{41}$$

$$\boldsymbol{O}_i = \omega\left(\frac{\boldsymbol{Q}_i(\boldsymbol{K}_i)^T}{\sqrt{d_k}}\right)\boldsymbol{V}_i, \quad \text{where} \quad i \in \{1, ..., h\}, \tag{42}$$

$$\text{MH}_{h,\omega}(\boldsymbol{Q}, \boldsymbol{K}, \boldsymbol{V}) = \|_{i=1}^{h}\boldsymbol{O}^{(i)}, \tag{43}$$

$$\boldsymbol{Y} = \text{LN}(\boldsymbol{S} + \text{MH}_{h,\omega}(\boldsymbol{S}, \boldsymbol{S})). \tag{44}$$

Since it will be useful in the following we introduce, we denote by MAB the output of the multihead self-attention block, namely

$$\text{MAB}(\boldsymbol{S}) = \text{LN}(\boldsymbol{S} + \text{MH}_{h,\omega}(\boldsymbol{S}, \boldsymbol{S})). \tag{45}$$

2. As described for AllSetTransformer (Chien et al., 2022), in the feed-forward module a MLP is applied to the feature matrix, followed by a sum operation and Layer Normalization. After that, the output of the overall Transformer architecture is obtained:

$$\boldsymbol{Y}_{out} = \text{LN}(\boldsymbol{Y} + \text{MLP}(\boldsymbol{Y})). \tag{46}$$

**WHATsNet** In Choe et al. (2023), authors introduce a classification of edge-dependent node labels. In the following section, we will describe their method and define all the components. The author introduced WithinATT, inspired by the Transformer model Vaswani et al. (2017), which adjusts a node embedding by focusing on interactions with other nodes within the same hyperedge. Specifically, it applies a set of node embeddings as queries, keys, and values in the attention mechanism. This approach effectively captures relationships between nodes by computing the dot-product for each node pair within the hyperedge. However, calculating the dot-product for all pairs results in a computational complexity that is quadratic relative to the hyperedge size, posing challenges for large-scale, real-world hypergraphs. To address this, they utilize the inducing point method from SetTransformer Lee et al. (2019), which achieves performance comparable to all-pair attention but with significantly greater efficiency. More precisely, Lee et al. (2019) introduce a matrix of trainable inducing points $I$ of shape way smaller than the shape of $Q$ and $K$ the $\text{MAB}(Q, K)$ is approximated by $\text{MAB}(Q, \text{MAB}(I, K))$.

Let us now describe the WHATsNet model described in Choe et al. (2023), first we need to define the WithinATT module. Let $\mathbf{X}_e^{(t)}$ be the set of embeddings of nodes in a hyperedge $e$ in the $t$-th layer. Along with $\mathbf{X}_e^{(t)} \in \mathbb{R}^{|e| \times d}$, WithinATT uses number of trainable inducing points $\mathbf{I}_w \in \mathbb{R}^{k \times d}$, where $k$ is typically much smaller than $\max_{e \in \mathcal{E}} |e|$, in particula in their experiments they fix $k = 4$. Then, WithinATT is formally expressed as follows:

$$\text{WithinATT}(\boldsymbol{X}_e^{(t)}, \boldsymbol{I}_w) = \text{MAB}(\boldsymbol{X}_e^{(t)}, \text{MAB}(\boldsymbol{I}_w, \boldsymbol{X}_e^{(t)})). \tag{47}$$

where MAB is defined in 45). They also define WithinOrderPE, a positional encodings used in edge-dependent attention between nodes within each hyperedge. To introduce WithinOrderPE, the authors first define the order of each element $a$ in a set $A$ as follows:

$$Order(a, A) = \sum_{a' \in A} \mathbb{1}_{a' \leq a}. \tag{48}$$

Then, given node centralities $F \in \mathbb{R}^{N \times d_f}$, where $d_f$ represents the number of centrality measures, which corresponds to the dimensionality of the positional encodings, they define the WithinOrderPE of a node $v$ within a hyperedge $e$ as follows:

$$\text{WithinOrderPE}(v, e; F) = ||_{i=1}^{d_f} \frac{1}{|e|} Order(F_{v,i}, \{F_{u,i}\}_{u \in e}). \tag{49}$$

This WithinOrderPE is then added to the node embeddings, which are subsequently fed into WithinATT. Four centrality measures are used: degree, eigenvector centrality, PageRank, and coreness. The notation $X_{e,\boxplus}^{(t)}$ refers to the node embedding that incorporates the positional encodings from WithinOrderPE. To integrate these positional encodings, they first align the dimensionality of the positional encodings with that of the node features at layer $t$ by using a learnable weight matrix. We are now prepared to outline the hyperedge and node update embeddings as presented in Choe et al. (2023). Specifically, the a hyperedge $e$'s embedding $\boldsymbol{z}_e^{(t)}$ are obtained as

$$\boldsymbol{X}_{e,\boxplus}^{(t)} = \{\boldsymbol{x}_v^{(t)} \boxplus \text{WithinOrderPE}(v, e) : v \in e\}, \tag{50}$$

$$\tilde{\boldsymbol{X}}_{e,\boxplus}^{(t)} = \text{WithinATT}(\boldsymbol{X}_{e,\boxplus}^{(t)}) \tag{51}$$

$$\boldsymbol{z}_e^{(t)} = \text{MAB}(\boldsymbol{z}_e^{(t-1)}, \tilde{\boldsymbol{X}}_e^{(t)}) \tag{52}$$

Similarly, node embeddings are updated by aggregating node-dependent hyperedge embeddings from WithinATT and WithinOrderPE. Specifically, a node $v$ embedding $\boldsymbol{x}_v^{(t)}$ is updated as follows,

$$\boldsymbol{Z}_{v,\boxplus}^{(t)} = \{\boldsymbol{z}_e^{(t)} \boxplus \text{WithinOrderPE}(v, e) : e \in \mathcal{E}_v\}, \tag{53}$$

$$\tilde{\boldsymbol{Z}}_{v,\boxplus}^{(t)} = \text{WithinATT}(\boldsymbol{Z}_{v,\boxplus}^{(t)}) \tag{54}$$

$$\boldsymbol{x}_v^{(t)} = \text{MAB}(\boldsymbol{x}_v^{(t-1)}, \tilde{\boldsymbol{Z}}_v^{(t)}) \tag{55}$$

## C   Proof of Propositions

### C.1   Proof of Proposition 4.1

UniGCNII inherits the same hyperedge update rule of other hypergraph models (e.g. HNN (Feng et al., 2019), HyperGCN (Yadati et al., 2019)), so it directly follows from Theorem 3.4 of (Chien et al., 2022) that it can be expressed through 5. By looking at the definition of the node update rule of UniGCNII (Eq. 24 and 25), we can re-express it as

$$\boldsymbol{x}_v^{(t+1)} = ((1-\beta)\boldsymbol{I} + \beta\boldsymbol{W}^{(t)})\left((1-\alpha)\frac{1}{\sqrt{d_v}}\sum_{e\in\mathcal{E}_v}\frac{\boldsymbol{z}_e^{(t+1)}}{\sqrt{d_e}} + \alpha\boldsymbol{x}_v^{(0)}\right) = f_{\mathcal{E}\to\mathcal{V}}(\{\boldsymbol{z}_e^{(t+1)}\}_{e\in\mathcal{E}_v}; \boldsymbol{x}_v^{(0)}). \tag{56}$$

Note that this is a particular instance of the extended AllSet node update rule 7 where only a residual connection to the initial node features is considered. Lastly, it is straightforward to see that $f_{\mathcal{E}\to\mathcal{V}}$ is permutation invariant w.r.t the set $\{\boldsymbol{z}_e^{(t+1)}\}_{e\in\mathcal{E}_v}$, as it processes the set through a weighted mean.   □

### C.2   Proof of Proposition 4.2

We prove this proposition by showing that we can obtain AllSet update rules 5-6 and 7 from our proposed MultiSet framework 8-9-10. This can easily follow by not distinguishing node representations among hyperedges, so $\mathbb{X}_v^{(t)} = \{\boldsymbol{x}_{v,e}^{(t)}\}_{e\in\mathcal{E}_v} = x_v^{(t)}$. With this particular choice, we directly get 5 from 8, and 7 can be obtained from 9 by further disregarding the hyperedge subscript –as there is only a single node representation to update. Analogously, we can get 6 from 9 if we additionally do not consider node residual connections, so $\{\mathbb{X}_v^{(k)}\}_{k=0}^t$ simply becomes $x_v^{(t)}$. Finally, the readout 10 can be defined as the identity function applied to the node representations at the last message passing step $T$.   □

### C.3   Proof of Proposition 4.3

We also prove this proposition by showing that we can obtain EDHNN (Wang et al., 2023) update rules from our proposed MultiSet framework 8-9-10. EDHNN hyperedge update rule can be expressed as

$$z_e^{(t+1)} = \sum_{u\in e}\hat{\phi}(x_u^{(t)}) = f_{\mathcal{V}\to\mathcal{E}}(\{x_u^{(k)}\}_{u\in\mathcal{V}}),$$

which is a particular instance of 8 given that $\{x_u^{(k)}\}_{u\in\mathcal{V}}\subset\mathbb{X}_v^{(t)}$. As for the node update rule, we have

$$x_v^{(t+1)} = \hat{\psi}\left(x_v^{(t)}, \sum_{e\in\mathcal{E}_v}\hat{\rho}(x_v^{(t)}, z_e^{(t+1)}), x_v^0, d_v\right) = f_{\mathcal{E}\to\mathcal{V}}\left(\{\boldsymbol{z}_e^{(t+1)}\}_{e\in\mathcal{E}_v}; \{\boldsymbol{x}_v^{(k)}\}_{k\in\{0,t\}}\right), \tag{57}$$

where we recall that $d_v := |\mathcal{E}_v|$. By disregarding the hyperedge superscript in Eq. 9 –essentially making all hyperedge-based node representations the same one–, it is straightforward to see that the previous expression 57 is a particular case of the MultiSet node update rule. Lastly, the readout step 10 can be just defined as the identity function, given that all involved hyperedge-based messages to nodes (i.e. $\hat{\rho}(x_v^{(t)}, z_e^{(t+1)})$) are already being aggregated at each iteration while updating the node state.   □

### C.4   Proof of Proposition 4.4

The node updates in WHATsNet output a single node feature, while in MultiSet, the output of node updates results in multiple features for each node, i.e., one for each hyperedge to which it belongs. MultiSet framework preserves a higher level of generality by not enforcing pooling for edge-wise node representations at each layer, but a pooling operation could be intrinsically defined as part of the function in our Equation 9, leading to a single feature for each node as output.

We denote by $\boldsymbol{X}_e^{(t)}$ the set of the embeddings of nodes belonging to edge $e$. The functions are :

$$\boldsymbol{z}_e^{(t)} = \mathrm{MAB}(\boldsymbol{z}_e^{t-1}, \mathrm{WithinATT}(\{\boldsymbol{x}_v^{(t)} \boxplus \mathrm{WithinOrderPE}(v,e) : v\in e\})) = f_{\mathcal{V}\to\mathcal{E}}(\{\boldsymbol{x}_v^t\}_{v\in e}, z_e^{(t-1)}) \tag{58}$$

with WithinOrderPE defined as in equation 49. Similarly,

$$\boldsymbol{x}_v^{(t)} = \mathrm{MAB}(\boldsymbol{x}_v^{t-1}, \mathrm{WithinATT}(\{\boldsymbol{z}_e^{(t)} \boxplus \mathrm{WithinOrderPE}(v,e) : e \in \mathcal{E}_v\})) = f_{\mathcal{E} \to \mathcal{V}}(\{\boldsymbol{z}_e^t\}_{e \in \mathcal{E}_v}, x_v^{(t-1)}). \quad (59)$$

where MAB is defined according to Equation 45. In the work of Choe et al. (2023), the centrality measures used to compute the vectors in WithinOrderPE$(v, e)$ rely solely on the topology of the graph. However, centrality measures based on node and edge features could also be considered, and our framework is capable of capturing these as well.□

## C.5   Proof of Proposition 4.5

It is straightforward that functions $f_{\mathcal{V} \to \mathcal{E}}$, $f_{\mathcal{E} \to \mathcal{V}}$ and $f_{\mathcal{V} \to \mathcal{V}}$ defined in MultiSetMixer (Equations 11-13) are permutation invariant w.r.t their first argument: hyperedge update rule 11 and readout 13 process it through a mean operation, whereas the node update rule only receives a single-element set. The rest of the proof follows from the proof of Proposition 4.1 of Chien et al. (2022). □

## D   Experiments

### D.1   Hyperparameters optimization

In order to implement the benchmark models, we followed the procedure described in Chien et al. (2022); in particular, the maximum epochs were set to 200 for all the models. The models were trained with categorical cross-entropy loss, and the best architecture was selected at the end of training depending on validation accuracy. For the AllDeepSets (Chien et al., 2022), AllSetTransformer (Chien et al., 2022), UniGCNII (Huang & Yang, 2021), CEGAT, CEGCN, HCHA (Bai et al., 2021), HNN (Feng et al., 2019), HNHN (Dong et al., 2020), HyperGCN (Yadati et al., 2019), HAN(Wang et al., 2019), and HAN (mini-batching) (Wang et al., 2019) and MLP, we performed the same hyperparameter optimization proposed in Chien et al. (2022). For both the proposed model and the introduced baseline, we conducted a thorough hyperparameter search across the following values:

- learning rate within the range of $0.001, 0.01$;

- weight decay values from the set $0.0, 1e-5, 1$;

- MLP hidden layer sizes of $64, 128, 256, 512$;

- mini-batch sizes set to $256, 512$, with full-batch utilization when memory resources allow;

- the number of sampled neighbors per hyperedge ranged from $2, 3, 5, 10$.

It's important to note that the limitation of the number of sampled neighbors per hyperedge to this small range was intentional. This limitation showcases that even for datasets with large hyperedges, effective processing can be achieved by considering only a subset of neighbors.

The models' hyperparameters were optimized for a 50% split and subsequently applied to all the other splits.

**Reproducibility.**   We are committed to providing a comprehensive overview of our experimental setup, encompassing machine specifications, environmental details, and the optimal hyperparameters selected for each model. Additionally, the source code, including training/validation/test splits, will be supplied with both the initial release and the camera-ready version, ensuring the reproducibility of our results.

### D.2   Further information about the datasets

For our experiments we utilized various benchmark datasets from existing literature on hypergraph neural networks, the statistical properties of which are in Table 7. For what concerns co-authorship networks (Cora-CA and DPBL-CA) and co-citation networks (Cora, Citeseer, and Pubmed), we relied on the datasets

provided in Yadati et al. (2019). Additionally, we employed the Princeton ModelNet40 (Wu et al., 2015) and the National Taiwan University (Chen et al., 2003) dataset introduced for 3D object classification. For these two datasets, we complied with what Feng et al. (2019) and Yang et al. (2020) proposed for the construction of the hypergraphs, using both MVCNN (Su et al., 2015) and GVCNN (Feng et al., 2019) features. Additionally, we tested our model on three datasets with categorical attributes, namely 20Newsgroups, Mushroom, and ZOO, obtained from the UCI Categorical Machine Learning Repository (Dua et al., 2017). In order to construct hypergraphs for these datasets, we followed the approach described in Yadati et al. (2019), where a hyperedge is defined for all data points sharing the same categorical feature value.

The 20Newsgroups dataset consists of 100-dimensional attributes, where each attribute corresponds to a TF-IDF value. Hyperedges are formed by grouping all data points that share the same value for a categorical feature, with each hyperedge assigned a uniform weight of 1. For our analysis, we utilized a preprocessed version of the 20Newsgroups dataset (Zhou et al., 2006), where data samples (nodes) are represented by binary occurrence values for 100 words across 16,242 articles. These articles are categorized into four topics, aligned with the highest-level groupings of the original 20Newsgroups dataset available at the UCI ML Repository (Mitchell, 1997). The topic-specific group sizes in the preprocessed dataset are $4,605$, $3,519$, $2,657$, and $5,461$, respectively. However, the exact filtering criteria used in Zhou et al. (2006) to reduce the sample count from $20,000$ (in the original 20Newsgroups dataset (Mitchell, 1997)) to $16,242$ or to determine the number of classes remain unclear. We hypothesize that the filtering process may have removed documents with insufficient content or relevance to the top 100 words. Furthermore, we observed that the original version of the dataset (from the UCI ML Repository) is balanced, with approximately $1,000$ samples per class. In contrast, the preprocessed dataset (Zhou et al., 2006) shows a skewed distribution with uneven group sizes of $4,605$, $3,519$, $2,657$, and $5,461$. Based on statistical analysis of the group distributions in the UCI dataset, we hypothesize that some original groups (e.g., *alt*, *comp*, *misc*, *rec*, *sci*, *soc*, *talk*) were merged during preprocessing. As a result, we were unable to reproduce the dataset from scratch and relied on the preprocessed version for our experiments. While we acknowledge that the preprocessed version has certain limitations, such as its restricted 100-term vocabulary, the construction of hyperedges directly from node features, and challenges with reproducibility, it nonetheless provides an illustrative case for studying the limitations of HNN models. Despite its artificial nature, the dataset exhibits interesting characteristics, such as large hyperedges, which make it valuable for highlighting these challenges.

Table 7: Statistics of hypergraph datasets: $|e|$ denotes the size of hyperedges while $d_v$ denotes the node degree.

| | Cora | Citeseer | Pubmed | CORACA | DBLP-CA | ZOO | 20Newsgroups | Mushroom | NTU2012 | ModelNet40 |
|---|---|---|---|---|---|---|---|---|---|---|
| $|\mathcal{E}|$ | 1579 | 1079 | 7963 | 1072 | 22363 | 43 | 100 | 298 | 2012 | 12311 |
| # classes | 7 | 6 | 3 | 7 | 6 | 7 | 4 | 2 | 67 | 40 |
| min $|e|$ | 2 | 2 | 2 | 2 | 2 | 1 | 29 | 1 | 5 | 5 |
| med $|e|$ | 3 | 2 | 3 | 3 | 3 | 40 | 537 | 72 | 5 | 5 |
| max $d_v$ | 145 | 88 | 99 | 23 | 18 | 17 | 44 | 5 | 19 | 30 |
| min $d_v$ | 0 | 0 | 0 | 0 | 1 | 17 | 1 | 5 | 1 | 1 |
| avg $d_v$ | 1.77 | 1.04 | 1.76 | 1.69 | 2.41 | 17 | 4.03 | 5 | 5 | 5 |
| med $d_v$ | 1 | 0 | 0 | 2 | 2 | 17 | 3 | 5 | 5 | 4 |
| CE Homophily | 89.74 | 89.32 | 95.24 | 80.26 | 86.88 | 82.88 | 75.25 | 85.33 | 46.07 | 24.07 |
| $\frac{1}{|\mathcal{V}|}\sum_{v\in\mathcal{V}} h_v^0$ | 84.10 | 78.25 | 82.05 | 80.81 | 88.86 | 91.13 | 81.26 | 88.05 | 53.24 | 42.16 |
| $\frac{1}{|\mathcal{V}|}\sum_{v\in\mathcal{V}} h_v^1$ | 78.08 | 74.18 | 75.73 | 76.51 | 86.01 | 85.79 | 74.78 | 84.41 | 41.95 | 29.42 |

We downloaded co-citation and co-authoring networks from Yadati et al. (2019). Below are the details on how the hypergraph was constructed. **Cora-CA**, **DBLP**: all documents co-authored by an author are in one hyperedge, following what was done in Yadati et al. (2019). Co-citation data **Citeseer**, **Pubmed**, **Cora**: all documents cited by a document are connected by a hyperedge. Each hypernode (document abstract) is represented by bag-of-words features (feature matrix $\boldsymbol{X}$).

**Citation and co-authorship datasets.** In the co-citation and co-authorship networks datasets, the node features are the bag-of-words representations of the corresponding documents. In co-citation datasets (Cora,

Citeseer, PubMed) all documents cited by a document are connected by a hyperedge. In co-authored datasets (CORA-CA, DBLP-CA), all documents co-authored by an author belong to the same hyperedge.

**Computer vision/graphics.** The hyperedges are constructed using the $k$-nearest neighbor algorithm in which $k = 5$.

**Categorical datasets.** There are instances with categorical attributes within the datasets. To construct the hypergraph, each attribute value is treated as a hyperedge, meaning that all instances (hypernodes) with the same attribute value are contained in a hyperedge. The node features of 20Newsgroups are the TF-IDF representations of news messages. The node features of mushrooms (in Mushroom dataset) represent categorical descriptions of 23 species. The node features of a zoo (in ZOO dataset) are a combination of categorical and numeric measurements describing various animals.

Table 8: Node Connectivity Statistics. For brevity we use the following notation in this table: under the columns labeled $|\mathcal{E}_v| = k$, we report the amount of nodes that belong to $k$ hyperedges. This value can be expressed in a more formal way as $|v \in \mathcal{V} : |\mathcal{E}_v| = k|$. Moreover, $|\mathcal{E}_v| = 0$, denotes the number of isolated nodes. In addition, the columns labeled "$\% \ |\mathcal{E}_v| = k$" indicate the percentage of nodes belonging to $k$ hyperedges relatively to the total number of nodes. Finally, $\sum_{e \in \mathcal{E}} |e|$ corresponds to the number of hyperedge-dependent node representations.

| | $|\mathcal{V}|$ | $|\mathcal{E}_v| = 0$ | $|\mathcal{E}_v| = 1$ | $|\{v : |\mathcal{E}_v| = 2\}|$ | $|\mathcal{E}_v| = 3$ | $|\mathcal{E}_v| > 3$ | $\% \ |\mathcal{E}_v| = 0$ | $\% \ |\mathcal{E}_v| = 1$ | $\% \ |\mathcal{E}_v| = 2$ | $\% \ |\mathcal{E}_v| = 31$ | $\% \ |\mathcal{E}_v| > 3$ | $\sum_{e \in \mathcal{E}} |e|$ |
|---|---|---|---|---|---|---|---|---|---|---|---|---|
| Cora | 2708 | 1274 | 575 | 327 | 156 | 376 | 47.05 | 21.23 | 12.08 | 5.76 | 13.88 | 6060 |
| Citeseer | 3312 | 1854 | 798 | 307 | 144 | 209 | 55.98 | 24.09 | 9.27 | 4.35 | 6.31 | 5307 |
| Pubmed | 19717 | 15877 | 339 | 313 | 292 | 2896 | 80.52 | 1.72 | 1.59 | 1.48 | 14.69 | 50506 |
| CORA-CA | 2708 | 320 | 995 | 951 | 287 | 155 | 11.82 | 36.74 | 35.12 | 10.60 | 5.72 | 4905 |
| DBLP-CA | 41302 | 0 | 8998 | 16724 | 9249 | 6331 | 0.00 | 21.79 | 40.49 | 22.39 | 15.33 | 99561 |
| Mushroom | 8124 | 0 | 0 | 0 | 0 | 8124 | 0.00 | 0.00 | 0.00 | 0.00 | 100.00 | 40620 |
| NTU2012 | 2012 | 0 | 173 | 256 | 296 | 1287 | 0.00 | 8.60 | 12.72 | 14.71 | 63.97 | 10060 |
| ModelNet40 | 12311 | 0 | 1491 | 1755 | 1594 | 7471 | 0.00 | 12.11 | 14.26 | 12.95 | 60.69 | 61555 |
| 20newsW100 | 16242 | 0 | 3053 | 3149 | 2720 | 7320 | 0.00 | 18.80 | 19.39 | 16.75 | 45.07 | 65451 |
| ZOO | 101 | 0 | 0 | 0 | 0 | 101 | 0.00 | 0.00 | 0.00 | 0.00 | 100.00 | 1717 |

## E  Experiment results

### E.1  Additional experiments with heterophilic datasets

In this section, we broaden our experimental scope by including a set of datasets that were previously used in Wang et al. (2023). These datasets, namely Senate, Congress, House, and Walmart, are classified as heterophilic based on the CE homophily measure. Notably, they present an interesting challenge as they do not have inherent node features. Therefore, generating artificial node attributes is necessary before applying hypergraph models. Due to this constraint, we have decided to postpone the exploration of these datasets to future work, as we acknowledge the significance of addressing such limitations for a more thorough analysis. Table 9 displays the performance results of MultiSetMixer, EDHNN, and AllSet-like architectures on the mentioned datasets. Notably, EDHNN consistently outperforms other models across all datasets, demonstrating superior results. MultiSetMixer ranks as the second-best model, deviating by one standard deviation from EDHNN on Senate and House datasets, and performing similarly to AllSetTransformer and AddDeepSets on Congress and Walmart.

Table 9: Additional hypergraph model performance benchmarks on heterophilic datasets (test accuracy in %). Results for AllDeepSets and AllSetTransformer are taken from Wang et al. (2023).

| | Senate | Congress | House | Walmart |
|---|---|---|---|---|
| **AllDeepSets** | $48.17 \pm 5.67$ | $91.80 \pm 1.53$ | $67.82 \pm 2.40$ | $64.55 \pm 0.33$ |
| **AllSetTransformer** | $51.83 \pm 5.22$ | $92.16 \pm 1.05$ | $69.33 \pm 2.20$ | $65.46 \pm 0.25$ |
| **EDHNN** | $\mathbf{64.79 \pm 5.14}$ | $\mathbf{95.00 \pm 0.99}$ | $\mathbf{72.45 \pm 2.28}$ | $\mathbf{66.91 \pm 0.41}$ |
| **MultiSetMixer** | $61.34 \pm 3.45$ | $92.13 \pm 1.30$ | $70.77 \pm 2.03$ | $64.23 \pm 0.41$ |

### E.2 Rewiring experiment

Figure 6 visualizes the impact of connectivity modifications across different architectures and datasets. We organize results in a matrix format where each column represents a distinct rewiring strategy, and each row corresponds to a dataset. Model variants are color-coded, with dashed lines showing their original performance as baselines. The HNN models show little deviation from the corresponding baselines, regardless of the rewiring strategy. In contrast, CEGCN demonstrates significant improvement, particularly with the trimming and random strategies.

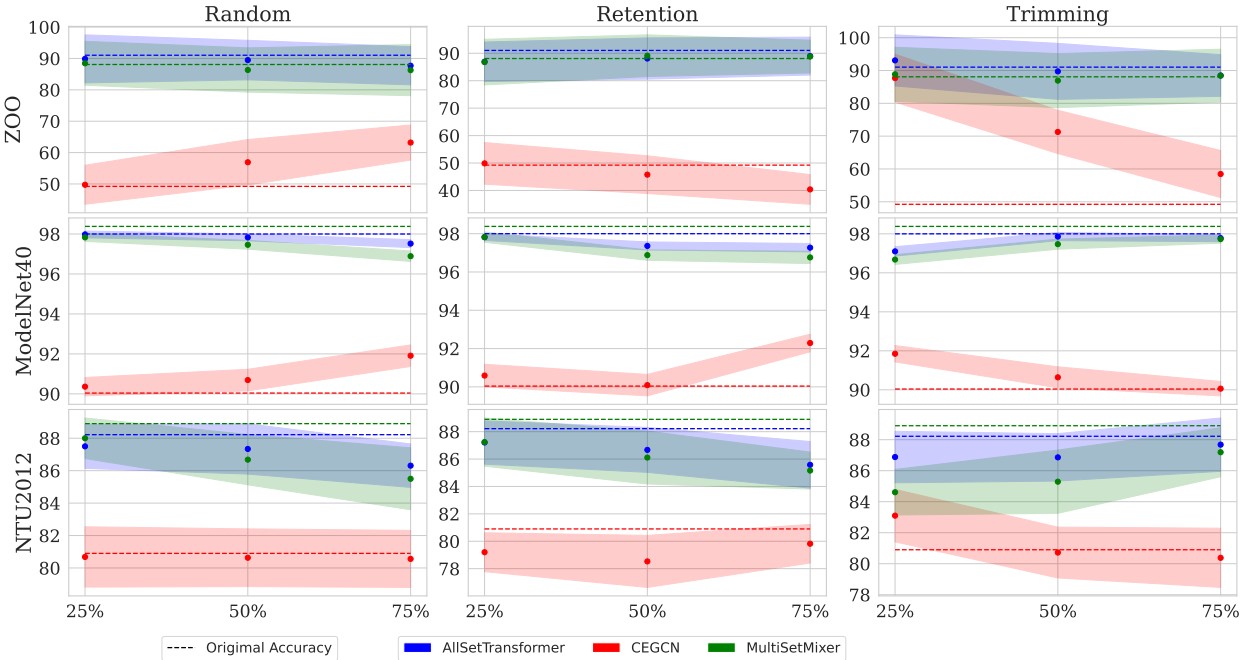

Figure 6: Visualization of model performance under different rewiring strategies. Columns show connectivity modifications, rows show datasets. Solid lines indicate modified architectures while dashed lines show baseline performance. Each color represents a different model variant.

### E.3 Analysis of $\mu$ parameter in $\Delta$ Homophily

This Section sheds more light on the impact of parameter $\mu$ in our proposed $\Delta$ homophily. To that end, we show in Figure 7 how $\Delta$-homophily evolves as $\mu$ varies across multiple datasets.

We first observe that $\Delta$-homophily generally increases with $\mu$. However, conceived as an homophily framework, the relevant aspect here is the relative homophily values across datasets –and not the specific absolute values. In this regard, we can see that the relative homophilic ranking is quite stable for different values of $\mu$, and in particular around the 0.1 value (our choice for the homophily analysis in Section 5.2).

### E.4 Benchmarking models across multiple training proportions splits

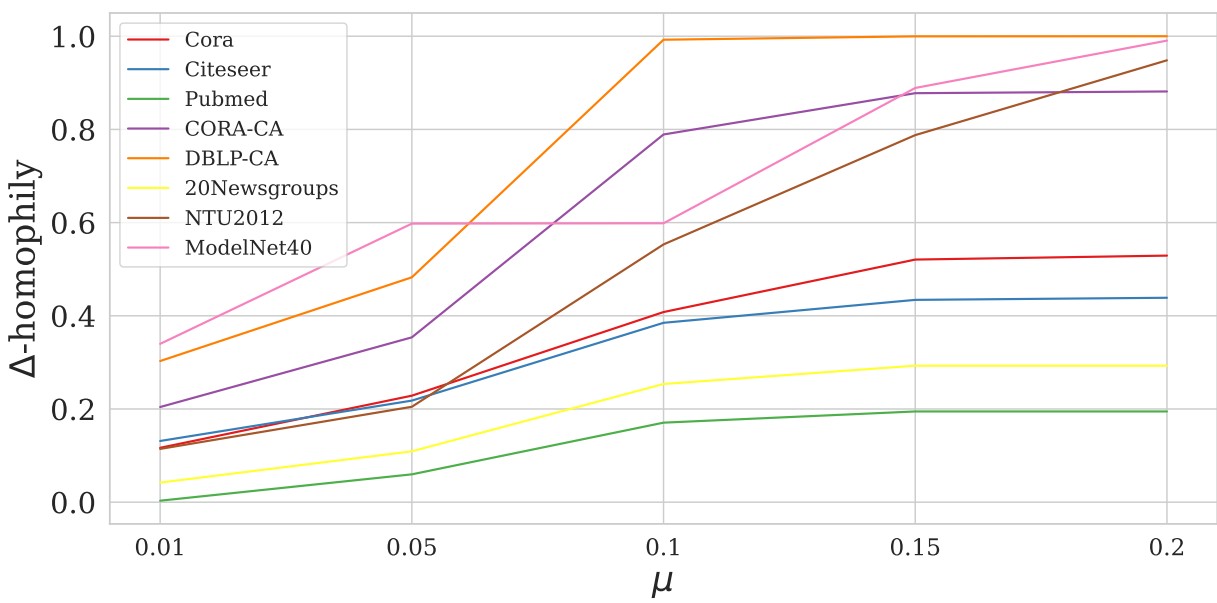

Figure 7: Δ-homophily scores across several models for different values of $\mu$ parameter (Eq. 4).

Table 10: Hypergraph model performance benchmarks. Test accuracy in % averaged over 15 splits. Training split: 50%.

| Model | Cora | Citeseer | Pubmed | CORA-CA | DBLP-CA | Mushroom | NTU2012 | ModelNet40 | 20Newsgroups | ZOO | avg. ranking |
|---|---|---|---|---|---|---|---|---|---|---|---|
| AllDeepSets | 77.11 ± 1.00 | 70.67 ± 1.42 | 89.04 ± 0.45 | 82.23 ± 1.46 | 91.34 ± 0.27 | 99.96 ± 0.05 | 86.49 ± 1.86 | 96.70 ± 0.25 | 81.19 ± 0.49 | 89.10 ± 7.00 | 6.80 |
| AllSetTransformer | 79.54 ± 1.02 | 72.52 ± 0.88 | 88.74 ± 0.51 | 84.43 ± 1.14 | 91.61 ± 0.19 | 99.95 ± 0.05 | 88.22 ± 1.42 | 98.00 ± 0.12 | 81.59 ± 0.59 | 91.03 ± 7.31 | 3.25 |
| UniGCNII | 78.46 ± 1.14 | 73.05 ± 1.48 | 88.07 ± 0.47 | 83.92 ± 1.02 | 91.56 ± 0.18 | 99.89 ± 0.07 | 88.24 ± 1.56 | 97.84 ± 0.16 | 81.16 ± 0.49 | 89.61 ± 8.09 | 4.75 |
| EDHNN | **80.74 ± 1.00** | 73.22 ± 1.14 | **89.12 ± 0.47** | **85.17 ± 1.02** | **91.94 ± 0.23** | 99.94 ± 0.11 | 88.04 ± 1.65 | 97.70 ± 0.19 | 81.64 ± 0.49 | 89.49 ± 6.99 | **2.90** |
| CEGAT | 76.53 ± 1.58 | 71.58 ± 1.11 | 87.11 ± 0.49 | 77.50 ± 1.51 | 88.74 ± 0.31 | 96.81 ± 1.41 | 82.27 ± 1.60 | 92.79 ± 0.44 | NA | 44.62 ± 9.18 | 12.11 |
| CEGCN | 77.03 ± 1.31 | 70.87 ± 1.19 | 87.01 ± 0.62 | 77.55 ± 1.65 | 88.12 ± 0.25 | 94.91 ± 0.44 | 80.90 ± 1.74 | 90.04 ± 0.47 | NA | 49.23 ± 6.81 | 12.56 |
| HCHA | 79.53 ± 1.33 | 72.57 ± 1.06 | 86.97 ± 0.55 | 83.53 ± 1.12 | 91.21 ± 0.28 | 98.94 ± 0.54 | 86.60 ± 1.96 | 94.50 ± 0.33 | 80.75 ± 0.53 | 89.23 ± 6.81 | 7.85 |
| HGNN | 79.53 ± 1.33 | 72.24 ± 1.08 | 86.97 ± 0.55 | 83.45 ± 1.22 | 91.26 ± 0.26 | 98.94 ± 0.54 | 86.71 ± 1.48 | 94.50 ± 0.33 | 80.75 ± 0.52 | 89.23 ± 6.81 | 7.95 |
| HNHN | 77.68 ± 1.08 | 73.47 ± 1.36 | 87.88 ± 0.47 | 78.53 ± 1.15 | 86.73 ± 0.40 | **99.97 ± 0.04** | 88.28 ± 1.50 | 97.84 ± 0.15 | 81.53 ± 0.55 | 89.23 ± 7.85 | 5.55 |
| HyperGCN | 74.78 ± 1.11 | 66.06 ± 1.58 | 82.32 ± 0.62 | 77.48 ± 1.14 | 86.07 ± 3.32 | 69.51 ± 4.98 | 47.65 ± 5.01 | 46.10 ± 7.95 | 80.84 ± 0.49 | 51.54 ± 9.88 | 14.30 |
| HAN | 80.73 ± 1.37 | **73.69 ± 0.95** | 86.34 ± 0.61 | 84.19 ± 0.81 | 91.10 ± 0.20 | 91.33 ± 0.91 | 83.78 ± 1.75 | 93.85 ± 0.33 | 79.67 ± 0.55 | 80.26 ± 6.42 | 8.90 |
| HAN minibatch | 80.24 ± 2.17 | 73.55 ± 1.13 | 85.41 ± 2.32 | 82.04 ± 2.56 | 90.52 ± 0.50 | 93.87 ± 1.04 | 80.62 ± 2.00 | 92.06 ± 0.63 | 79.76 ± 0.56 | 70.39 ± 11.29 | 10.60 |
| MultiSetMixer | 78.06 ± 1.24 | 71.85 ± 1.50 | 87.19 ± 0.53 | 82.74 ± 1.23 | 90.68 ± 0.19 | 99.58 ± 0.16 | **88.90 ± 1.30** | **98.38 ± 0.21** | **88.57 ± 1.96** | 88.08 ± 8.04 | 6.20 |
| MLP CB | 74.06 ± 1.26 | 71.93 ± 1.53 | 85.83 ± 0.51 | 74.39 ± 1.40 | 84.91 ± 0.44 | 99.93 ± 0.08 | 85.43 ± 1.51 | 96.41 ± 0.32 | 86.13 ± 2.82 | 81.61 ± 10.98 | 10.30 |
| MLP | 73.27 ± 1.09 | 72.07 ± 1.65 | 87.13 ± 0.49 | 73.27 ± 1.09 | 84.77 ± 0.41 | 99.91 ± 0.08 | 79.70 ± 1.56 | 95.31 ± 0.28 | 80.93 ± 0.59 | 85.13 ± 6.90 | 11.50 |
| Transformer | 74.15 ± 1.17 | 71.82 ± 1.51 | 87.37 ± 0.49 | 73.61 ± 1.55 | 85.26 ± 0.38 | 99.95 ± 0.08 | 82.88 ± 1.93 | 96.29 ± 0.29 | 81.17 ± 0.54 | 88.72 ± 10.25 | 9.85 |

Table 11: Hypergraph model performance benchmarks. Test accuracy in % averaged over 15 splits. Training split: 40%.

| Model | Cora | Citeseer | Pubmed | CORA-CA | DBLP-CA | Mushroom | NTU2012 | ModelNet40 | 20Newsgroups | ZOO | avg. ranking |
|---|---|---|---|---|---|---|---|---|---|---|---|
| AllDeepSets | 76.09 ± 1.22 | 70.32 ± 1.39 | 88.58 ± 0.46 | 81.32 ± 1.27 | 90.96 ± 0.24 | 99.94 ± 0.08 | 85.60 ± 1.46 | 96.71 ± 0.21 | 81.11 ± 0.43 | 89.57 ± 5.91 | 7.20 |
| AllSetTransformer | 78.81 ± 0.99 | 71.65 ± 1.05 | 88.17 ± 0.45 | 83.26 ± 1.12 | 91.26 ± 0.24 | 99.94 ± 0.09 | 87.04 ± 1.07 | 97.92 ± 0.14 | 81.30 ± 0.41 | **91.72 ± 6.38** | 3.75 |
| UniGCNII | 77.78 ± 1.15 | 72.30 ± 1.45 | 87.86 ± 0.37 | 83.39 ± 0.95 | 91.32 ± 0.19 | 99.88 ± 0.09 | 87.30 ± 1.34 | 97.86 ± 0.16 | 81.14 ± 0.45 | 89.68 ± 6.42 | 4.45 |
| EDHNN | **80.46 ± 0.91** | 72.62 ± 0.98 | **88.70 ± 0.34** | **84.41 ± 0.87** | **91.60 ± 0.20** | 99.95 ± 0.09 | 87.33 ± 1.20 | 97.65 ± 0.20 | 81.41 ± 0.37 | 90.75 ± 5.54 | **2.00** |
| CEGAT | 75.68 ± 1.09 | 70.59 ± 0.89 | 86.39 ± 0.47 | 76.91 ± 1.22 | 88.18 ± 0.31 | 96.72 ± 1.50 | 80.97 ± 1.30 | 92.46 ± 0.29 | NA | 45.27 ± 9.41 | 11.67 |
| CEGCN | 76.19 ± 1.06 | 70.08 ± 1.26 | 86.22 ± 0.50 | 76.17 ± 1.44 | 87.61 ± 0.26 | 95.00 ± 0.38 | 79.41 ± 1.26 | 89.79 ± 0.39 | NA | 51.40 ± 7.24 | 12.72 |
| HCHA | 78.87 ± 1.04 | 71.73 ± 0.91 | 86.28 ± 0.43 | 83.05 ± 0.99 | 91.04 ± 0.23 | 99.00 ± 0.48 | 85.53 ± 1.43 | 94.53 ± 0.28 | 80.77 ± 0.31 | 90.54 ± 5.29 | 7.40 |
| HGNN | 78.87 ± 1.04 | 71.44 ± 1.00 | 86.28 ± 0.43 | 82.95 ± 1.06 | 91.06 ± 0.24 | 99.00 ± 0.48 | 85.71 ± 1.37 | 94.53 ± 0.28 | 80.77 ± 0.31 | 90.54 ± 5.29 | 7.40 |
| HNHN | 76.47 ± 0.90 | 72.25 ± 1.10 | 87.17 ± 0.45 | 77.27 ± 1.11 | 86.61 ± 0.31 | **99.96 ± 0.08** | 87.14 ± 1.23 | 97.82 ± 0.17 | 81.28 ± 0.49 | 89.46 ± 6.50 | 6.10 |
| HyperGCN | 73.56 ± 0.91 | 64.65 ± 1.28 | 82.09 ± 0.67 | 76.44 ± 1.06 | 86.22 ± 2.93 | 69.49 ± 5.02 | 46.78 ± 4.61 | 45.34 ± 7.34 | 80.82 ± 0.59 | 52.80 ± 8.90 | 14.00 |
| HAN | 79.89 ± 0.78 | **73.16 ± 1.04** | 86.11 ± 0.56 | 83.84 ± 0.91 | 90.96 ± 0.19 | 91.39 ± 0.93 | 82.79 ± 1.16 | 93.83 ± 0.27 | 79.52 ± 0.47 | 80.11 ± 6.46 | 8.75 |
| HAN minibatch | 80.07 ± 1.68 | 69.44 ± 6.58 | 86.08 ± 0.84 | 82.33 ± 1.91 | NA | 93.60 ± 0.91 | 79.41 ± 1.62 | NA | 79.43 ± 0.91 | 64.84 ± 12.59 | 11.56 |
| MultiSetMixer | 77.14 ± 1.25 | 70.96 ± 1.66 | 86.67 ± 0.42 | 82.09 ± 0.79 | 90.45 ± 0.25 | 99.61 ± 0.23 | **87.41 ± 1.10** | **98.31 ± 0.17** | **88.82 ± 1.56** | 88.71 ± 6.33 | 6.15 |
| MLP CB | 72.60 ± 1.44 | 71.08 ± 1.68 | 85.14 ± 0.47 | 72.63 ± 1.31 | 84.63 ± 0.31 | 99.91 ± 1.03 | 83.72 ± 1.40 | 96.25 ± 0.25 | 87.28 ± 3.50 | 81.42 ± 11.55 | 10.15 |
| MLP | 70.61 ± 7.44 | 70.96 ± 1.65 | 86.60 ± 0.40 | 70.70 ± 7.33 | 84.42 ± 0.28 | 99.91 ± 0.09 | 77.83 ± 1.63 | 95.24 ± 0.23 | 80.95 ± 0.54 | 85.38 ± 8.02 | 11.35 |
| Transformer | 72.65 ± 1.15 | 70.70 ± 1.50 | 86.79 ± 0.34 | 71.96 ± 1.03 | 84.97 ± 0.27 | 99.92 ± 0.09 | 80.69 ± 1.55 | 96.18 ± 0.24 | 80.95 ± 0.46 | 89.68 ± 7.31 | 9.80 |

Table 12: Hypergraph model performance benchmarks. Test accuracy in % averaged over 15 splits. Training split: 30%.

| Model | Cora | Citeseer | Pubmed | CORA-CA | DBLP-CA | Mushroom | NTU2012 | ModelNet40 | 20Newsgroups | ZOO | avg. ranking |
|---|---|---|---|---|---|---|---|---|---|---|---|
| AllDeepSets | 74.78 ± 1.02 | 69.10 ± 1.34 | 88.01 ± 0.39 | 79.69 ± 1.44 | 90.57 ± 0.20 | **99.95 ± 0.06** | 84.04 ± 1.39 | 96.49 ± 0.26 | 80.99 ± 0.41 | 87.04 ± 6.74 | 7.25 |
| AllSetTransformer | 77.67 ± 0.92 | 71.06 ± 1.00 | 87.62 ± 0.34 | 82.14 ± 0.96 | 91.10 ± 0.18 | 99.94 ± 0.09 | 85.73 ± 1.38 | 97.73 ± 0.21 | 81.05 ± 0.49 | 90.19 ± 6.18 | 4.05 |
| UniGCNII | 76.29 ± 1.05 | 71.38 ± 1.32 | 87.48 ± 0.35 | 81.93 ± 1.09 | 90.97 ± 0.20 | 99.88 ± 0.08 | 85.59 ± 1.60 | 97.89 ± 0.16 | 81.10 ± 0.44 | 88.33 ± 6.29 | 4.55 |
| EDHNN | **79.30 ± 1.03** | **71.85 ± 0.84** | **88.25 ± 0.38** | **83.13 ± 1.03** | **91.37 ± 0.18** | 99.94 ± 0.11 | **86.09 ± 1.44** | 97.55 ± 0.23 | 81.26 ± 0.34 | 89.44 ± 6.38 | **2.35** |
| CEGAT | 74.25 ± 1.24 | 69.75 ± 0.90 | 85.41 ± 0.44 | 75.24 ± 1.05 | 87.54 ± 0.23 | 96.86 ± 1.27 | 79.12 ± 1.60 | 91.89 ± 0.30 | NA | 42.87 ± 8.96 | 11.94 |
| CEGCN | 74.84 ± 1.35 | 69.17 ± 0.93 | 85.17 ± 0.41 | 74.83 ± 1.72 | 87.10 ± 0.25 | 95.08 ± 0.40 | 78.13 ± 1.35 | 89.34 ± 0.40 | NA | 48.70 ± 5.96 | 12.44 |
| HCHA | 77.81 ± 1.07 | 71.10 ± 1.11 | 84.97 ± 0.41 | 81.81 ± 1.08 | 90.85 ± 0.18 | 99.00 ± 0.47 | 84.34 ± 1.61 | 94.38 ± 0.28 | 80.78 ± 0.43 | **90.65 ± 5.58** | 7.40 |
| HGNN | 77.81 ± 1.07 | 70.88 ± 1.05 | 84.97 ± 0.41 | 81.78 ± 1.13 | 90.85 ± 0.16 | 99.00 ± 0.47 | 84.40 ± 1.38 | 94.38 ± 0.28 | 80.78 ± 0.43 | **90.65 ± 5.58** | 7.60 |
| HNHN | 74.85 ± 1.14 | 71.34 ± 1.03 | 86.34 ± 0.39 | 75.46 ± 1.02 | 86.33 ± 0.25 | **99.95 ± 0.07** | 84.93 ± 1.49 | 97.75 ± 0.20 | 81.10 ± 0.48 | 85.37 ± 7.96 | 6.30 |
| HyperGCN | 71.54 ± 1.26 | 63.82 ± 1.34 | 81.87 ± 0.56 | 74.44 ± 1.25 | 85.63 ± 2.89 | 69.44 ± 5.02 | 46.70 ± 4.01 | 45.28 ± 8.18 | 80.64 ± 0.48 | 55.46 ± 6.78 | 14.20 |
| HAN | 78.60 ± 1.28 | **72.44 ± 1.05** | 85.89 ± 0.44 | 82.69 ± 0.77 | 90.85 ± 0.19 | 91.47 ± 0.79 | 81.54 ± 1.44 | 93.79 ± 0.20 | 79.51 ± 0.58 | 79.81 ± 6.61 | 7.90 |
| HAN minibatch | 78.84 ± 1.19 | 72.26 ± 0.93 | 85.70 ± 0.81 | 79.81 ± 1.53 | NA | 93.59 ± 0.84 | 77.97 ± 1.63 | NA | 79.46 ± 1.10 | 45.74 ± 13.83 | 9.88 |
| MultiSetMixer | 76.03 ± 1.37 | 70.60 ± 1.15 | 86.12 ± 0.36 | 80.58 ± 1.11 | 90.19 ± 0.20 | 99.56 ± 0.16 | 85.95 ± 1.43 | **98.20 ± 0.17** | **88.20 ± 1.24** | 86.11 ± 6.96 | 5.90 |
| MLP CB | 71.14 ± 1.09 | 70.21 ± 1.14 | 84.24 ± 0.63 | 71.14 ± 1.61 | 84.17 ± 0.24 | 99.02 ± 0.08 | 81.18 ± 1.79 | 96.10 ± 0.22 | 85.94 ± 5.83 | 79.58 ± 8.22 | 10.40 |
| MLP | 66.14 ± 11.37 | 69.92 ± 1.32 | 85.86 ± 0.29 | 66.14 ± 11.37 | 83.96 ± 0.25 | 99.89 ± 0.11 | 75.08 ± 1.69 | 95.05 ± 0.31 | 80.82 ± 0.45 | 82.87 ± 7.37 | 11.70 |
| Transformer | 70.41 ± 1.13 | 69.75 ± 1.33 | 86.05 ± 0.28 | 69.93 ± 0.92 | 84.57 ± 0.25 | 99.93 ± 0.10 | 78.20 ± 1.73 | 95.92 ± 0.20 | 80.87 ± 0.37 | 86.85 ± 9.97 | 10.25 |

Table 13: Hypergraph model performance benchmarks. Test accuracy in % averaged over 15 splits. Training split: 20%.

| Model | Cora | Citeseer | Pubmed | CORA-CA | DBLP-CA | Mushroom | NTU2012 | ModelNet40 | 20Newsgroups | ZOO | avg. ranking |
|---|---|---|---|---|---|---|---|---|---|---|---|
| AllDeepSets | 72.56 ± 1.60 | 67.49 ± 1.57 | 87.24 ± 0.36 | 77.24 ± 1.52 | 89.91 ± 0.26 | **99.92 ± 0.07** | 80.71 ± 1.32 | 96.30 ± 0.24 | 80.59 ± 0.33 | 84.72 ± 7.95 | 7.05 |
| AllSetTransformer | 75.69 ± 1.09 | 69.39 ± 1.30 | 86.63 ± 0.40 | 80.54 ± 0.94 | 90.72 ± 0.17 | **99.92 ± 0.07** | 82.58 ± 1.31 | 97.48 ± 0.24 | 80.82 ± 0.28 | 85.61 ± 7.29 | 3.95 |
| UniGCNII | 74.11 ± 1.28 | 70.51 ± 1.48 | 86.97 ± 0.41 | 79.41 ± 1.23 | 90.47 ± 0.19 | 99.90 ± 0.06 | 82.54 ± 1.60 | 97.83 ± 0.15 | 80.88 ± 0.32 | 88.21 ± 5.55 | 4.20 |
| EDHNN | **77.09 ± 1.30** | 70.76 ± 1.39 | 87.52 ± 0.40 | 81.06 ± 1.27 | 90.91 ± 0.20 | 99.91 ± 0.10 | **83.45 ± 1.34** | 97.37 ± 0.17 | 80.92 ± 0.35 | 88.54 ± 6.38 | **2.10** |
| CEGAT | 71.86 ± 1.42 | 68.11 ± 1.34 | 84.03 ± 0.51 | 73.18 ± 1.32 | 86.98 ± 0.27 | 96.19 ± 1.38 | 76.14 ± 1.20 | 91.34 ± 0.31 | NA | 41.22 ± 6.16 | 11.67 |
| CEGCN | 73.25 ± 1.35 | 67.23 ± 1.39 | 83.47 ± 0.52 | 72.50 ± 1.44 | 86.29 ± 0.21 | 94.98 ± 0.31 | 75.55 ± 1.37 | 88.60 ± 0.41 | NA | 49.51 ± 6.31 | 12.44 |
| HCHA | 76.04 ± 1.30 | 69.90 ± 1.25 | 83.65 ± 0.37 | 80.03 ± 0.87 | 90.53 ± 0.17 | 99.03 ± 0.43 | 81.48 ± 1.29 | 94.31 ± 0.20 | 80.60 ± 0.29 | **89.35 ± 5.89** | 6.50 |
| HGNN | 76.04 ± 1.30 | 69.59 ± 1.22 | 83.65 ± 0.37 | 80.02 ± 0.88 | 90.51 ± 0.19 | 99.03 ± 0.43 | 81.60 ± 1.24 | 94.31 ± 0.20 | 80.60 ± 0.29 | **89.35 ± 5.89** | 6.60 |
| HNHN | 72.47 ± 1.06 | 69.44 ± 1.21 | 85.10 ± 0.47 | 73.16 ± 0.99 | 85.82 ± 0.21 | 99.88 ± 0.10 | 81.56 ± 1.54 | 97.61 ± 0.24 | 80.48 ± 0.32 | 82.60 ± 7.71 | 7.90 |
| HyperGCN | 68.59 ± 1.79 | 62.08 ± 1.28 | 81.57 ± 0.43 | 71.42 ± 1.27 | 85.45 ± 2.31 | 69.56 ± 8.16 | 44.01 ± 3.47 | 46.40 ± 8.41 | 80.38 ± 0.37 | 53.25 ± 8.74 | 14.10 |
| HAN | 76.73 ± 1.18 | **71.21 ± 1.13** | 85.72 ± 0.44 | 80.83 ± 0.89 | 90.56 ± 0.15 | 91.50 ± 0.98 | 79.46 ± 1.30 | 93.77 ± 0.23 | 79.33 ± 0.45 | 78.54 ± 6.50 | 7.45 |
| HAN minibatch | 76.89 ± 1.51 | NA | 85.59 ± 0.72 | 78.55 ± 1.43 | NA | 93.22 ± 1.34 | 73.79 ± 1.54 | NA | 79.50 ± 0.42 | 47.72 ± 14.96 | 10.14 |
| MultiSetMixer | 73.93 ± 1.15 | 68.95 ± 1.37 | 85.00 ± 0.62 | 78.32 ± 0.94 | 89.80 ± 0.19 | 99.42 ± 0.20 | 82.40 ± 1.41 | **98.01 ± 0.19** | **87.85 ± 1.53** | 81.06 ± 7.04 | 6.60 |
| MLP CB | 68.07 ± 1.50 | 68.64 ± 1.34 | 83.06 ± 0.58 | 68.37 ± 1.23 | 83.43 ± 0.25 | 99.84 ± 0.38 | 77.01 ± 1.69 | 95.85 ± 0.27 | 85.66 ± 5.70 | 75.61 ± 9.73 | 10.45 |
| MLP | 51.73 ± 17.51 | 68.20 ± 1.21 | 85.09 ± 0.34 | 51.73 ± 17.51 | 83.22 ± 0.21 | 99.84 ± 0.12 | 69.35 ± 9.49 | 94.69 ± 0.34 | 80.58 ± 0.31 | 78.54 ± 8.98 | 11.70 |
| Transformer | 67.34 ± 1.26 | 68.06 ± 1.39 | 85.31 ± 0.40 | 66.61 ± 1.50 | 83.93 ± 0.27 | 99.86 ± 0.13 | 74.17 ± 1.65 | 95.65 ± 0.29 | 80.51 ± 0.38 | 81.45 ± 6.81 | 10.70 |

Table 14: Hypergraph model performance benchmarks. Test accuracy in % averaged over 15 splits. Training split: 10%.

| Model | Cora | Citeseer | Pubmed | CORA-CA | DBLP-CA | Mushroom | NTU2012 | ModelNet40 | 20Newsgroups | ZOO | avg. ranking |
|---|---|---|---|---|---|---|---|---|---|---|---|
| AllDeepSets | 68.51 ± 1.64 | 64.50 ± 1.43 | 85.55 ± 0.38 | 73.67 ± 1.79 | 88.82 ± 0.25 | **99.88 ± 0.08** | 73.44 ± 1.91 | 95.96 ± 0.21 | 79.61 ± 0.36 | 76.81 ± 7.05 | 7.05 |
| AllSetTransformer | 71.82 ± 1.18 | 65.96 ± 1.48 | 84.71 ± 0.55 | 76.16 ± 1.36 | 90.09 ± 0.18 | 99.86 ± 0.09 | 75.78 ± 1.96 | 96.93 ± 0.21 | 80.18 ± 0.31 | 75.22 ± 10.78 | 4.70 |
| UniGCNII | 69.36 ± 1.63 | 66.41 ± 1.59 | 85.51 ± 0.50 | 75.84 ± 1.13 | 89.70 ± 0.25 | 99.86 ± 0.10 | 74.86 ± 2.20 | **97.58 ± 0.18** | 80.44 ± 0.26 | **79.86 ± 7.97** | 4.10 |
| EDHNN | **72.78 ± 1.54** | 67.90 ± 1.51 | **86.00 ± 0.51** | 77.20 ± 1.31 | **90.30 ± 0.17** | 99.86 ± 0.10 | **76.60 ± 1.79** | 96.91 ± 0.27 | 80.47 ± 0.29 | 79.06 ± 7.88 | **2.30** |
| CEGAT | 68.08 ± 1.65 | 64.15 ± 1.60 | 81.83 ± 0.43 | 69.04 ± 1.60 | 85.92 ± 0.23 | 96.01 ± 1.31 | 69.26 ± 2.27 | 90.17 ± 0.37 | NA | 39.20 ± 6.19 | 12.00 |
| CEGCN | 70.22 ± 1.62 | 62.68 ± 1.49 | 82.13 ± 0.44 | 67.45 ± 1.54 | 85.41 ± 0.26 | 94.85 ± 0.36 | 68.31 ± 2.08 | 87.28 ± 0.39 | NA | 49.20 ± 5.69 | 12.11 |
| HCHA | 72.76 ± 1.82 | 66.15 ± 1.27 | 82.41 ± 0.36 | 76.97 ± 0.95 | 90.00 ± 0.19 | 98.93 ± 0.41 | 74.44 ± 2.31 | 94.04 ± 0.21 | 80.23 ± 0.32 | 79.78 ± 7.89 | 5.95 |
| HGNN | 72.76 ± 1.82 | 65.69 ± 1.57 | 82.41 ± 0.36 | 76.96 ± 1.10 | 90.00 ± 0.18 | 99.00 ± 0.18 | 74.53 ± 2.44 | 94.04 ± 0.21 | 80.23 ± 0.32 | 79.78 ± 7.89 | 6.25 |
| HNHN | 67.43 ± 1.62 | 65.02 ± 1.40 | 82.33 ± 0.76 | 68.10 ± 1.67 | 84.74 ± 0.31 | 99.69 ± 0.18 | 73.82 ± 2.11 | 97.34 ± 0.25 | 80.00 ± 0.26 | 73.12 ± 6.57 | 8.80 |
| HyperGCN | 63.21 ± 1.95 | 57.81 ± 1.91 | 80.83 ± 0.46 | 65.58 ± 2.02 | 84.37 ± 1.73 | 67.56 ± 8.16 | 40.30 ± 3.67 | 45.92 ± 7.60 | 79.57 ± 0.38 | 51.96 ± 6.32 | 14.10 |
| HAN | 72.08 ± 1.47 | 67.86 ± 1.46 | 85.10 ± 0.43 | **77.48 ± 1.22** | 90.02 ± 0.17 | 91.67 ± 0.86 | 72.91 ± 1.88 | 93.52 ± 0.32 | 78.77 ± 0.49 | 70.94 ± 14.54 | 7.30 |
| HAN minibatch | 69.61 ± 6.86 | **68.25 ± 1.15** | 84.93 ± 0.65 | 76.27 ± 1.54 | NA | NA | 63.36 ± 2.66 | NA | NA | 43.62 ± 9.44 | 8.17 |
| MultiSetMixer | 69.69 ± 1.36 | 65.71 ± 1.46 | 83.01 ± 0.65 | 74.88 ± 1.10 | 89.11 ± 0.21 | 99.08 ± 0.33 | 74.82 ± 2.10 | 97.52 ± 0.20 | **86.92 ± 1.66** | 73.53 ± 7.58 | 6.10 |
| MLP CB | 62.42 ± 1.37 | 64.85 ± 1.30 | 81.43 ± 0.86 | 62.82 ± 1.80 | 82.02 ± 0.36 | 99.61 ± 0.18 | 68.80 ± 1.51 | 95.16 ± 0.26 | 85.21 ± 3.81 | 67.46 ± 9.14 | 10.70 |
| MLP | 38.64 ± 12.37 | 64.21 ± 1.53 | 83.56 ± 0.49 | 37.85 ± 11.79 | 81.88 ± 0.21 | 99.72 ± 0.15 | 63.39 ± 2.24 | 93.71 ± 0.38 | 79.63 ± 0.42 | 72.17 ± 9.21 | 11.60 |
| Transformer | 61.45 ± 1.66 | 63.75 ± 1.39 | 83.86 ± 0.50 | 60.65 ± 1.87 | 82.40 ± 0.47 | 99.74 ± 0.20 | 65.14 ± 1.61 | 94.66 ± 0.42 | 79.61 ± 0.39 | 68.82 ± 8.32 | 11.15 |

# F  Sampling analysis

As it has been discussed in Section 4.3, the proposed mini-batching procedure consists of two steps. At *step 1*, it samples $B$ hyperedges from $\mathcal{E}$. The hyperedge sampling over $\mathcal{E}$ can be either uniform or weighted (e.g. by taking into account hyperedge cardinalities). Then in *step 2* $L$ nodes are in turn sampled from each sampled hyperedge $e$, padding the hyperedge with $L - |e|$ special padding tokens if $|e| < L$ –consisting of **0** vectors that can be easily discarded in some computations. Overall, the shape of the obtained mini-batch $X$ has fixed size $B \times L$.

*Step 0* (hyperedge mini-batching) is particularly beneficial for large datasets; however, it can be skipped when the network fits fully into memory. Empirically, we found *step 1* (node mini-batching within a hyperedge) to be useful for two reasons: (i) pooling operations over a large set may over-squash the signal, and (ii) node batching leads to the training distribution shift, hence it can be useful to keep it even when the full hyperedge can be stored in memory.

When both *step 1* and *step 2* are employed, considering the hidden dimension size, the batch size required to be stored in memory during the forward pass is $B \times L \times d$, where $d$ represents the hidden dimension. If only *step 2* is employed, the batch size is $|\mathcal{E}| \times L \times d$, where $|\mathcal{E}|$ is the number of hyperedges within the hypernetwork. Finally, when no mini-batching steps are used, the batch size is $|\mathcal{E}| \times \max_{e \in \mathcal{E}} |e| \times d$, where $\max_{e \in \mathcal{E}} |e|$ is the size of the longest hyperedge.

**Theoretical analysis.** In this Section, we provide an analysis regarding the uniform sampling of the hyperedges in Step 1. We propose sampling $X$ mini-batches of a certain size $B$ at each iteration. At *step 1*, we sample $B$ hyperedges from $\mathcal{E}$; in *step 2*, for each hyperedge we sample a fixed number of nodes, that are randomly chosen among the ones belonging to that specific hyperedge. If the hyperedge does not contain enough samples, we use padding so that the size of the set of sampled nodes is increased to the desired value. By choosing to sample the nodes uniformly at random from the hyperedge, there is no guarantee that we will eventually sample all the nodes of each hyperedge. Indeed, sampling uniformly at random $c$ items from a set of size $n$, the probability of not sampling our desired one is $1 - \frac{c}{n}$. The probability of having to wait for $T$ independent trials before finding node $x$ among the sampled nodes is described by the geometric distribution.

Namely, let $x \in e$ and $|e| = n$, and assume the size of the considered mini-batch is $c$:

$$\mathbb{P}\left( \text{ Sample node } x \text{ from hyperedge } e \text{ for the first time at epoch } T\right) = \left(1 - \frac{c}{n}\right)^{T-1} \frac{c}{n}. \tag{60}$$

It follows that

$$\mathbb{E}\left[ \text{ \# of epochs to wait before sampling node } x\right] = \frac{n}{c}.$$

Assume now that a node $x$ belongs to $k$ hypergraphs $e_1, \ldots, e_k$ of respective sizes $n_1, \ldots, n_k$. The events {Node $x$ is sampled from hyperedge $e_i$ } and {Node $x$ is sampled from hyperedge $e_j$ } are independent if $i \neq j$. It follows that the random variable { # of epochs to wait until we sample node $x$ from all the hyperedges $e_1, \ldots, e_k$} is the maximum of $k$ independent non-identically distributed geometric distributions. Denote by $T_i$ the random variable that corresponds to the number of epochs we have to wait before sampling sample $x$ from edge $e_i$. The exact distribution for the random variable $T$, that is, the number of epochs we have to wait until we sample node $x$ from all hyperedges $e_1, \ldots e_k$ at least once, is

$$\mathbb{P}\left(T \leq h\right) = \mathbb{P}\left( \max_{i=1,\ldots,k} T_i \leq h \right) = \prod_{i=1}^{k} \mathbb{P}\left(T_i \leq h\right) = \prod_{i=1}^{k} \left(1 - p_i\right)^h$$

It follows that

$$\mathbb{E}\left( \text{ \# of epochs to wait until we sample node } x \text{ from all the hyperedges } e_1, \ldots, e_k\right) = \tag{61}$$

$$\mathbb{E}\left( \max_{i=1,\ldots,k} T_i \right) = \sum_{i=0}^{k} \left(1 - \prod_{i=1}^{h} \left(1 - \left(1 - p_i\right)^k\right)\right) \tag{62}$$

This can't be expressed in closed form: we can use the Moment Generating Function to bound the expected value of the maximum. Alternatively, we can also try to use the inequality due to Aven (1985), so that

$$\mathbb{E}\left( \max_{i=1,\ldots,k} T_i \right) \tag{63}$$

$$\leq \max_{i=1,\ldots k} \mathbb{E}\left(T_i\right) + \sqrt{\frac{k-1}{k} \sum_{i=1}^{k} \mathbb{V}\left(T_i\right)} = \max_{i=1,\ldots k} \frac{n}{c} + \sqrt{\frac{k-1}{k} k \left[\frac{n}{c}\left(1 - \frac{n}{c}\right)\right]} \tag{64}$$

$$= \frac{n}{c} + \sqrt{k - 1 \left[\frac{n}{c}\left(1 - \frac{n}{c}\right)\right]} \tag{65}$$

**Probability that a specific node is not sampled in one epoch.** Let $v$ be a node and let $d_v$ be its degree. In one epoch, we "see" all hyperedges but, of course, not necessarily all their nodes. It holds that

$$\mathbb{P}(\text{ node } v \text{ is sampled in epoch } T) = 1 - \mathbb{P}(\text{ node } v \text{ is not sampled in epoch } T) \qquad (66)$$

We can write the event

$$\{\text{ node } v \text{ is not sampled in epoch } T\} = \cap_{e \ s.t. v \in e} \{\text{ node } v \text{ is not sampled in } e\}.$$

It follows that

$$\mathbb{P}(\text{ node } v \text{ is sampled in epoch } T) = \max\left\{1 - \prod_{i=1}^{d_v} \frac{c}{|e_i| - 1}, \mathbb{1}_{\min_{i=1,\dots,d_v} |e_i| < c}\right\} \qquad (67)$$

Indeed, if any on the edges $v$ belongs to has a size smaller than the batch size for nodes ($c$), the node is for sure seen in the first epoch.

## G  Node Homophily

In this Section, we report the node homophily plots for the datasets not illustrated in Figure 1. For each dataset, we choose to illustrate 3 different levels of node homophily, respectively $0, 1$ and $10-$ level homophily, using Equation 2 at $t = 0, 1$ and $10$ (left, middle, and right plots respectively). Horizontal lines depict class mean homophily, with numbers above indicating the number of visualized points per class.

### G.1  Figure - node homophily

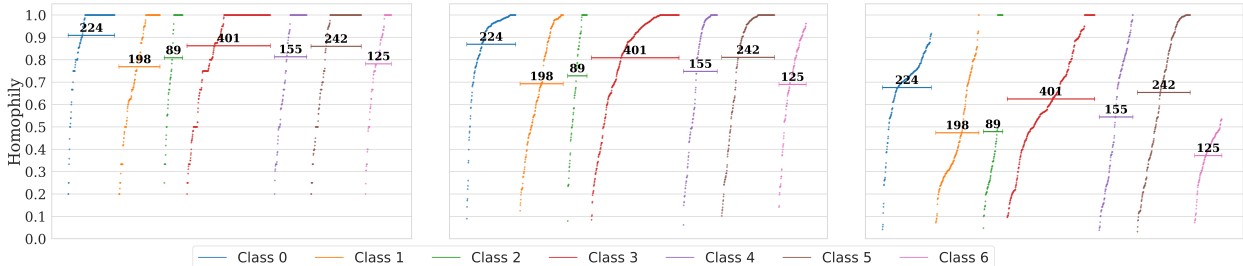

Figure 8: Node Homophily Distribution Scores for Cora.

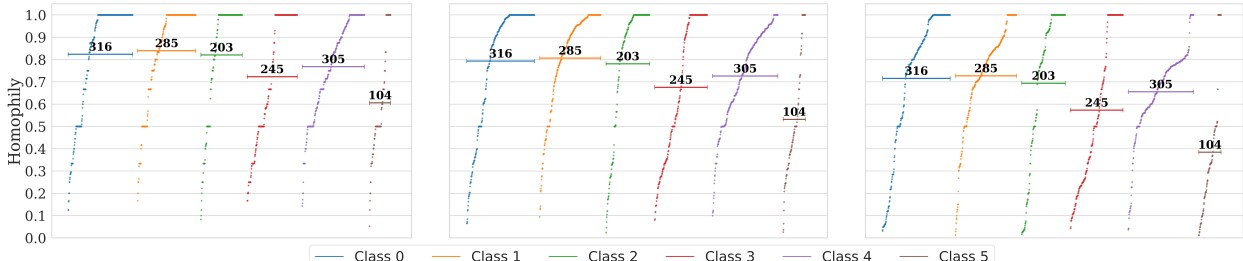

Figure 9: Node Homophily Distribution Scores for Citeseer.

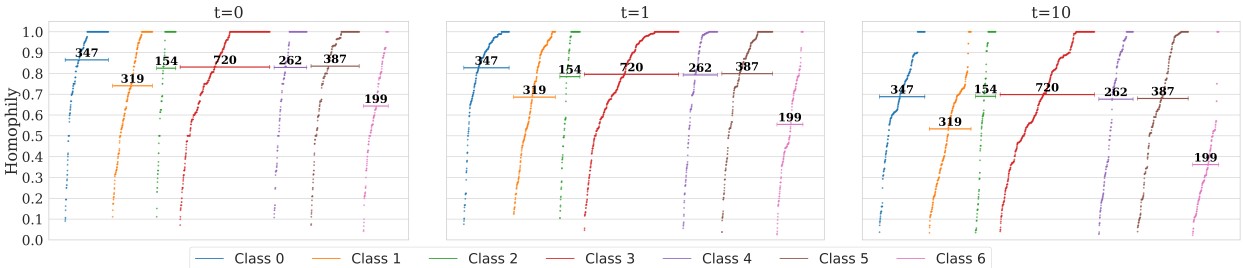

Figure 10: Node Homophily Distribution Scores for CORA-CA.

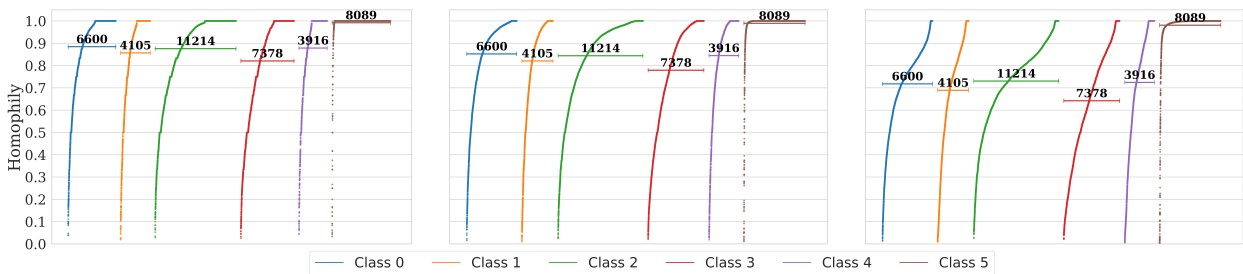

Figure 11: Node Homophily Distribution Scores for DBLP-CA.

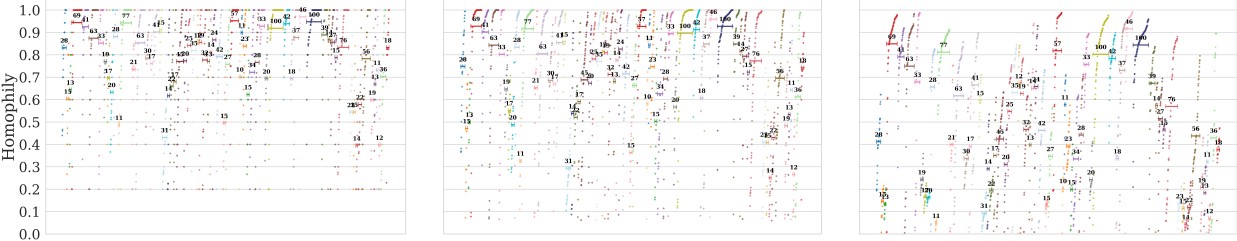

Figure 12: Node Homophily Distribution Scores for NTU2012.

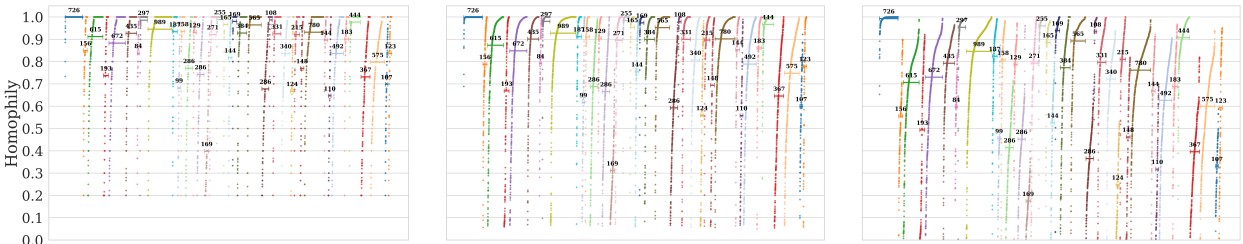

Figure 13: Node Homophily Distribution Scores for ModelNet40.

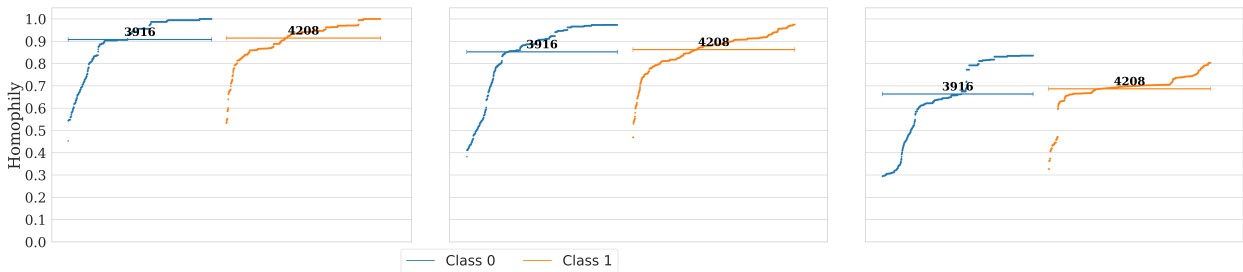

Figure 14: Node Homophily Distribution Scores for Mushroom.

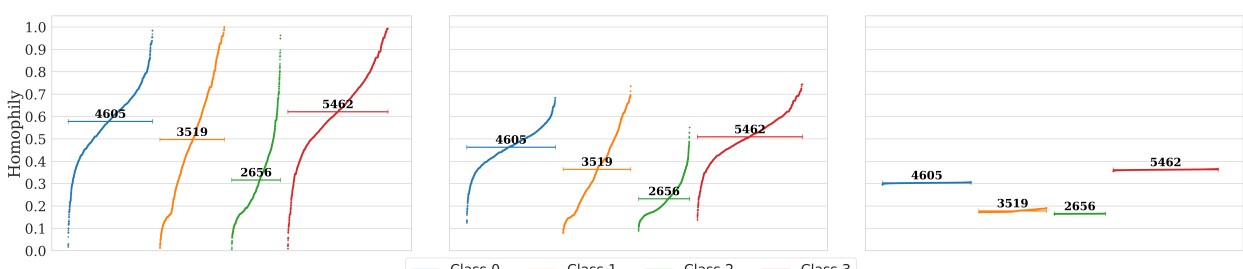

Figure 15: Node Homophily Distribution Scores for 20NewsW100.

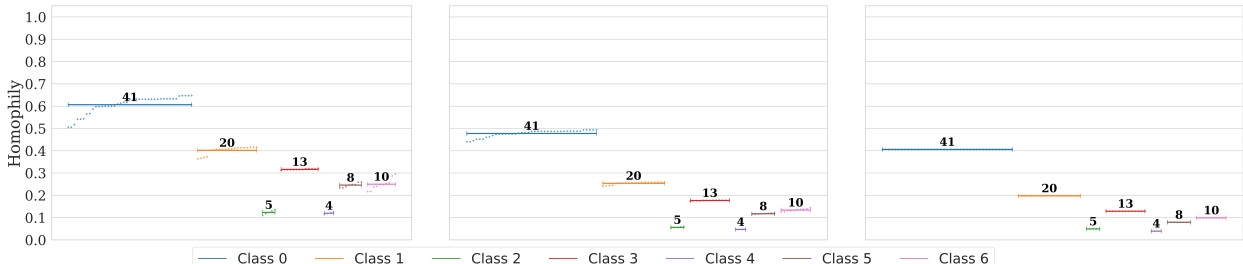

Figure 16: Node Homophily Distribution Scores for ZOO.

# H  Comparisons with others Homophily measure in literature

**K-uniform homophily measure**

**Hyperedge homophily.**  Veldt et al. (2023) defines the group homophily measure for $k$-uniform hypergraphs as $G_k = (\mathcal{V}, \mathcal{E})$. The type $t$-affinity score for each $t \in \{1, \ldots, k\}$, indicates the likelihood of a node belonging to class $c$ participating in groups in which exactly $t$ group members also belong to class $c$ and defined as in equation 68. $d_t(v)$ is the number of hyperedges that $v$ belongs to with exactly $t$ members from class $c$. The authors also consider a standard baseline score $b_t(c)$, equation 69, that measures the probability that a class-$c$ node is in the group where $t$ members are from class $c$, given that the other $k-1$ nodes were chosen uniformly at random.

$$\eta_t(c) = \frac{\sum_{v:y_v=c} d_t(v)}{\sum_{v:y_v=c} d_v}, \qquad (68) \qquad\qquad b_t(c) = \frac{\binom{n_c-1}{t-1}\binom{n-n_c}{k-t}}{\binom{n-1}{k-1}} \qquad (69)$$

$n_c$ is the number of nodes in class $c$ and $n$ is the total number of nodes in the hypergraph. The $k$-uniform hypergraph homophily measure can be expressed as a ratio of affinity and baseline scores, with a ratio value of 1 indicating that the group is formed uniformly at random, while any other number indicates that group interactions are either overexpressed or underexpressed for class $c$.

They suggest three possible ways for extending the concept of homophily to the hypergraph context. The first one is the *simple* homophily and it means that $\eta_t(c) > b_t(c)$ for $t = k$ and check whether a class has a higher-than-baseline affinity for group interactions that only involve members of their class. The second one is *order-j majority* homophily that is obtained when the top $j$ affinity scores for one class are higher than the baseline, i.e. $\eta_{k-j+1}(c) > b_{k-j+1}(c), \ldots, \eta_k(c) > b_k(c)$. The last one they consider is *order-j monotonic* homophily, which corresponds to the case when top $j$ ratio scores are increasing monotonic, i.e. $\eta_k(c)/b_k(c) > \eta_{k-1}(c)/b_{k-1}(c) > \cdots > \eta_{k-j+1}(c) > b_{k-j+1}(c)$.

Finally, considering that the value $\eta_t(c) - b_t(c)$ is the *bias* of class $c$ for type $t$, they introduce a type-$t$ *normalized bias* score that normalizes the maximum possible bias, hence the obtained metric is bounded in $[0, 1]$ and it is computed as:

$$f_t(c) = \begin{cases} \frac{\eta_t(c) - b_t(c)}{1 - b_t(c)} & \text{if } \eta_t(c) \geq b_t(c) \\ \frac{\eta_t(c) - b_t(c)}{b_t(c)} & \text{if } \eta_t(c) < b_t(c) \end{cases} \qquad (70)$$

**Comparisons to our measure** Unlike Veldt et al. (2023) our measure of homophily does not assume a k-uniform hypergraph structure and can be defined for any hypergraph. Furthermore, the proposed measure enables the definition of a score for each node and hyperedge for any neighborhood resolution, i.e., the connectivity of the hypergraph can be explicitly investigated. It gives a definition of homophily that puts more emphasis on the connections following the two-step message passing mechanism starting from the hyperedges of the hypergraph.

### H.1 Figure - $k$-uniform homophily

Some of the hypergraphs have a lot of different size for hyperedges, here, we report the plots only for some $k$ for each dataset for brevity. All additional plots can be found in the supplementary material's zip files.

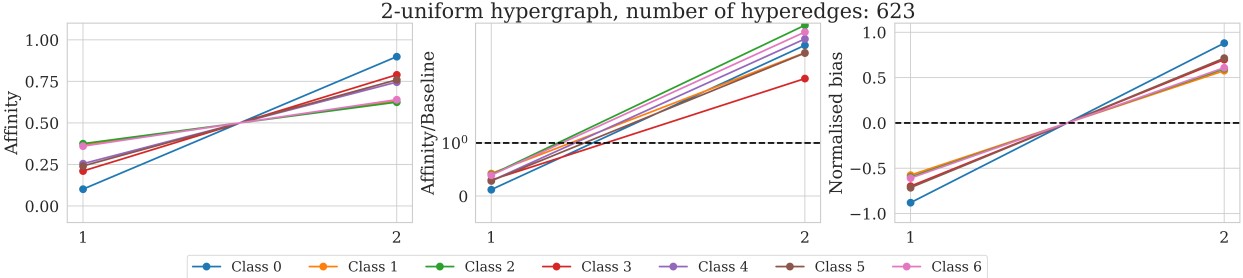

Figure 17: $k$-uniform homophily Cora.

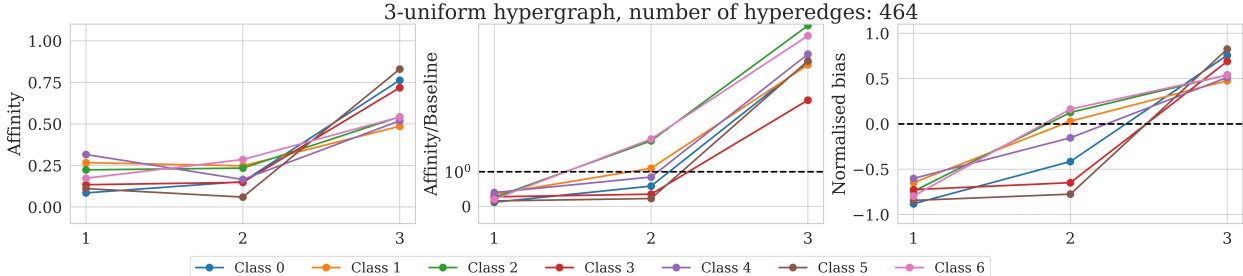

Figure 18: $k$-uniform homophily Cora.

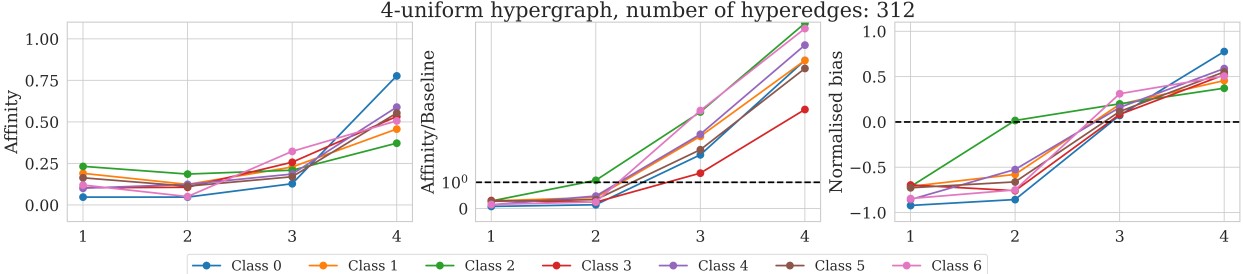

Figure 19: $k$-uniform homophily Cora.

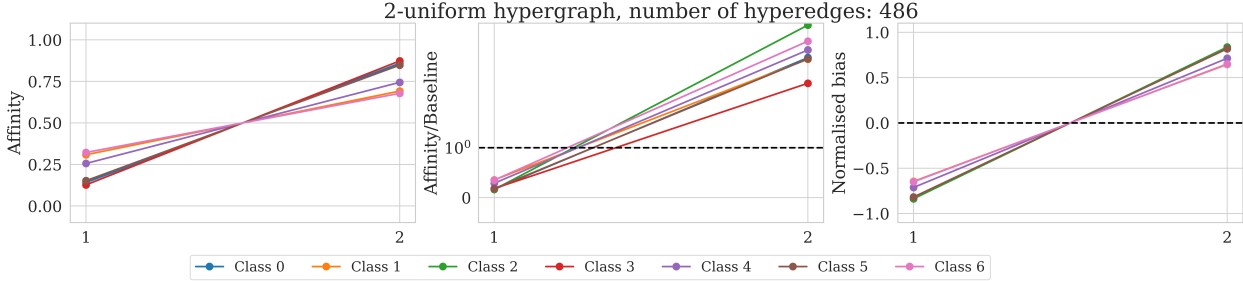

Figure 20: $k$-uniform homophily CORA-CA.

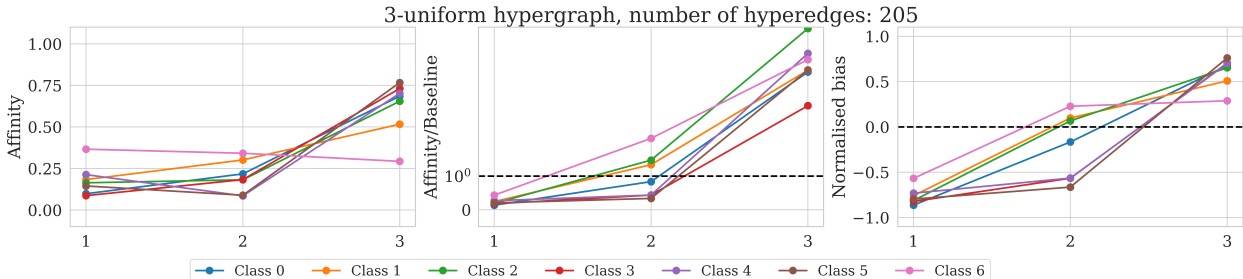

Figure 21: $k$-uniform homophily CORA-CA.

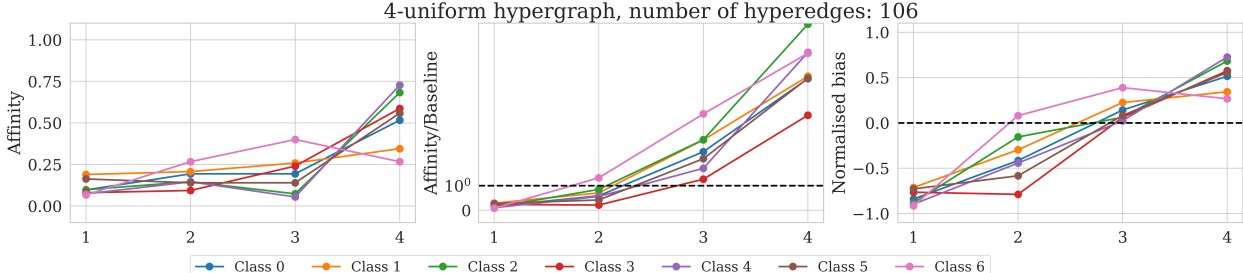

Figure 22: $k$-uniform homophily CORA-CA.

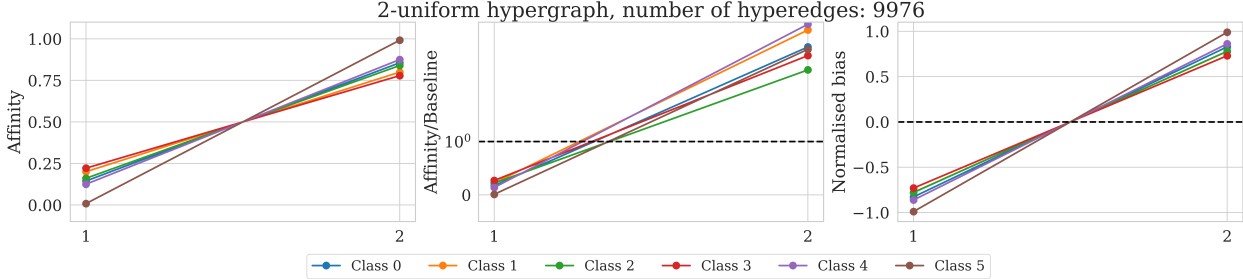

Figure 23: *k*-uniform homophily DBLP-CA.

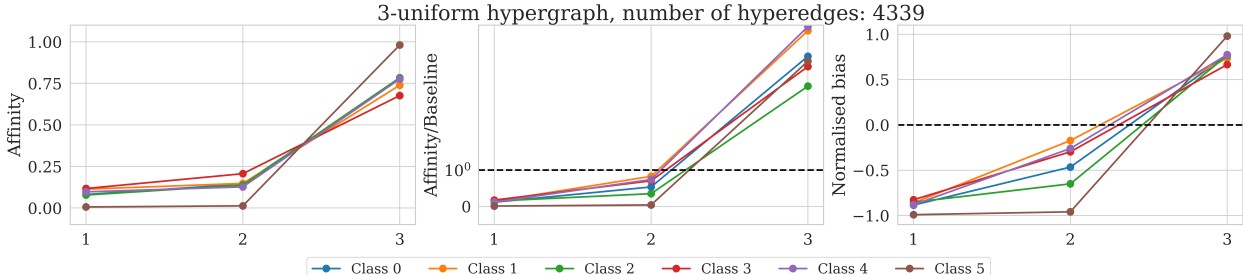

Figure 24: *k*-uniform homophily DBLP-CA.

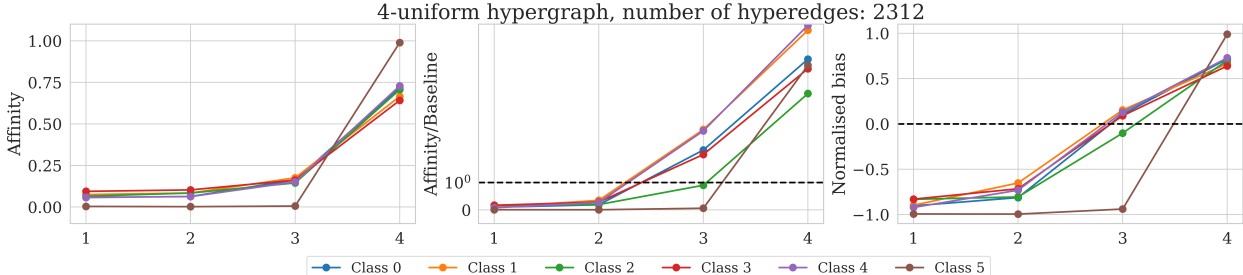

Figure 25: *k*-uniform homophily DBLP-CA.

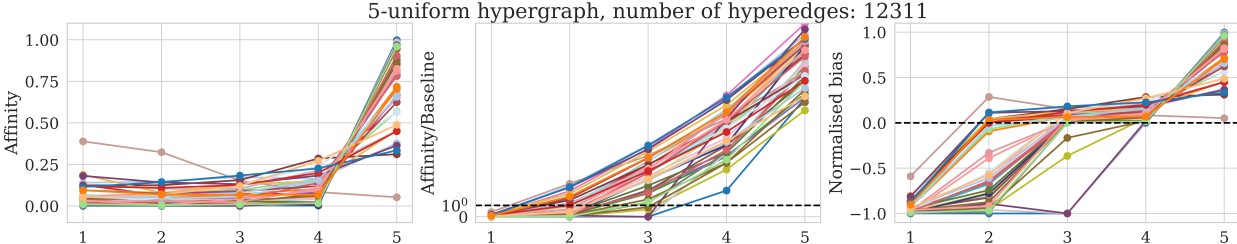

Figure 26: $k$-uniform homophily ModelNet40.

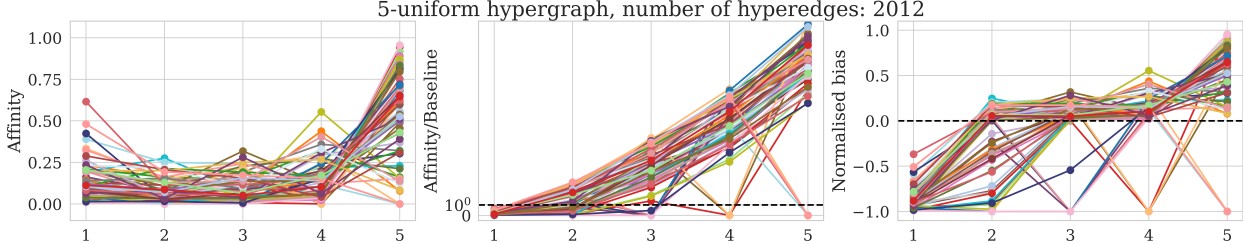

Figure 27: $k$-uniform homophily NTU2012.

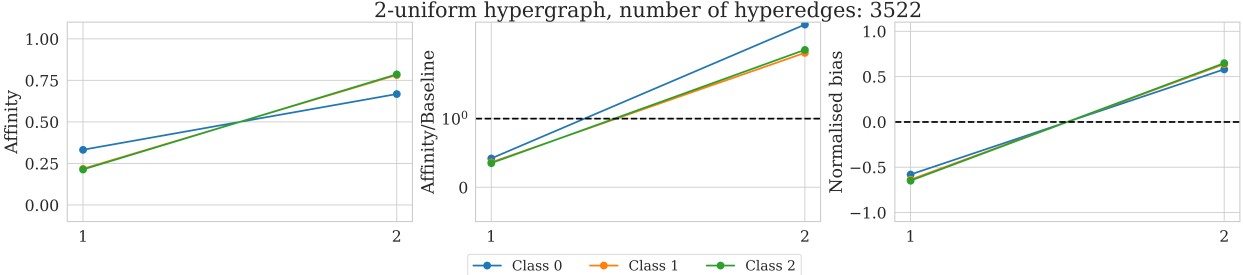

Figure 28: $k$-uniform homophily Pubmed.

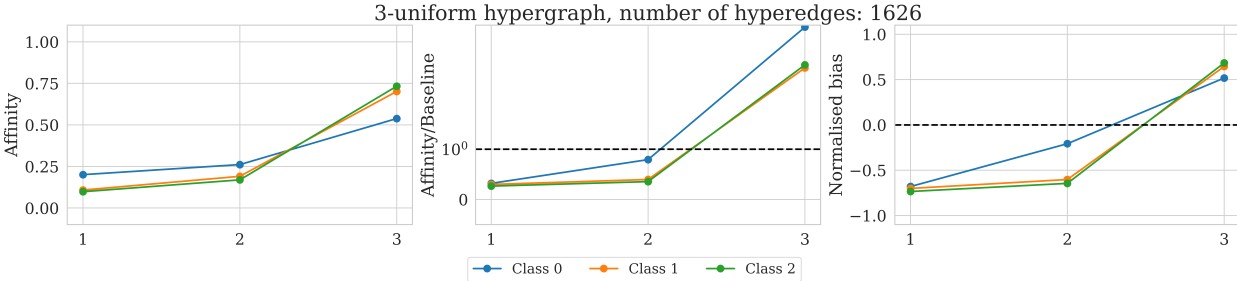

Figure 29: $k$-uniform homophily Pubmed.

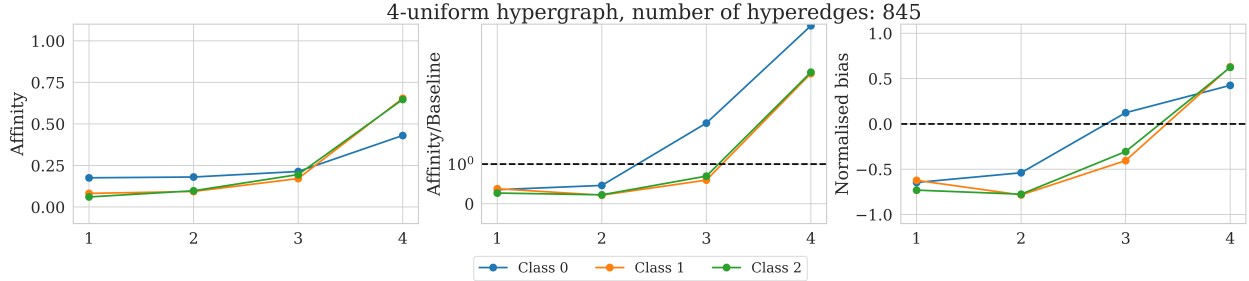

Figure 30: $k$-uniform homophily Pubmed.

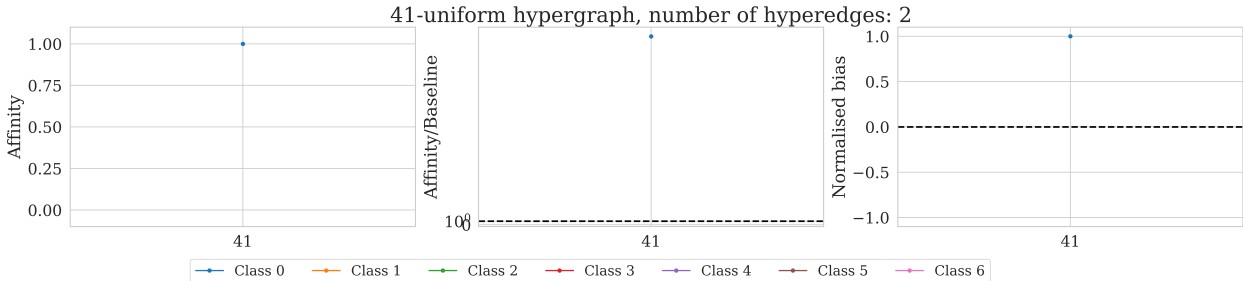

Figure 31: $k$-uniform homophily ZOO.

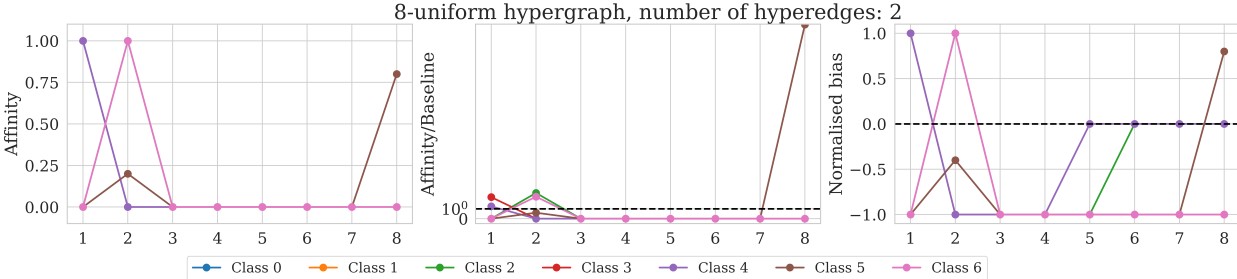

Figure 32: $k$-uniform homophily ZOO.

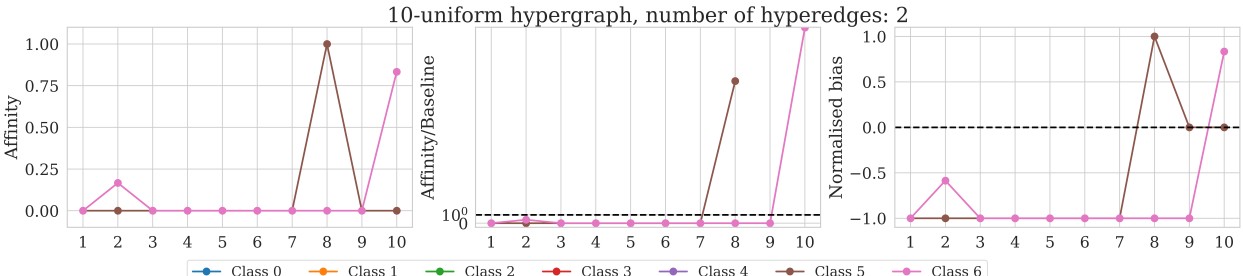

Figure 33: $k$-uniform homophily ZOO.

**Social equivalence**

The social equivalence Sun et al. (2023) is calculated through an expectation taken over pairs of users sampled from probability distributions. Specifically, $E(u, v \sim P(E_p))$ represents the expectation over positive user pairs sampled from the distribution of the hyperedges of the hypergraph $P(E_p)$. In contrast, $E(u_0, v_0 \sim P(V \times V \setminus E_p))$ represents the expectation over negative user pairs sampled from $P(V \times V \setminus E_p)$. The measure is a fraction between the numerator that involves the expected Jaccard index of environments for positive user pairs and the denominator, which comprises a similar calculation for negative user pairs. The Jaccard index is used to measure the similarity between two sets, and in this context, it assesses the similarity of hyperedges associated with positive user pairs. If the expected Jaccard index for positive user pairs is higher than that for negative pairs, the measure exceeds 1, indicating a significant level of observed social equivalence among users.

**Comparisons to our measure** The concept of Social Equivalence, as introduced by Sun et al. (2023), differs significantly from our definition of homophily. In our approach, the measure is initially defined at both the node and hyperedge levels. Our primary objective is to address the question: 'How similar a node is to its neighbors?' Following a message passing scheme, our definition allows us to examine different time points, attempting to answer how similar a node is to the nodes it can reach with $t$ steps of message passing. This consideration also extends to edges, where we seek to understand the coherence or uniformity of an edge within itself. The guiding notion of similarity in our work is belonging to the same class. Specifically, in level-0 homophily, we evaluate whether a node is more connected with nodes of the same classes. We can then aggregate the node-level/hyperedge-level homophily to provide a definition for hypergraph homophily. In contrast, in the work of Sun et al. (2023), the concept of social equivalence yields a single result for the entire hypergraph. The approach involves comparing the set of similar nodes that are connected with those that are not connected. The key question is whether the set of non-connected nodes is, on average, more similar or if the set of connected pairs exhibits greater similarity. It's important to note that this definition makes sense only for the entire hypergraph and captures a different notion of similarity.

**Social conformity**

Social conformity, as described in Sun et al. (2023), involves leveraging learned representations of users and hyperedges within a model to understand and quantify the level of conformity among users in a social network.

**Node homophily computed on the clique expansion of the hypergraph**

Clique-expanded (CE) homophily, employed in Wang et al. (2023), is determined by calculating node homophily Pei et al. (2020) on the graph derived from the clique-expanded hypergraphs.

**Comparisons to our measure** The node homophily for a graph computes the fraction of neighbors with the same class for all nodes and then averages these values across the nodes. In contrast to our metric, CE homophily is not defined directly on the hypergraph but necessitates an expanded clique representation. While sharing some similarities with our node-wise measure $h_0$, it lacks the dynamic aspect inherent in the MP homophily measure, consequently failing to capture the dynamic information within connections. Our analysis in Section 3 underscores the significance of this dynamic element in understanding the correlation between homophily measures and the observed patterns in Hypergraph Neural Networks (HNNs).

**Further consideration on the concepts of simulated social environment evolving and group entropy from Sun et al. (2023)**

Further exploration of the concepts of simulated social environment evolving and group entropy is presented by Sun et al. (Sun et al., 2023). In their study, a dynamic analysis of specific hypergraph characteristics is conducted through message passing. They specifically focus on the evolving proportion of 'significant nodes' within the hyperedge relative to the original nodes across different epochs. These 'significant nodes' are identified as those with a probability of belonging to the hyperedge greater than 0.5, initially determined by multiplying the representation of the node with that of the hyperedge (averaged over its constituent nodes).

A noteworthy distinction from our methodology lies in their reliance on representations provided by a model, in contrast to our representation-independent approach. Despite the shared use of message passing in both approaches, we underscore these methodological differences.

It's important to highlight that the concept of group entropy introduced by Sun et al. is also noteworthy, representing an evolving model concept; however, its computation requires node representations provided by a model.

We posit that our measure and the metrics employed in Sun et al. (2023)'s paper can complement each other effectively.

# I    Class Distribution Shift

We now report the results for the class distribution shift obtained by applying the mini-batch sampling procedure described in Section B. For each dataset, we choose to illustrate 3 different distributions: the one corresponding to the original labels ("Node"), the one obtained by applying both *Step 1* and *Step 2* described in the mini-batch paragraph of Section 4.2 and the one obtained by only applying *Step 2*.

## I.1    Figure - node homophily

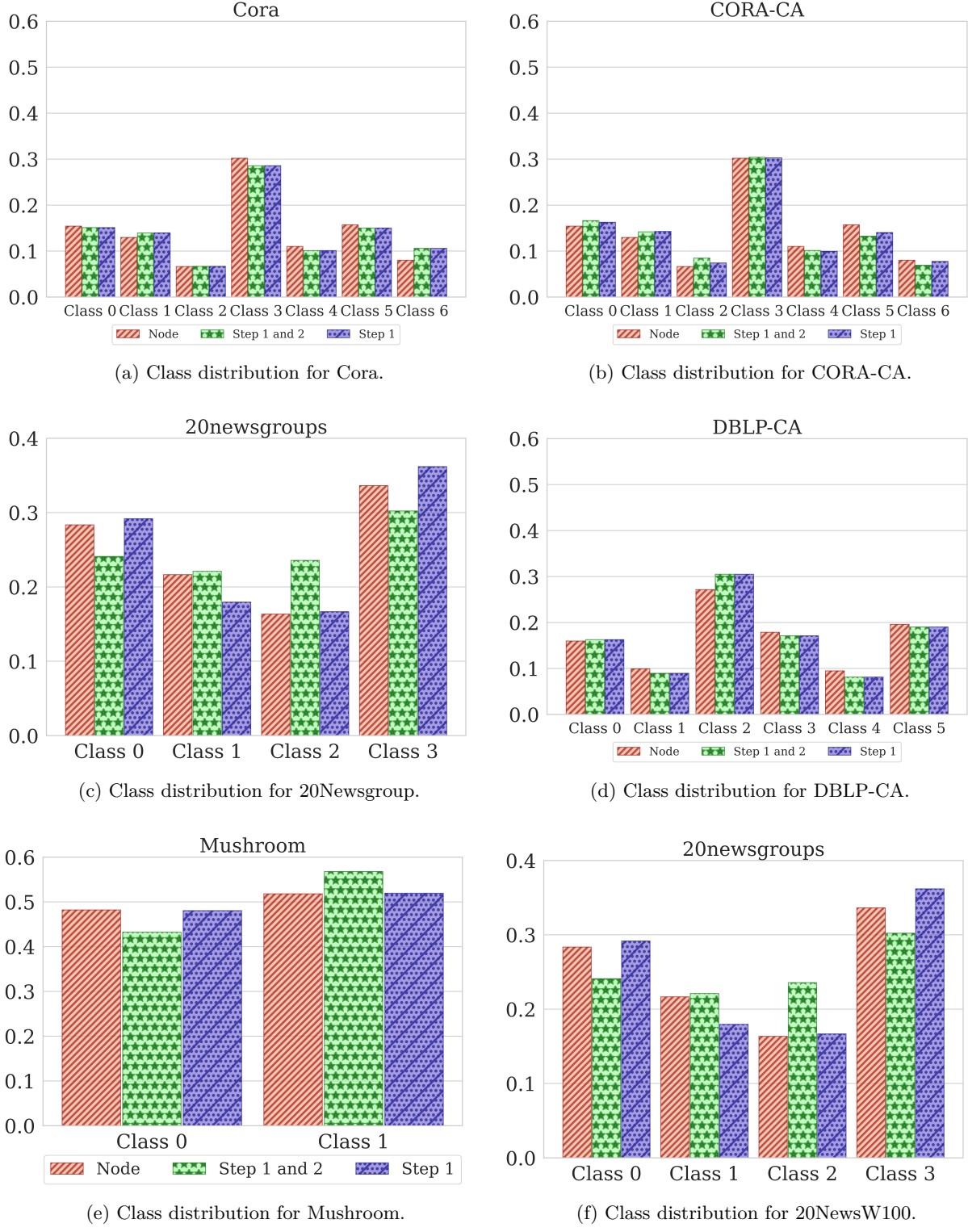

(a) Class distribution for Cora.

(b) Class distribution for CORA-CA.

(c) Class distribution for 20Newsgroup.

(d) Class distribution for DBLP-CA.

(e) Class distribution for Mushroom.

(f) Class distribution for 20NewsW100.

Figure 34: Class distribution shifts.

