# OpenReview forum: "Hypergraph Neural Networks through the Lens of Message Passing: A Common Perspective to Homophily and Architecture Design"
_TMLR — Accepted by TMLR_

### Review · Reviewer_CN4Y · 2024-12-05

**Summary Of Contributions:**

In this work, the authors explore and advance the broader field of hypergraph neural networks. In particular, the authors seek to answer three questions, regarding (a) the role that homophily (which is commonly considered in graph learning) can play in hypergraph tasks, (b) how previous attempts at hypergraph neural networks can be unified into a new architectural design that is more specific to the hypergraph setting, and (c) the quality of existing hypergraph benchmark datasets in terms of how well they are suited for measuring progress on hypergraph neural networks. The authors provide extensive empirical analysis to answer these questions.

**Audience:**

Yes

**Claims And Evidence:**

Yes

**Requested Changes:**

My weaknesses above largely reflect improvements that I would appreciate (while none are critical to my recommendation).

In addition, I want to point the authors to some minor points I noticed while reading the paper:

* At the beginning of 4.1, the authors write "We say that a function $f$ is a multiset function if it is permutation invariant w.r.t. each of its arguments in turn." From what I can see, this definition does not characterize a multiset function (which to me is a function which receives a multiset as input). Perhaps I misunderstood what the authors mean here, but this statement could likely be made more precise.
* 4.4, in the third paragraph, you write "Step 0". Do you mean Step 1? I was not able to find where a Step 0 was described.
* Last sentence of 5.3: the word "memory" appears twice.

**Strengths And Weaknesses:**

**Strengths:**

* The empirical study is extensive, considering both a large variety of hypergraph neural networks, simple baselines such as learning only on the individual features, as well as a large variety of datasets.
* In addition, the empirical analysis contains many ablations aimed at answering the three main questions posed in the abstract.
* The writing is clear and the entire paper is written along the narrative of answering the three main questions, which allows readers to quickly understand the importance of different sections. This is particularly the case in the experimental section.

**Weaknesses:**

* Compared to the detailed descriptions of the other experiments, the rewiring experiments receive little attention. Here, I would suggest to expand the description of the two rewiring approaches, perhaps even formally defining them. While I can roughly follow what the authors propose here, I am not too sure about the details after reading 5.4.2.
* Table 5 is a bit difficult to parse. In particular, since the different drop connectivity variants are evaluated at multiple levels, line plots would additionally improve the presentation of these results (with the x-axis showing the level of drop connectivity and the y-axis showing the performance). While of course performing such line plots for all methods and datasets would take up a lot of space, I would still recommend to show plots for the most important baselines and datasets (or a selection thereof). This, together with the raw numerical data in Table 5, would likely improve the presentation of these results.
* I had a hard time understanding Figure 1, in particular, because the Figure lacks an x-axis. I assume the authors sort every node in a dataset according to class and then plot the distribution of homophily scores for each? I think this should be described more clearly, for example in the figure caption.

---

> ### Author Response · Authors · 2024-12-21
> **Review response**
>
> Dear Reviewer CN4Y,
>
> Thank you for your time and thoughtful feedback. We were pleased to see your recognition of the extensive empirical study, thorough ablations, and clear writing style. Below, we address the points raised in your review.
>
> **Question 1**
> 1) We have refactored Section 5.4.2, improving the description and providing more details.
>
> 2) Thank you for your valuable suggestion regarding Table 5. We have incorporated some line plots for selected datasets and models in Figure 6 (Appendix E.2), which we agree greatly complement the extensive Table 5. By looking at these plots, we can see that HNN models show little deviation from the corresponding baselines, regardless of the rewiring strategy. In contrast, CEGCN demonstrates significant improvement, particularly with the trimming and random strategies.
>
>
> 3) Thank you for raising your concern regarding  Figure 1, we have updated the caption, highlighting that for each dataset class, points are sorted in ascending order of homophily and visualized sequentially along the x-axis.
>
>
> **Question 2**
>
> Regarding the multiset definition in Section 4.1, we acknowledge your point concerning the definition of the multiset function. We have adhered to the definition from the AllSet paper [1]  (see Definition 3.2) for consistency.
>
> Regarding your second comment ("Step 0"), you are correct: we indeed meant “Step 1”. Thank you for pointing this out! We have updated this, along with other typos you identified, in the revised manuscript.
>
> **References:**
>
> [1] “YOU ARE ALLSET: A MULTISET LEARNING FRAMEWORK FOR HYPERGRAPH NEURAL NETWORKS” Eli Chien et al.,

---

### Review · Reviewer_nGeB · 2024-12-07

**Summary Of Contributions:**

The paper proposes three main contributions (1) a measure of homophily in hyper graphs that can be used to evaluate the degree to which hyper node connectivity is given by hyper node labels, (2) MultiSetMixer — a hyper graph neural network architecture that explicitly considers hyper graph connectivity by representing nodes according to their learnable representations in all the hyper edges that contain them, and (3) a batching approach to reduce memory consumption when training HNNs.

Relying on the three components, the authors highlight that existing approaches to hypergraph representation learning do not fully consider hypergraph attributes. Based on their experimental results, the authors suggest that the proposed message-passing homophily measure can help identify properties of hyper graphs where existing models are limited and not expected to perform well, which can motivate future work.

**Audience:**

Yes

**Broader Impact Concerns:**

N/A, no ethical concerns.

**Claims And Evidence:**

No

**Requested Changes:**

See weaknesses and additional questions. In order of importance, the main adjustments I would suggest for the submission are:
* Explain how the datasets where MultiSetMixer perform favourably are constructed (particularly 20Newsgroups) and either adjust the claims or provide additional empirical justification for the method being broadly applicable compared to 'lifted' approaches. This could come in the form of results in additional datasets (incl. a synthetic dataset that baseline models can't beat, see Additional Question 4), extended analyses to explain the disparity in MultiSetMixer performance across dataset types, or an understanding on why existing baselines perform well where MultiSetMixer performs stat. sig. worse than baselines.
* Explain how parameters were chosen and provide, if possible, additional information and results about the impact of changing key parameters such as \mu or the choice of aggregation function.
* Evaluate the impact of training MultiSetMixer without the proposed batching approach. Practically, this would introduce a new row in Table 4 presenting the memory consumption of MultiSetMixer as a drop-in replacement of AllSetTransformer. This would help understand the impact of the proposed batching on its own, and would let practitioners assess whether the batching approach alone can be helpful in other hypergraph representation learning works.

**Strengths And Weaknesses:**

### Strengths:
* Overall: The paper clearly identifies limitations with existing methods for learning on hypergraphs and outlines a proposal to address these limitations.
* Claims: the proposed methods, model architecture, and experiments are reasonable and well justified (see Weaknesses for concerns about the claims _based_ on the experiments).
* Proofs: formal results appear to be correct, with the connection between MultiSetMixer and existing models being well defined. The proof that MultiSetMixer is a generalization of existing methods is correct and identifies weaknesses of existing approaches.
* Experiments: the authors conduct experiments on 11 datasets and 16 modelling approaches, plus several perturbation methods to evaluate the predictive performance, memory cost, and susceptibility to connectivity changes of different HNN models.
* Readability and form: the paper is clear, well written and is structured in a coherent, readable manner.
* Broader interest: the authors highlight the need to identify when and where hypergraph-specific aspects are required for modelling, and provide theoretical and practical examples where ‘lifted’ graph neural network architectures applied to hyper graphs are limited in their modelling power.

### Weaknesses:
* In its current state, the paper's claims about the advantages of hypergraph-specific architectures are not fully supported by the empirical results. The proposed method outperforms baselines only on datasets where the hyper graphs are constructed from non-relational data (NTU2012, ModelNet40, and 20Newsgroups), where two computer vision and one text classification datasets are cast as hyper graph problems. In contrast, in relational datasets that naturally contain nodes and hyper-edges like citation and co-authorship networks, the proposed method does not outperform the baselines. The authors claim that “features are being more representative than connectivity in most considered hypergraph datasets”, but this can be turned on its head to ask whether the approach to measure connectivity is meaningful given performance in these datasets.
* Certain parameters seem to have been picked without an explanation (e.g. \mu, see Additional Question 1.) but control key aspects of how the homophily measure functions and the results that follow from it, i.e. Figure 4.
* The sensitivity of the model to various hyperparameters (beyond \mu) is not thoroughly explored, particularly the impact of the mixer-block design choices and aggregation functions on model performance and memory usage (see Additional Question 2.).
* The evaluation of scalability for MultiSetMixer is tangled with the proposed batching approach. These two contributions should be studied separately through:
    * In Section 5.3, the authors compare MultiSetMixer with their proposed batching approach versus AllSetTransformer with and without hyper edge batching.
    * It would be valuable to understand what is the performance of MultiSetMixer with ‘vanilla’ batching to further motivate the need for hypergraph-aware batching and training procedures, and to isolate the memory impact of each component versus the baseline.
* While the technical foundations are sound, the paper does not conclusively demonstrate that this more complex approach provides meaningful advantages over simpler methods for practical hypergraph learning problems, especially given concerns regarding dataset construction (see Additional Questions 3 and 4).

### Additional questions:
1. How is the value of \mu chosen? The authors rely on this threshold to determine whether their proposed homophily value changes between propagation timestamp. However:
    * It is not clear how sensitive the reported results are to different values of \mu (e.g. what about \mu \in {0.01, 0.05, 0.15, 0.2}?).
    * The reported results in Figure 4. show the rank shift between CE Homophily and \Delta-homophily. The relationship between those metrics is hard to understand, especially as there is no table or figure summarising the values of \Delta-homophily (including Table 7. in the appendix, which reports _average_ homophily values at t=0 and t=1 for each graph).

2. How sensitive is MultiSetMixer to the choice of aggregation function? It is a well known result in the GNN literature that certain aggregation functions are more expressive than others for e.g. isomorphism detection. Exploring this question is important, as the mean is particularly known to be less expressive than the sum (see Xu et al. (2019) [1]).

3. How is the 20 Newsgroups dataset modelled as a hyper graph dataset? The study highlights MultiSetMixer’s comparably superior performance against the baselines, but the task being modelled and dataset being used are both not clear.
    * To my knowledge, [20 Newsgroups](https://archive.ics.uci.edu/dataset/113/twenty+newsgroups) is a standard 20-label multi-class text dataset where documents are represented by their contents, or, frequently, by TF-IDF representations of the terms in the collection. The task is to classify each document on one out of 20 disjoint classes.
    * Table 7 indicates that there are only 4 classes, and authors refer to Yadati et al. (2019)’s [2] approach to generate hyper graphs by defining a hyper-edge for all data points sharing a categorical feature.
    * Furthermore, in Section 5., authors report reusing the dataset and benchmarking approach provided by Chien et al. (2022) [3], which also refers to the UCI Categorical ML Repository without indicating the procedure, processing, or task adjustments made on the dataset. Chien et al.’s work also indicates that the cardinality of hyper node features is 100, which seems unexpectedly low for TF-IDF document representations (and implies a vocabulary of 100 terms).
    * Can the authors clarify what the dataset, features, classes and task are in this case? Otherwise, it is not possible to understand whether their results are applicable in a practical dataset, or instead a by-product of how the dataset was constructed.
    * Understanding the precise dataset construction is crucial because it would help clarify whether the performance gains come from the model architecture itself or from how the hypergraph structure was imposed on what is fundamentally a text classification task.

4. The direction of the work presented is valuable, and my concerns about performance per dataset and the provenance of data, indicate that existing benchmarks are not appropriate to assess the presented work. Thus, what properties should we look for in benchmark datasets? Have the authors identified any connectivity patterns that their method would perform well on, and that could be mapped to practical applications and datasets? In other words, it would be valuable to see if authors can identify or construct a dataset for which their method can perform well, but existing baselines cannot.

### Minor comments:
* In table 5, the avg. rank for CEGAT and CEGCN seem to be populated with data from a different column — across the whole column, avg. ranks are rounded in 0.05 intervals and have the best performing value per row on the first original, unperturbed value.

### References:
[1] Keyulu Xu, Weihua Hu, Jure Leskovec, & Stefanie Jegelka (2019). How Powerful are Graph Neural Networks?. In International Conference on Learning Representations.

[2] Yadati, N., Nimishakavi, M., Yadav, P., Nitin, V., Louis, A., & Talukdar, P. (2019). HyperGCN: A New Method For Training Graph Convolutional Networks on Hypergraphs. Advances in Neural Information Processing Systems (NeurIPS).

[3] Eli Chien, Chao Pan, Jianhao Peng, & Olgica Milenkovic (2022). You are AllSet: A Multiset Function Framework for Hypergraph Neural Networks. In International Conference on Learning Representations.

---

> ### Author Response · Authors · 2024-12-21
> **Review response**
>
> Dear Reviewer nGeB,
>
> Thank you for taking the time to review our paper. Your feedback is really valuable, as it has allowed us to further clarify several key points of our work. In particular, we have updated the description of the 20newsgroups dataset (Appendix D.2), added a discussion on the \mu parameter selection (see Appendix E.3), and evaluated MultiSetMixer training without the proposed batching approach with the corresponding update of Table 4.
>
> Regarding the minor comment:
> Thank you for your comment. We would like to clarify that the data presented in Table 5 for CEGAT and CEGCN is accurate. This phenomenon is discussed in detail in Sections 5.4.1 and 6 (Q1).
>
> **Regarding the 20newsgroups dataset:**
> This is a valid observation, and we indeed overlooked the proper description and reference. The 20newsgroups dataset at UCI is different from the one we are using. Our dataset is a subset of the original dataset, which we believe was first introduced in 2006 [1, Section 8].
>
> Hyperedge Construction: Each instance in the 20newsgroups dataset is represented by 100 attributes. Each attribute corresponds to a specific TF-IDF instance. A hyperedge with a weight of one is formed by grouping all data points that share the same value for a categorical feature. The weights for all hyperedges are uniformly set to 1.
>
> Dataset Size: In [1], the author presents a modified 20newsgroups dataset where data samples (nodes) have binary occurrence values for 100 words across 16,242 articles. These articles are categorized into four topics corresponding to the highest level of the original 20 newsgroups (which can be found at the UCI Categorical ML Repository). The sizes of the topic-specific groups are 4,605, 3,519, 2,657, and 5,461, respectively.
>
>  We have added additional clarification and included the appropriate citations in Appendix D.2.
>
> **References:**
>
> [1] “Learning with Hypergraphs: Clustering, Classification, and Embedding” by Dengyong Zhou et al.,
>
> [2] “The Total Variation on Hypergraphs - Learning on Hypergraphs Revisited” by Matthias Hein et al.,
>
> [3] “Semi-supervised Hypergraph Node Classification on Hypergraph Line Expansion” Chaoqi Yang et al.,
>
> [4] “YOU ARE ALLSET: A MULTISET LEARNING FRAMEWORK FOR HYPERGRAPH NEURAL NETWORKS” Eli Chien et al.,
>
> **Regarding the $\mu$ in $\Delta$-homophily: homophily:**
> Thank you for bringing up the question about the choice of $\mu$. This parameter has been a key focus in our experiments. To address your concern, we have included a plot that illustrates how Δ-homophily evolves as $\mu$ varies across multiple datasets.
>
> We first observe that Δ-homophily generally increases with $\mu$. However, conceived as a homophily framework, the relevant aspect here is the relative homophily values across datasets –and not the specific absolute values. In this regard, we can see that the relative homophilic ranking is quite stable for different values of $\mu$, and in particular around the 0.1 value (our choice for the homophily analysis in Section 5.2.) Please refer to Figure 7 in Appendix E.3 for sensitivity analysis of the Delta-homophily as a function of \mu across various datasets.
>
> **Regarding the aggregation function in the MP homophily framework:**
> For message-passing homophily, the “mean” aggregation function is the most straightforward choice. In contrast, the “sum” aggregation function may lead to unbounded values that depend on the size of the hyperedges. While the “max” aggregation function could be a potential option, it introduces challenges in defining delta-homophily, as the homophily change would manifest as a stepwise function.
>
> **Regarding the aggregation function in MultiSetMixer:**
> We primarily focused on the mean aggregation in our experiments, and have not yet systematically explored how MultiSetMixer’s performance varies with other aggregation functions such as sum. The usage of sum aggregation function can be problematic for very large hyperedges due to potential numerical instability and scaling issues –although theoretically allowing for greater expressivity. Normalization techniques (e.g., dividing by set size or applying other normalization layers) can help mitigate these challenges, and we agree this can be a meaningful analysis in follow-up work.
>
>
> 1/2

---

> ### Author Response · Authors · 2024-12-21
> **Review response**
>
> **Batching:**
> Thank you for this valuable suggestion. We evaluated MultiSetMixer training without the proposed batching approach and included the corresponding memory consumption results in Table 4. It can be seen that MultiSetMixer without batching might require greater memory consumption; however, it relies on the dataset's connectivity, i.e., the number of hyperedges and their average size.
>
>
> **Potential Benefits of MultiSetMixer:**
> Indeed, this comment opens an interesting discussion. While unfortunately we haven't found yet a particular dataset where MultiSetMixer really stand out in terms of performance, we do have identified some strengths that can guide the application either of the full model, or of some of its novel methodologies.
>
> For instance, the studied approach seems particularly advantageous for large and heterophilic datasets. The proposed batching method enables robust performance and allows the control of memory consumption (see Section 5.3), and the new experiments on heterophilic data (see Appendix E.1) demonstrate that the hyperedge-dependent node representations model consistently achieves strong average results on heterophilic data --performing on par with EDHNN and outperforming AllDeepSet and AllSetTransformer models.
>
> Furthermore, we believe the proposed MultiSet framework is highly beneficial for the recently introduced hyperedge-dependent node classification tasks, which we describe in the Related Works section. Last, but not least, in Section 4.2 we demonstrate that the MultiSet framework generalizes the recent state-of-the-art model, WHATSnew. Hence, the proposed techniques can potentially be incorporated and tested in the corresponding scenario.
>
>
> **References:**
>
> [1] “Classification of edge-dependent labels of nodes in hypergraphs” Choe et al.,
>
>
>
> **Regarding the following statement:** “The authors claim that “features are being more representative than connectivity in most considered hypergraph datasets”, but this can be turned on its head to ask whether the approach to measure connectivity is meaningful given performance in these datasets.”
>
> In Section 5.2, Table 2, we evaluate the impact of models that incorporate connectivity compared to those that rely solely on features. Additionally, in Section 5.4 (Table 5), we examine how changes in connectivity influence performance. This experiment aims to assess both the effect of utilizing inductive bias (connectivity) and the role of connectivity in determining the final performance. These experiments allow us to draw a conclusion that most HNNs models test to ignore the connectivity of Citeseer, Pubmed, Mushroom, 20Newsgroups.
>
>
> 2/2

---

> > ### Comment · Reviewer_nGeB · 2024-12-30
> > **Thank you for the additional comments.**
> >
> > Dear authors, thank you for your comments and helpful remarks. I will reply inline to both comments below.
> >
> > > **Re: 20newsgroups dataset**
> > > * A hyperedge with a weight of one is formed by grouping all data points that share the same value for a categorical feature. The weights for all hyperedges are uniformly set to 1.
> > > * Data samples (nodes) have binary occurrence values for 100 words across 16,242 articles.
> > > * These articles are categorized into four topics corresponding to the highest level of the original 20 newsgroups
> >
> > Thank you for tracking down the original source of the dataset. My original concern is whether this dataset is a representative way of capturing the ability of the proposed method to incorporate hyperedge information and perform well in hypergraph settings. As it stands, I believe this dataset is not representative of a real world task:
> > * A 100-term vocabulary is quite limited
> > * There seems to be redundancy between the data samples having binary occurrence values for the vocabulary, and hyper-edges for each term connecting all documents that contain it.
> > * It is not clear what the classes are in this case (I looked at the dataset again, and [there are more than 4 top level groups for the user-groups](http://kdd.ics.uci.edu/databases/20newsgroups/20newsgroups.data.html): `alt, comp, misc, rec, sci, soc, talk`)
> >
> > I would suggest being explicit about this in the manuscript, especially because I believe that although _artificial_, this dataset may indicate a genuine case for the limitations of other models. Ignoring both unclear class labels and small vocabulary, the task seems to model the following:
> > > Given a document with a set of terms that appear in it, and hyperedges connecting this document to all the documents that share any of the terms contained in it, can we predict its class?
> >
> > Correct me if I am mistaken: the difference of MultiSetMixer with other methods is that it represents each document (node) for each term with respect to all the documents containing that term (hyperedge). Each node -> hyperedge message can modulate 'this document is (di)similar to all other documents in the hyperedge'—which is likely a good signal for telling classes apart.
> >
> > I would expect to see an argument of this sort when motivating the results in Section 5.3 rather than class imbalance (which at a quick glance e.g. also holds for Cora, and likely others). This could be drawn from analysing whether my suggestion re: 20 Newsgroups holds, or from a synthetic dataset where baselines are unable to perform well _by design_ (e.g. constructing a task that consider nodes within each hyperedge that contains them in a way that baselines will oversquash meaningful signals).
> >
> > > **Re: $\mu$ in $\Delta$-homophily**
> >
> > The additions to the manuscript are helpful and show that the impact of $\mu$ is limited. I understood (and agree!) with the comment that what matters are relative $\Delta$-homophily values, so I was expecting e.g. the outcome of plotting Figure 4. with different values of $\mu$. That said, the paper and the appendix are already quite complete and I am satisfied with the analysis in Appendix E3, showing that the overall  $\Delta$-homophily per dataset is stable.
> >
> > > **Re: Aggregation functions for MP homophily framework and MultiSetMixer**
> >
> > I agree that the mean is the natural aggregation function for MP homophily as a measure of 'which proportion of nodes / hyper edges belong to the class label'. I also agree that studying the expressivity / normalisation approaches for aggregation methods in MultiSetMixer is valuable follow-up work—part of the question is whether you had performed any ablation studies yet, but I am satisfied with the manuscript as it is. I would look forward for additional theoretical results regarding expressivity, esp. if GNN findings do not extrapolate directly.
> >
> > > **Re: Batching**
> >
> > I appreciate for the extended results for MultiSetMixer without batching, it helps clarify the contribution of the batching approach on its own.
> >
> > > **Re: Potential Benefits of MultiSetMixer**
> >
> > I believe this discussion can be covered by my comment above regarding 20 Newsgroups, including a potential argument for MultiSetMixer. Note that regarding the claim:
> >
> > > These experiments allow us to draw a conclusion that most HNNs models test to ignore the connectivity of Citeseer, Pubmed, Mushroom, 20Newsgroups.
> >
> > Indeed, results in Table 1. show the performance of most HNNs are similar to the feature-only methods in some datasets. However, I would focus on the cases where this is _not_ the case (i.e. Cora, Cora-CA, DBLP) & the best performing baseline (EDHNN) leverages connectivity and after drop/rewiring still outperforms well—I'd expect an explanation for why MultiSetMixer underperforms in those cases.
> >
> > ----
> >
> > Other than continuing the discussion re: 20Newsgroups and how the impact of hyper-edge connectivity and features affect performance, I am satisfied with the changes introduced by the authors.

---

> > > ### Author Response · Authors · 2025-01-04
> > >
> > > Dear Reviewer nGeB, thank you for your response, and we are happy to hear that some of your concerns have been addressed.
> > >
> > > **Regarding 20newsgroup dataset:**
> > >
> > > Upon further exploration of the 20Newsgroups dataset, we discovered that Scikit-learn also provides a version of the dataset, which contains 18,846 nodes due to specific preprocessing. Unfortunately, we could not determine the exact filtering criteria used in [1] to reduce the sample count from 20,000 to 16,242 or the rationale for defining the number of classes. We hypothesize that this filtering process may have involved removing documents lacking sufficient content or relevance to the top 100 words.
> > >
> > > Additionally, we observed that the initial versions of the 20Newsgroups dataset (from both the UCI Repository and Scikit-learn) are balanced, with approximately 1,000 samples per class. In contrast, the Reduced 20Newsgroups dataset \citep{zhou2006learning} exhibits a skewed distribution, with 4,605, 3,519, 2,657, and 5,461 samples in the four groups, respectively. Based on further statistical analysis of the group distributions in the UCI and Scikit-learn dataset versions, we hypothesize that some of the original groups (e.g., alt, comp, misc, rec, sci, soc, talk) were merged during preprocessing.
> > >
> > > Acknowledging the importance of understanding the dataset characteristics, we have updated the manuscript to include these findings and a discussion of the issues with the 20Newsgroups dataset in Appendix D2.
> > >
> > >
> > > | **Version**                    | **Samples** | **Classes** | **Notes**                                                                                      |
> > > |--------------------------------|-------------|-------------|------------------------------------------------------------------------------------------------|
> > > | Scikit-learn                   | 18,846     | 20          | Preprocessed, removes duplicates, and provides full fine-grained classes.                    |
> > > | UCI Repository                 | 20,000     | 20          | Raw dataset with duplicates and no filtering.                                                |
> > > | Reduced 20newsgroup [1]   | 16,242     | 4           | Reduced classes (grouped into four categories) and filtered samples for hypergraph modeling. Unknown filtration criteria.    |
> > >
> > > | Category | UCI Repository | Scikit-learn |
> > > |----------|--------|--------|
> > > | alt      | 1000   | 799    |
> > > | comp     | 5000   | 4891   |
> > > | misc     | 1000   | 975    |
> > > | rec      | 4000   | 3979   |
> > > | sci      | 4000   | 3952   |
> > > | soc      | 997    | 997    |
> > > | talk     | 4000   | 3253   |
> > >
> > >
> > >
> > > **Regarding the MultiSet framework and MultiSetMixer:**
> > >
> > > Thank you for your observations about the MultiSet framework and MultiSetMixer. You're right on both counts - regarding MultiSet's enhanced signal modeling capabilities and the multiple factors affecting MultiSetMixer's performance.
> > >
> > > In Section 4.2 (Proposition 4.4), we prove that MultiSet generalizes the WHATsNet architecture, which processes hypergraph networks using the MultiSet formulation. This approach has demonstrated superior performance on hyperedge-dependent node classification tasks where baseline models tend to oversquash meaningful signals [2]. Regarding MultiSetMixer's performance, you're correct that the distribution shift is not the only determining factor. We've expanded our discussion of this into two sections:
> > >
> > > - Section 5.3 now directs readers to Section 5.5 for a comprehensive analysis of hyperedge-dependent node representations in hypergraphs.
> > >
> > > - We have updated the second and third paragraphs of Section 5.5, which now contain an expanded analysis of MultiSetMixer's performance across different datasets, along with a detailed discussion of the benefits and challenges of hyperedge-dependent node representations.
> > >
> > > We want to thank you again for your thoughtful comments and for engaging in this valuable discussion!
> > >
> > > **References:**
> > >
> > > [1] “Learning with Hypergraphs: Clustering, Classification, and Embedding” by Dengyong Zhou et al.,
> > >
> > > [2] “Classification of edge-dependent labels of nodes in hypergraphs” Choe et al.,

---

> > > > ### Comment · Reviewer_nGeB · 2025-01-05
> > > > **Thank you.**
> > > >
> > > > Dear authors, thank you for addressing my remaining concerns and for the detailed discussion.
> > > >
> > > > I believe the latest changes to section 5.5 provide a thorough, balanced analysis which was missing in the original submission. In my opinion, the claims in the paper are better presented and substantiated now, and I am satisfied with the status of the submitted manuscript. Thus, I recommend acceptance as I believe both claims and interest criteria are satisfied.
> > > >
> > > > I also thank you and want to highlight the importance of the additional information you added about the 20 Newsgroups dataset, which helps readers understand the experimental claims. It also shows how as a community we experiment with datasets and tasks which, through no fault of your own, are not clearly described or understood and hard to reproduce. The extensions to Appendix D2 help remove some of the gaps, and address the pending concerns I had with experiments contained in the manuscript. I hope future works on HNNs will reproduce or reformulate the dataset and task, especially as the approach to generate the hyper graphs should be applicable to any multi class text classification dataset.

---

### Review · Reviewer_iCjf · 2024-12-10

**Summary Of Contributions:**

The paper introduces a novel definition of homophily for general hypergraphs ($\Delta Homophily$) and demonstrates that this measure correlates to HNN model performances. It further proposes MultiSet as a unified HNN framework that encompasses most prior architectures for hypergraphs. The paper also proposes specific mini-batch sampling strategies that improve the computational efficiency of MultiSet models. Finally, an experimental evaluation is conducted to compare the performance of HNNs in the context of homophily and investigate the efficacy of existing hypergraph benchmark datasets to assess model performance.

**Audience:**

Yes

**Claims And Evidence:**

Yes

**Requested Changes:**

1. Clarify the definition of the MultiSet Framework (see Weakness 1, critical)
2. Improve the presentation of the work (see Weakness 2+3)

**Strengths And Weaknesses:**

Strengths:
1. The proposed measure of $\Delta$ Homophily seems to characterize datasets effectively and is also quite intuitive.
2. The experiments are thorough and insightful.
3. The discussion clearly links the numerous experimental results to specific research questions.

Weaknesses:
1. I think there is a discrepancy in Equation 8. It states that $\mathbb{X}_u^{(t)}$ is used to update $\mathbf{z}_e$. Based on Equation 11 it seems to me like this should say $\mathbf{x}^{(t)} _{u,e}$. Figure 2b on the other hand suggests that $\mathbf{x}_v^{(t)}$ is used to update $\mathbf{z}_e$. Overall, this seems somewhat confusing. Is this a notation error or did I misunderstand the MultiSet framework?
2. In Section 4.4 the second to last paragraph refers to "step  0" and "step 1" while other paragraphs use "step 1-2".
3. The manuscript also has some typos that should be fixed ("a a hyperedge"  (Page 4), "We represents by" (Page 4))

---

> ### Author Response · Authors · 2024-12-21
> **Review response**
>
> Dear reviewer iCjf,
> We want to thank you for your valuable review and for appreciating the strengths of our submission.  We address your points below.
>
> **Question 1**
> Regarding the ``discrepancy'' between Eq8 and Eq11:
>
> Eq8 provides the general theoretical MultiSet framework, allowing the update of ${z}\_{e}$  to be based on any subset of vectors from $\mathbb{X}\_{v}\^{(t)}= \{ x\_{v,e}\^{(t)} \}\_{e \in \mathcal{E}\_v}$. Eq11, on the other hand, is a specific instantiation of this general approach, achieved by applying certain practical design choices. In this particular implementation, the function $f\_{\mathcal{V} \rightarrow \mathcal{E}}$ is chosen to depend only on a single subset of $\mathbb{X}\_{v}^{(t)}$, namely ${x}\_{v,e}^{(t)}$ itself. More broadly, any model that uses a subset of $\mathbb{X}\_{v}^{(t)}$ within Eq8 can be viewed as a special case of the MultiSet framework.
>
> In Figure 2b, we recognize that it appears as though ${z}\_e$ is updated solely with the shared node features ${x}\_v$​, rather than the hyperedge-specific features ${x}\_{v,e}$.
> This occurs because we have serialized the update of only one layer and avoided clutter in the notation at step $t$. As shown at the top of the figure, we obtain ${x}\_{v\_{2}, {e}\_{1}^{t+1}}$ and ${x}\_{v\_{2}, {e}\_{2}^{t+1}}$, which can subsequently be used to update the corresponding hyperedges and nodes. We chose to simplify the diagram this way to emphasize which hyperedges each node belongs to, but in practice, hyperedge-dependent node features do factor into the model. We have additionally added the clarification in Section 4.2.
>
> **Question 2**
> Thank you for your observations; we have corrected all the typos and additionally performed a review of the entire document to ensure its quality.

---

### Author Response · Authors · 2024-12-10

Dear Editors,

We have received three reviews for our submission, with the two-week response period set to end on December 24. Given the upcoming holiday season, we kindly request an extension to avoid overlap with the holidays and to ensure sufficient time to prepare a thorough response.

---

### Author Response · Authors · 2024-12-21
**General response**

We would like to thank all reviewers for their detailed evaluations. We are grateful for the time and attention they dedicated to our work. We look forward to continuing our discussions and providing further clarifications upon request.

We have addressed the reviewers' comments and made the necessary corrections to the manuscript, with all updates highlighted in blue for your convenience. The caption of Figure 1 has been updated, and the description of the rewiring connectivity experiment in Section 5.4.2 has been revised. Section 4.2 has been updated, adding clarification to Figure 2.  We have also added Table 4 and included Appendix E.1, E.2, and E.3, where E.1 presents additional experiments with heterophilic datasets;  E.2 expands the analysis of the rewiring experiment, and E.3 provides a sensitivity analysis of the Delta-homophily as a function of \mu across various datasets. Additionally, Appendix D.2 now contains a more detailed description of the 20Newsgroups dataset construction.

Please find individual, point-by-point replies to all the comments below. Once more, we would be pleased to engage in deeper discussions that could help answer your concerns and further improve the quality of our manuscript.

---

> ### Author Response · Authors · 2024-12-27
>
> Dear Reviewers,
>
> We would like to sincerely apologize for our delayed response, as we had been anticipating a possible review extension during the holiday period but did not receive confirmation.
>
> In the meantime, we wish to kindly remind you that we are more than happy to engage in further discussions or address any additional questions or concerns you may have regarding the paper.
>
> Once more, we greatly appreciate your time and efforts in reviewing our work.

---

### Decision · Action_Editor_NNjJ · 2025-02-02

**Recommendation:** Accept as is

**Comment:**

The reviewers initially raised several concerns about the original submission, including unclear presentation of certain technical aspects and experiments, insufficient experimental support for the claims, a lack of analysis or justification for hyperparameters, and the absence of important ablation studies in model learning. In their response, the authors addressed these concerns effectively by improving the overall clarity, providing additional experiments and thorough analysis, and including details on dataset construction.

Following the discussion phase, the paper received a consensus of positive recommendations from all three reviewers (2 Accept and 1 Leaning Accept), reflecting its key strengths:
1. The work is well-motivated, identifying and addressing limitations in existing methods for hypergraph learning.
2. The proposed methods, model architecture, theoretical results, and experiments are well-justified and reasonable.
3. The experimental evaluations are comprehensive and convincing, and the analysis provides valuable insights.
4. The paper is clearly and effectively written.

The AE agrees with the reviewers' recommendation to accept the paper, noting that the authors have effectively addressed the initial concerns. The work presents valuable contributions to the ML community, supported by clear evidence that substantiates its claims, thereby meeting the standards of TMLR.

**Audience:**

The topic of hypergraph Neural Networks is of interest to the geometric deep learning/graph learning community.

**Claims And Evidence:**

The paper addresses three open questions in hypergraph network learning and offers three key contributions: (1) a novel definition of homophily for hypergraphs that correlates with HNN model performance; (2) MultiSet, a unified HNN framework that generalizes most prior hypergraph models, along with mini-batch sampling strategies to enhance computational efficiency; and (3) a comprehensive set of experiments evaluating HNN performance and the efficacy of existing hypergraph benchmark datasets.

After the author rebuttal and discussion, the paper received unanimous approval from all three reviewers. The work is well-motivated, clearly written, and substantiates its theoretical and experimental claims with sufficient evidence. The proposed framework is technically sound, provides novel insights, and includes a thorough empirical study that supports its claims.